# Tiered Reinforcement Learning: Pessimism in the Face of Uncertainty and Constant Regret

**Jiawei Huang**[1*]   **Li Zhao**[2]   **Tao Qin**[2]   **Wei Chen**[2]   **Nan Jiang**[1]   **Tie-Yan Liu**[2]

[1] Department of Computer Science, University of Illinois at Urbana-Champaign
`{jiaweih, nanjiang}@illinois.edu`
[2] Microsoft Research Asia
`{lizo, taoqin, weic, tyliu}@microsoft.com`

## Abstract

We propose a new learning framework that captures the tiered structure of many real-world user-interaction applications, where the users can be divided into two groups based on their different tolerance on exploration risks and should be treated separately. In this setting, we simultaneously maintain two policies $\pi^O$ and $\pi^E$: $\pi^O$ ("O" for "online") interacts with more risk-tolerant users from the first tier and minimizes regret by balancing exploration and exploitation as usual, while $\pi^E$ ("E" for "exploit") exclusively focuses on exploitation for risk-averse users from the second tier utilizing the data collected so far. An important question is whether such a separation yields advantages over the standard online setting (i.e., $\pi^E = \pi^O$) for the risk-averse users. We individually consider the gap-independent vs. gap-dependent settings. For the former, we prove that the separation is indeed not beneficial from a minimax perspective. For the latter, we show that if choosing Pessimistic Value Iteration as the exploitation algorithm to produce $\pi^E$, we can achieve a constant regret for risk-averse users independent of the number of episodes $K$, which is in sharp contrast to the $\Omega(\log K)$ regret for any online RL algorithms in the same setting, while the regret of $\pi^O$ (almost) maintains its online regret optimality and does not need to compromise for the success of $\pi^E$.

## 1 Introduction

Reinforcement learning (RL) has been applied to many real-world user-interaction applications to provide users with better services, such as in recommendation systems [Afsar et al., 2021] and medical treatment [Yu et al., 2021, Lipsky and Sharp, 2001]. In those scenarios, the users take the role of the environments and the interaction strategies (e.g. recommendation or medical treatment) correspond to the agents in RL. In the theoretical study of such problems, most of the existing literature adopts the online interaction protocol, where in each episode $k \in [K]$, the learning agent executes a policy $\pi_k$ to interact with users (i.e. environments), receives new data to update the policy, and moves on to the next episode. While this formulation *treats each user equivalently* when optimizing the regret, many scenarios have a special "**Tiered Structure**"[2]: *users can be divided into multiple groups depending on their different preference and tolerance about the risk that results from the necessary exploration to improve the policy*, and such grouping is available to the learner in advance so it would be better to treat them separately. As a concrete example, in medical treatment, after a new treatment plan comes out, some courageous patients or paid volunteers (denoted as $G^O$; "O" for "Online") may prefer it given the potential risks, while some conservative patients (denoted as $G^E$; "E" for "Exploit") may tend to receive mature and well-tested plans, even if the new one is promising to be more effective.

---

*Work done during the internship at Microsoft Research Asia.

[2]We consider the cases with two tiers in this paper.

36th Conference on Neural Information Processing Systems (NeurIPS 2022).

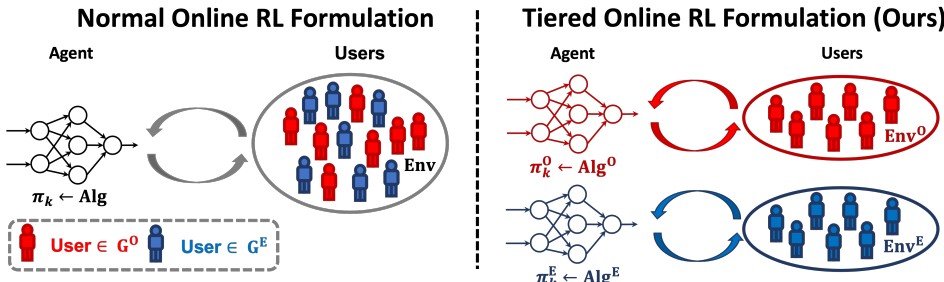

Figure 1: Comparison between the standard setting and our tiered RL setting (#Tiers = 2), where we use red and blue to color users from different groups. The main difference is that, in the standard setting (LHS), the learner does not distinguish users from different groups and treats them equivalently with a single policy $\pi_k$ produced by algorithm Alg, while in our setting (RHS), we leverage the tier information and interact with different groups with different policies $\pi^E$ and $\pi^O$.

As another example, companies offering recommendation services may recruit paid testers or use bonus to attract customers ($G^O$) to interact with the system to shoulder the majority of the exploration risk during policy improvement, which may result in better service (low regret) for the remaining customers ($G^E$). Moreover, many online platforms have free service open for everyone ($G^O$), while some users are willing to pay for enhanced service ($G^E$). If we follow the traditional online setting and treat the users in these two groups equivalently, then in expectation each group will suffer the same regret and risk. In contrast, if we leverage the group information by using policies with different risk levels to interact with different groups, it is potentially possible to transfer some exploration risks from users in $G^E$ to $G^O$, while the additional risks suffered by $G^O$ will be compensated in other forms (such as payment, the users' inherent motivation, or the free service itself).

To make our objective more clear, we abstract the problem setting into Frw. 1 and compare it with the standard online setting in Fig. 1, where we use $\text{Alg}^O$ and $\text{Alg}^E$ to denote the two algorithms producing policies $\pi^E$ and $\pi^O$ to interact with users in $G^O$ and $G^E$, respectively. To enable theoretical analyses, we do adopt a few simplification assumptions while still modelling the core challenges in the aforementioned scenarios: firstly, at each iteration of Frw. 1, the algorithms will interact with and collect one trajectory from each group, which assumes that users from two groups will come to seek for service in pair with the same frequency. In practice, usually, the users come in random order and the frequencies from different groups are not the same; see Appx. B for why our abstraction is still a valid surrogate and how our results can be generalized. Secondly, for convenience, *we only use the samples generated from $G^O$*, because $\text{Alg}^E$ is expected to best exploit the available information and not encouraged to perform intelligent exploration. Nonetheless, our results hold with minor modifications if one also chooses to use trajectories from $G^E$. Thirdly, we assume that the dynamics and rewards during the interactions with users in different groups are all the same (i.e. $\text{Env} = \text{Env}^O = \text{Env}^E$). It is possible that the users in different tiers can behave differently, and we leave the relaxation of such an assumption to future work.

---

**Framework 1:** The Tiered RL Framework

---

**1** Input: $\text{Env}^O$ and $\text{Env}^E$ // Note that $\text{Env}^O = \text{Env}^E$
**2** Initialize $D_1 \leftarrow \{\}$.
**3** **for** $k = 1, 2, ..., K$ **do**
**4**     $\pi_k^O \leftarrow \text{Alg}^O(D_k); \pi_k^E \leftarrow \text{Alg}^E(D_k)$.
**5**     $\pi_k^O$ interacts with customers/users/patients in $G^O$ (i.e. $\text{Env}^O$), and collect data $\tau_k^O$.
**6**     $\pi_k^E$ interacts with customers/users/patients in $G^E$ (i.e. $\text{Env}^E$), and collect data $\tau_k^E$.
**7**     $D_{k+1} = D_k \cup \{\tau_k^O\}$. // We do not consider to use $\tau_k^E$ in this paper.
**8** **end**

---

Similar to the online setting, we use the expected pseudo-regret to measure the performance of the algorithm, which is formalized in Def. 2.1. The key problem we would like to investigate is provable benefits of leveraging the tiered structure by Frw. 1 comparing with the standard online setting:

**Is it possible for Regret(Alg$^{\text{E}}$) to be strictly lower than any online learning algorithms in certain scenarios, while keeping Regret(Alg$^{\text{O}}$) near-optimal?**

Note that we still expect regret of Alg$^{\text{O}}$ to enjoy near-optimal regret guarantees, which is a reasonable requirement as the experience of users in G$^{\text{O}}$ also matters in many of our motivating applications. We regard the above problem formulation as **our first contribution**, which is mainly conceptual.

As **our second contribution**, Sec. 3 shows that Alg$^{\text{E}}$ has the same minimax gap-independent lower bound as online learning algorithms. This result reveals the difficulty to leverage tiered structure in standard tabular MDPs, and motivates us to investigate the benefits under the gap-dependent setting, which is frequently considered in the Multi-Armed Bandit (MAB) [Lattimore and Szepesvári, 2020, Rouyer and Seldin, 2020] and RL literature [Xu et al., 2021, Simchowitz and Jamieson, 2019].

As **our third contribution** and our main technical results, Sec. 4 establishes provable benefits of Frw. 1 by proposing a new algorithmic framework and showing Regret(Alg$^{\text{E}}$) is constant and independent of the number of episodes $K$, which is in sharp contrast with the $\Omega(\log K)$ regret lower bound for any algorithms in the standard online setting that do not leverage the tiered structure. Specifically, we use Pessimistic Value Iteration (PVI) as Alg$^{\text{E}}$ for exploitation to interact with G$^{\text{E}}$, while Alg$^{\text{O}}$ can be arbitrary online algorithms with near-optimal regret. Concretely, we first study stochastic MABs as a warm-up, where we choose Alg$^{\text{E}}$ to be LCB (Lower Confidence Bonus), a degenerated version of PVI in bandits, and choose UCB (Upper Confidence Bonus) as Alg$^{\text{O}}$ for a concrete case study. We prove that Alg$^{\text{E}}$ can achieve constant pseudo-regret $\widetilde{O}\big(\sum_{\Delta_i > 0}(A - i)\big(\frac{1}{\Delta_i} - \frac{\Delta_i}{\Delta_{i-1}^2}\big)\big)$ with $A$ being the number of actions and $\Delta_i$'s being the gaps with $\Delta_1 \geq ... \geq \Delta_{A-1} \geq \Delta_A = 0$, while Alg$^{\text{O}}$ is near-optimal due to the regret guarantee of UCB. After that, Sec. 4.2 extends the success of PVI to tabular MDPs, and establishes results that apply to a wide range of online algorithms Alg$^{\text{O}}$ with near-optimal regret. Although the benefits of pessimism have been widely recognized in offline RL [Jin et al., 2021], to our knowledge, we are the first to study PVI in a gap-dependent online setting. We also contribute several novel techniques for overcoming the difficulties in achieving constant regret, and defer their summary to Sec. 4.2. Moreover, in Appx.H, we report experiment results to demostrate the advantage of leveraging tiered structure as predicted by theory.

**Closest Related Work** Due to space limit, we only discuss the closest related work here and defer the rest to Appx. A. To our knowledge, there is no previous works on leveraging tiered structure in MDPs. In the bandit setting, there is a line of related works studying decoupling exploration and exploitation [Avner et al., 2012, Rouyer and Seldin, 2020], where [Rouyer and Seldin, 2020] studied "best of both worlds" methods and reported a similar constant regret. First, in stochastic bandits, there are many cases when our result is tighter than theirs, (see a detailed comparison in Sec. 4.1), and more importantly, our methods can naturally extend to RL (i.e., MDPs), whereas a similar extension of their techniques can run into serious difficulties: they relied on *importance sampling* to provide unbiased off-policy estimation for policy value, which incurs the infamous "curse of horizon", a.k.a., a sample complexity *exponential* in the planning horizon $H$ in long-horizon RL (see examples in Sec. 2 in [Liu et al., 2018]). Our approach overcomes this difficulty by developing a pessimism-based learning framework, which is fundamentally different from their approach and requires several novel techniques in the analyses. Second, they did not provide any guarantee for the regret of exploration algorithm, whereas in our results the regret of Alg$^{\text{O}}$ can be near-optimal, which we believe is also important as the experience of users in G$^{\text{O}}$ also matters in many of our motivating applications. Third, their bandit results require a unique best arm, whereas we allow the optimal arms/policies to be non-unique, which can cause non-trivial difficulties in the analyses as we will discuss in Sec. 4.2.3.

## 2 Preliminary and Problem formulation

**Stochastic Multi-Armed Bandits (MABs)** The MAB model consists of a set of arms $\mathcal{A} = \{1, 2, ..., A\}$. When sampling an arm $i \in \mathcal{A}$, the agent observes a random variable $r_i \in [0, 1]$. We use $\mu_i = \mathbb{E}[r_i]$ to denote the mean value for arm $i$ for each $i \in \mathcal{A}$. We allow the optimal arms to be non-unique. For simplicity of notation, we assume the arms are ordered such that $\mu_1 \leq \mu_2... \leq \mu_A$.

**Finite-Horizon Tabular Markov Decision Processes (MDPs)** For the reinforcement learning (RL) setting, we consider the episodic tabular MDPs denoted by $M(\mathcal{S}, \mathcal{A}, H, P, r)$, where $\mathcal{S}$ is the finite

state space, $\mathcal{A}$ is the finite action space, $H$ is the horizon length, and $\mathbb{P} = \{\mathbb{P}_h\}_{h=1}^H$ and $r = \{r_h\}_{h=1}^H$ are the time-dependent transition and reward functions, respectively. We assume all steps share the state and action space (i.e. $\mathcal{S}_1 = \mathcal{S}_2... = \mathcal{S}_H = \mathcal{S}$, $\mathcal{A}_1 = \mathcal{A}_2... = \mathcal{A}_H = \mathcal{A}$) while the transition and reward functions can be different. At the beginning of each episode, the environment will start from a fixed initial state $s_1$ (w.l.o.g.). Then, for each time step $h \in [H]$, the agent selects an action $a_h \in \mathcal{A}$ based on the current state $s_h$, receives the reward $r_h(s_h, a_h)$, and observes the system transition to the next state $s_{h+1}$, until $s_{H+1}$ is reached. W.l.o.g., we assume the reward function $r$ is deterministic and our results can be easily extended to handle stochastic rewards.

A time-dependent policy is specified as $\pi = \{\pi_1, \pi_2, ..., \pi_H\}$ with $\pi_h : \mathcal{S} \to \Delta(\mathcal{A})$ for all $h \in [H]$. Here $\Delta(\mathcal{A})$ denotes the probability simplex over the action space. With a slight abuse of notation, when $\pi_h$ is a deterministic policy, we use $\pi_h : \mathcal{S} \to \mathcal{A}$ to refer to a deterministic mapping. $V_h^\pi(s)$ and $Q_h^\pi(s,a)$ denote the value function and Q-function at step $h \in [H]$, which are defined as: $V_h^\pi(s) = \mathbb{E}[\sum_{h'=h}^H r_{h'}(s_{h'}, a_{h'})|s_h = s, \pi]$, $Q_h^\pi(s,a) = \mathbb{E}[\sum_{h'=h}^H r_{h'}(s_{h'}, a_{h'})|s_h = s, a_h = a, \pi]$.

We use $V_h^*(\cdot) := \max_\pi V_h^\pi(\cdot)$ and $Q_h^*(\cdot, \cdot) = \max_\pi Q_h^\pi(\cdot, \cdot)$ to refer to the optimal state/action-value functions, and $\Pi^*(s_h) := \{a_h|Q^*(s_h, a_h) = V_h^*(s_h)\}$ to denote the collection of all optimal actions at state $s_h$. With an abuse of notation, we define $\Pi^* := \{\pi : V_1^\pi(s_1) = V_1^*(s_1)\}$, i.e., the set of policies that maximize the total expected return. In this paper, when we say that the MDP has "unique optimal (deterministic) policy", it is up to the occupancy measure, that is, all policies in $\Pi^*$ share the same state-action occupancy $d^\pi(s_h, a_h) := \Pr(S_h = s_h, A_h = a_h|S_1 = s_1, \pi)$ for all $h \in [H], s_h \in \mathcal{S}_h, a_h \in \mathcal{A}_h$. In the following, we use $|\Pi^*| = 1$ to refer to the case of unique optimal (deterministic) policy, where the cardinality of $\Pi^*$ is counted up to the equivalence of occupancies. Besides, for any function $V : \mathcal{S} \to \mathbb{R}$, we denote $\mathbb{P}_h V(s_h, a_h) := \mathbb{E}_{s_{h+1} \sim \mathbb{P}_h(\cdot|s_h, a_h)}[V(s_{h+1})]$.

**Gap-Dependent Setting** We follow the standard formulation of gap-dependent setting in previous bandits [Lattimore and Szepesvári, 2020] and RL literature [Simchowitz and Jamieson, 2019, Xu et al., 2021, Dann et al., 2021]. In bandits, the gap w.r.t. arm $i$ is defined as $\Delta_i := \max_{j \in [A]} \mu_j - \mu_i, \forall i \in [A]$, and we assume that there exists a strictly positive value $\Delta_{\min}$ such that, either $\Delta_i = 0$ or $\Delta_i \geq \Delta_{\min}$. For tabular RL setting, we define $\Delta_h(s_h, a_h) := V^*(s_h) - Q^*(s_h, a_h), \forall h \in [H], s_h \in \mathcal{S}_h, a_h \in \mathcal{A}_h$. We use the same notation $\Delta_{\min}$ to refer to the minimal gap in tabular setting and assume that either $\Delta_h(s_h, a_h) = 0$ or $\Delta_h(s_h, a_h) \geq \Delta_{\min}$.

**Performance Measure** We use Pseudo-Regret defined below to measure the performance of $\mathrm{Alg}^O$ and $\mathrm{Alg}^E$. In the following, we will also use "exploitation regret" to refer to $\mathrm{Regret}_K(\mathrm{Alg}^E)$.

**Definition 2.1** (Pseudo-Regret). *We define the regret of $\mathrm{Alg}^O$ and $\mathrm{Alg}^E$ to be:*

$$\mathrm{Regret}_K(\mathrm{Alg}^O) := \mathbb{E}\left[\sum_{k=1}^K V_1^*(s_1) - V_1^{\pi_k^O}(s_1)\right]; \quad \mathrm{Regret}_K(\mathrm{Alg}^E) := \mathbb{E}\left[\sum_{k=1}^K V_1^*(s_1) - V_1^{\pi_k^E}(s_1)\right],$$

*where $\pi_k^O$ and $\pi_k^E$ are generated according to the procedure in Framework 1 and the expectation is taken over the randomness in data generation and algorithms.*

## 3 Lower Bound of $\mathrm{Regret}(\mathrm{Alg}^E)$ without Gap Assumption

In this section, we show that, in normal tabular RL setting, for arbitrary algorithm pair $(\mathrm{Alg}^O, \mathrm{Alg}^E)$, even if we do not constrain $\mathrm{Alg}^O$ to be near-optimal, the regret of $\mathrm{Alg}^E$ has the same minimax lower bound as algorithms in online setting. We defer the formal statement and proof to Appendix C.1.

**Theorem 3.1.** *[Lower Bound for $\mathrm{Alg}^E$ without Gap Assumption] There exist positive constants $c, \varepsilon_0, \delta_0$, such that, for arbitrary $S \geq 4, A \geq 2, H \geq 2, K \geq \frac{c}{\varepsilon_0^2} H^3 SA$, and arbitrary algorithm pair $(\mathrm{Alg}^O, \mathrm{Alg}^E)$, there must exist a hard tabular MDP $M_{hard}$, $\mathbb{E}_{(\mathrm{Alg}^O, \mathrm{Alg}^E), M_{hard}}\left[\sum_{k=1}^K V^* - V^{\pi_k^E}\right] \geq \delta_0 \sqrt{cH^3 SAK}$, where the expectation is taken over the randomness of algorithms and MDP.*

The theorem above is stating that, comparing with the regret lower bound for online algorithms $\widetilde{O}(\sqrt{H^3 SAK})$ in Theorem 9 of Domingues et al. [2021], the exploitation algorithm cannot reduce the dependence on any of parameters $H, S, A, K$ in hard MDPs, even if we allow $\mathrm{Alg}^O$ to sacrifice its performance to gather the best possible data for $\mathrm{Alg}^E$. Also, the lower bound would still hold

even if we allow both $\text{Alg}^O$ and $\text{Alg}^E$ to additionally use the data $\tau^E$ generated by $\pi^E$. This negative result implies that without any further assumptions, the separation is not beneficial from a minimax optimality perspective, and we can simply choose both $\text{Alg}^E$ and $\text{Alg}^O$ to be the same near-optimal online algorithm as without worrying about separating them.

However, in the next section, we will show that, in tabular MDPs with strictly positive gaps, in contrast with the $\Omega(\log K)$ lower bound for online algorithms, we can have $\text{Alg}^E$ such that its regret is constant and independent on the number of time horizon $K$, which reveals the fundamental differences between the pure online setting and the Tiered RL setting considered in this paper.

## 4  Pessimism in the Face of Uncertainty and Constant Regret

In this section, we consider the gap-dependent setting and contribute to identifying the possibility to achieve constant regret by using pessimistic algorithms for $\text{Alg}^E$. Intuitively, the main reason why PVI can lead to a constant regret is that the quality of the policy returned by PVI is positively correlated to the accumulation of optimal trajectories in the dataset $D$, which is directly connected with $\text{Regret}(\text{Alg}^O)$. As a result, on the one hand, the regret minimization objective of $\text{Alg}^E$ coincidentally aligns with the optimality constraint of $\text{Alg}^O$. On the other hand, thanks to the positive gap assumption, $\pi^E$ will gradually converge to the optimal policy with high probability when $\text{Alg}^E$ is PVI, so there will be no regret after that. In Sec. 4.1, we start with stochastic MAB as a warm-up, and in Sec. 4.2 we extend our success to tabular RL setting. We defer the proofs in this section to Appendix D.

### 4.1  Warm-Up: Gap-Dependent Regret Bound for Stochastic Multi-Armed Bandits

---
**Algorithm 2:** UCB-Exploration-LCB-Exploitation
---
1  **Initialize**: $\alpha > 1$;    $N_i(1) \leftarrow 0$, $\widehat{\mu}_i(1) \leftarrow 0$, $\forall i \in \mathcal{A}$;    $f(k) := 1 + 16A^2(k+1)^2$
2  **for** $k = 1, 2, ..., K$ **do**
3  $\quad$ $\pi_k^O \leftarrow \arg\max_i \widehat{\mu}_i(k) + \sqrt{\frac{2\alpha \log f(k)}{N_i(k)}}$, $\qquad$ $\pi_k^E \leftarrow \arg\max_i \widehat{\mu}_i(k) - \sqrt{\frac{2\alpha \log f(k)}{N_i(k)}}$.
4  $\quad$ Interact with $G^E$ and $G^O$ by $\pi_k^E$ and $\pi_k^O$, and observe reward $r(\pi_k^E)$ and $r(\pi_k^O)$, respectively.
5  $\quad$ **for** $i = 1, 2, ..., A$ **do**
6  $\quad\quad$ $N_i(k+1) \leftarrow N_i(k) + \mathbb{I}[\pi_k^O = i]$; $\qquad$ $\widehat{\mu}_i(k+1) \leftarrow \widehat{\mu}_i(k)\frac{N_i(k)}{N_i(k+1)} + r(\pi_k^O)\frac{\mathbb{I}[\pi_k^O=i]}{N_i(k+1)}$.
7  $\quad$ **end**
8  **end**
---

Our main algorithm for bandit setting is shown in Alg 1, where we consider the UCB algorithm [Lattimore and Szepesvári, 2020] as $\text{Alg}^O$ and choose the LCB as $\text{Alg}^E$, which flips the sign of the bonus term in UCB. We use $N_i(k)$ to denote the number of times that arm $i$ was pulled previous to step $k$, and use $\widehat{\mu}_i$ to record the empirical average of arm $i$. Besides, we assume $1/N_i(\cdot) = +\infty$ if $N_i(\cdot) = 0$, which implies that at the first $|\mathcal{A}|$ steps the algorithm will pull each arm one by one. Moreover, as we will show later, the choice of $\alpha > 1$ is crucial to avoiding dependence on $K$ in $\text{Regret}(\text{Alg}^E)$ with our techniques. For Alg. 2, we have the following guarantee:

**Theorem 4.1.** *[Exploitation Regret] In Algorithm 2, by choosing arbitrary $\alpha > 1$, there exists an absolute constant $c$, such that, for arbitrary $K \geq 1$, the pseudo-regret of $\text{Alg}^E$ is upper bounded by:*
$\text{Regret}_K(\text{Alg}^E) \leq \widetilde{O}\left(\frac{A}{\alpha-1} + \alpha \sum_{\Delta_i > 0}(A-i)\left(\frac{1}{\Delta_i} - \frac{\Delta_i}{\Delta_{i-1}^2}\right)\right)$ *where* $\Delta_0 := \infty$ *so* $\frac{\Delta_1}{\Delta_0^2} = 0$.

Our result implies that by choosing PVI as $\text{Alg}^E$, we can achieve constant regret while keeping $\text{Alg}^O$ near-optimal. Besides the advantages discussed in the related work paragraph in Sec. 1, our guarantee is also more favorable in certain cases compared to the $O(\sqrt{\frac{A}{\Delta_{\min}}}\sqrt{\sum_{\Delta_i > 0}\frac{1}{\Delta_i}})$ result in Rouyer and Seldin [2020]: while it is not easy to verify whether our guarantee dominates theirs, in many cases ours can be strictly better (or at least no worse) than theirs. For example, consider the following two representative cases: $\Delta_1 = \Delta_2 = ...\Delta_{A-1} = \Delta_{\min}$ (uniform gap) and $\Delta_1 = \Delta_2 = ... = \Delta_{A-2} \gg \Delta_{A-1} = \Delta_{\min}$ (small last gap); our result achieves $\widetilde{O}(\frac{A}{\Delta_{\min}})$ and $\widetilde{O}(\frac{1}{\Delta_{\min}})$, respectively, in contrast to their $\widetilde{O}(\frac{A}{\Delta_{\min}})$ and $\widetilde{O}(\frac{\sqrt{A}}{\Delta_{\min}})$.

**Proof Sketch**: The proof consists of two novel technique lemmas with a carefully chosen failure rate $\delta_k \sim O(1/k^{\Theta(\alpha)})$ so that the accumulative failure probability $\sum_{k=1}^{\infty} \delta_k < +\infty$. The first one is Lem. 4.2, where we show that w.p. $1 - \delta_k$, LCB will not prefer $i$ with $\Delta_i > 0$ as long as another better arm has been visited enough times in the dataset. The second step is to identify a key property of UCB algorithm as stated in Lem. 4.3, where we provide a high probability upper bound that $N_i(k) \leq k/\lambda$ if $k \geq \widetilde{\Theta}(\lambda/\Delta_i^2)$ for arbitrary $\lambda \in [1, 4A]$, and it serves to indicate that the condition required by the success of LCB is achievable as long as $k$ is large enough [3].

**Lemma 4.2.** *[Blessing of Pessimism] With the choice that $f(k) = 1 + 16A^2(k+1)^2$, for arbitrary $i$ with $\Delta_i > 0$, for the LCB algorithm in Alg 2, and arbitrary $j$ satisfying $\Delta_j < \Delta_i$, we have:*
$\Pr\left( \{i = \pi_k^E\} \cap \{\Delta_j < \Delta_i\} \cap \left\{ N_j(k) \geq \frac{8\alpha \log f(k)}{(\Delta_j - \Delta_i)^2} \right\} \right) \leq \frac{2}{k^{2\alpha}}.$

**Lemma 4.3.** *[Property of UCB] With the choice that $f(k) = 1 + 16A^2(k+1)^2$, there exists a constant $c$, for arbitrary $i$ with $\Delta_i > 0$ and arbitrary $\lambda \in [1, 4A]$, in UCB algorithm, we have:*
$\Pr(N_i(k) \geq \frac{k}{\lambda}) \leq \frac{2}{k^{2\alpha-1}}, \quad \forall k \geq \lambda + c \cdot \frac{\alpha\lambda}{\Delta_i^2} \log(1 + \frac{\alpha A}{\Delta_{\min}}).$

Directly combining the above two results, we can obtain an upper bound for Regret(Alg$^E$) of order $\widetilde{O}(A/\Delta_{\min}^{-2})$, which is already independent of $K$. To achieve better dependence on $\Delta_{\min}$ in the regret, we conduct a finer analysis. For each arm $i$ with $\Delta_i > 0$, we separate all the arms including $i$ into two groups based on whether its gap exceeds $\Delta_i/2$: $G_i^{\text{lower}} = \{j : \Delta_j > \Delta_i/2\}$ and $G_i^{\text{upper}} = \{j : \Delta_j \leq \Delta_i/2\}$. As a result of Lem. 4.2, we know that $\pi_k^E$ will not prefer arm $i$ as long as there exists $j \in G_i^{\text{upper}}$ such that $N_j(k) = \widetilde{\Omega}(4\Delta_i^{-2}) = \widetilde{\Omega}(\Delta_i^{-2})$. Based on Lem. 4.3, we know it is true with high probability, as long as $k \geq \widetilde{\Theta}(A \cdot \Delta_i^{-2})$, since at that time $N_l(k) \leq k/A$ holds for arbitrary $l \in G_i^{\text{lower}}$, which directly implies that $\max_{j:j \in G_i^{\text{upper}}} N_j(k) \geq \widetilde{\Omega}(\Delta_i^{-2})$. Then, combining Lem. 4.2, with high probability, the regret resulting from taken arm $i$ cannot be higher than $\widetilde{\Theta}(A \cdot \Delta_i^{-2}) \cdot \Delta_i = \widetilde{\Theta}(A \cdot \Delta_i^{-1})$, which results in a $\widetilde{O}(\sum_{\Delta_i > 0} A/\Delta_i)$ regret bound. As for the techniques leading to the further improvement in our final result, please refer to Lem. D.1 and the proof of Thm. 4.1 in Appx. D.

## 4.2 Constant Regret of Alg$^E$ in Tabular MDPs

In this section, we establish constant regret of Alg$^E$ based on realistic conditions for Alg$^O$ and Alg$^E$. We highlight the key steps of our analysis and our technical contributions here.

First of all, in Sec.4.2.1, we propose the concrete PVI algorithm, and inspired by the clipping trick used for optimistic online algorithms [Simchowitz and Jamieson, 2019], we develop a high-probability gap-dependent upper bound for the sub-optimality of $\pi^E$, which is related to the accumulation of the optimal trajectories in dataset $D_k$. Secondly, in Sec. 4.2.2, we first introduce a general condition (Cond. 4.6) for the chocie of Alg$^O$, based on which we quantify the accumulation of optimal trajectories in $D_k$ with the regret of Alg$^O$, and connect the exploration by Alg$^O$ and the optimality of Alg$^E$. We also supplement some details about how to relax such a condition and inherit the guarantees by the doubling-trick in Appx. G, which may be of independent interest. In Sec. 4.2.3, i.e. the last part of analysis, we bring the above two steps together and complete the proof. However, there is an additional challenge when the tabular MDP has multiple deterministic optimal policies, which is possible when there are non-unique optimal actions at some states. We overcome this difficulty by Thm. 4.8 about policy coverage. To our knowledge, the only paper that runs into a similar challenge is [Papini et al., 2021], and they bypass the difficulty by assuming the uniqueness of optimal policy. Finally, Section 4.2.4 provide some interpretation and implications of our results.

### 4.2.1 Pessimistic Value Iteration as Alg$^E$ and its Property

The full details of our algorithm for tiered RL setting is provided in Alg. 3, where we use PVI as Alg$^E$. Here we do not specify a concrete **Bonus** function, but provide general results for a range of

---

[3]Comparing with results in Thm. 8.1 of [Lattimore and Szepesvári, 2020], although our upper bounds of $N_i(k)$ is linear w.r.t. $k$ rather than log scale, we want to highlight that ours hold with high probability $O(1 - k^{-\Theta(\alpha)})$ while [Lattimore and Szepesvári, 2020] only upper bounded the expectation.

**Algorithm 3:** Tiered-RL Algorithm with Pessimistic Value Iteration as Alg$^{\text{E}}$

---

1  **Input**: Episode number $K$; Confidence level $\{\delta_k\}_{k=1}^K$; Bonus function **Bonus**$(\cdot,\cdot)$

2  **for** $k = 1, 2, ..., K$ **do**

3     $\{b_{k,1}(\cdot,\cdot), b_{k,2}(\cdot,\cdot), ..., b_{k,H}(\cdot,\cdot)\} \leftarrow$ **Bonus**$(D_k, \delta_k)$. //Compute bonus function for PVI.

4     **for** $h = H, H-1, ..., 1$ **do**

5         **for** $s_h \in \mathcal{S}_h, a_h \in \mathcal{A}_h$ **do**

6             $N_{k,h}(s_h, a_h) \leftarrow$ the number of times $s_h, a_h$ occurs in the dataset $D_k$.

7             $N_{k,h}(s_h, a_h, s_{h+1}) \leftarrow$ the number of times $(s_h, a_h, s_{h+1})$ occurs in the dataset $D_k$.

8             $\widehat{\mathbb{P}}_{k,h}(\cdot|s_h, a_h) \leftarrow \begin{cases} 0, & \text{if } N_{k,h}(s_h, a_h) = 0; \\ \frac{N_{k,h}(s_h, a_h, \cdot)}{N_{k,h}(s_h, a_h)}, & \text{otherwise.} \end{cases}$

9         **end**

10       $\widehat{Q}_{k,h}(\cdot, \cdot) \leftarrow \max\{R(\cdot, \cdot) + \widehat{\mathbb{P}}_{k,h}\widehat{V}_{k,h+1}(\cdot, \cdot) - b_{k,h}(\cdot, \cdot), 0\}$.

11       $\widehat{V}_{k,h}(\cdot) = \max_{a_h \in \mathcal{A}} \widehat{Q}_{k,h}(\cdot, a_h), \quad \pi_{k,h}^{\text{PVI}}(\cdot) \leftarrow \arg\max_a \widehat{Q}_{k,h}(\cdot, a)$.

12     **end**

13     $\pi_k^{\text{E}} \leftarrow \{\pi_{k,1}^{\text{PVI}}, \pi_{k,2}^{\text{PVI}}, ...\pi_{k,H}^{\text{PVI}}\}$

14     // **Step 2**: Use Alg$^{\text{O}}$ satisfying Cond. 4.6 to compute $\pi_k^{\text{O}}$ for G$^{\text{O}}$

15     $\pi_k^{\text{O}} \leftarrow$ Alg$^{\text{O}}(D_k)$.

16     // **Step 3**: Sample trajectories and collect new data

17     Interact with G$^{\text{E}}$ and G$^{\text{O}}$ by $\pi_k^{\text{E}}$ and $\pi_k^{\text{O}}$, and observe $\tau_k^{\text{E}}$ and $\tau_k^{\text{O}}$, respectively.

18     $D_{k+1} \leftarrow D_k \cup \{\tau_k^{\text{O}}\}$.

19 **end**

---

qualified bonus functions satisfying Cond. 4.4 below. Cond. 4.4 can be satisfied by many bonus term considered in online literatures, and we briefly dicuss some examples in Appx. F.1.

**Condition 4.4** (Condition on Bonus Term for Alg$^{\text{E}}$). *We define the following event at iteration $k \in [K]$ during the running of Alg. 3:* $\mathcal{E}_{\textbf{\textit{Bonus}},k} := \bigcap_{h\in[H], s_h\in\mathcal{S}_h, a_h\in\mathcal{A}_h} \Big\{ \{|\widehat{\mathbb{P}}_{k,h}\widehat{V}_{k,h+1}(s_h, a_h) - \mathbb{P}_h\widehat{V}_{k,h+1}(s_h, a_h)| < b_{k,h}(s_h, a_h)\} \cap \{b_{k,h}(s_h, a_h) \leq B_1\sqrt{\frac{\log(B_2/\delta_k)}{N_{k,h}(s_h, a_h)}}\} \Big\}$ *where $B_1$ and $B_2$ are parameters depending on $S, A, H$ and $\Delta$ but independent of $\delta_k$, $k$.[4] We assume that, **Bonus** function satisfies that, in Alg. 3, given arbitrary sequence $\{\delta_k\}_{k=1}^K$ with $\delta_1, \delta_2, ..., \delta_K \in (0, 1/2)$, at arbitrary iteration $k \in [K]$, we have $\Pr(\mathcal{E}_{\textbf{\textit{Bonus}},k}) \geq 1 - \delta_k$.*

Next, we provide an upper bound for the sub-optimality gap of $\pi_k^{\text{PVI}}$ with the clipping operator $\text{Clip}[x|\varepsilon] := x \cdot \mathbb{I}[x \geq \varepsilon]$. Previous upper bounds of PVI [e.g., Theorem 4.4 of Jin et al., 2021] do not leverage the strictly positive gap and can be much looser when $N_{k,h}$ is large, and directly applying those results to our analysis would result in a regret scaling with $\sqrt{K}$.

**Theorem 4.5.** *By running Algorithm 3 with confidence level $\delta_k$, a function **Bonus** satisfying Condition 4.4, and a dataset $D = \{\tau_1, ...\tau_k\}$ consisting of $k$ complete trajectories generated by executing a sequence of policies $\pi_1, ..., \pi_k$, on the event $\mathcal{E}_{\textbf{\textit{Bonus}}}$ defined in Condition 4.4:*

$$V_1^*(s_1) - V_1^{\pi_k^{PVI}}(s_1) \leq 2\mathbb{E}_{\pi^*}\left[\sum_{h=1}^H \text{Clip}\left[\min\left\{H, 2B_1\sqrt{\frac{\log(B_2/\delta_k)}{N_{k,h}(s_h, a_h)}}\right\}\middle|\varepsilon_{\text{Clip}}\right]\right]. \quad (1)$$

*where $\pi^*$ can be an arbitrary optimal policy, $\varepsilon_{\text{Clip}} := \frac{\Delta_{\min}}{2H+2}$ if $|\Pi^*| = 1$ and $\varepsilon_{\text{Clip}} := \frac{d_{\min}\Delta_{\min}}{2SAH}$ if $|\Pi^*| > 1$, where $d_{\min} := \min_{\pi \in \Pi^*, h\in[H], s_h\in\mathcal{S}_h, a_h\in\mathcal{A}_h} d^\pi(s_h, a_h)$ subject to $d^\pi(s_h, a_h) > 0$.*

### 4.2.2   Choice and Analysis of Alg$^{\text{O}}$

Next, we introduce our general condition for Alg$^{\text{O}}$ that the Alg$^{\text{O}}$ can achieve $O(\log k)$-regret with high probability. It is worth noting that many existing near-optimal online RL algorithms for gap-dependent settings may not directly satisfy the condition [Simchowitz and Jamieson, 2019, Xu et al.,

---

[4]Note that we do not require the knowledge of $\Delta_i$'s to compute $b_{k,h}$.

2021, Dann et al., 2021] since they use a fixed confidence interval $\delta$. In Appx. G, we will introduce a more realistic abstraction of those algorithms in Cond. G.1, and discuss in more details about how to close this gap with an algorithm framework inspired by the doubling trick.

**Condition 4.6** (Condition on $\text{Alg}^O$). *$\text{Alg}^O$ is an algorithm which returns deterministic policies at each iteration, and for arbitrary $k \geq 2$, we have:* $\Pr\left(\sum_{\widetilde{k}=1}^{k} V_1^*(s_1) - V_1^{\pi_{\widetilde{k}}^O}(s_1) > C_1 + \alpha C_2 \log k\right) \leq \frac{1}{k^\alpha}$, *where $C_1, C_2$ are parameters only depending on $S, A, H$ and gap $\Delta$ and independent of $k$.*

**Implication of Condition 4.6 for $\text{Alg}^O$** Intuitively, low regret implies high accumulation of optimal trajectories in the dataset collected by $\text{Alg}^O$. We formalize this intuition in Thm. 4.7 by establishing the relationship between the regret of $\text{Alg}^O$, $d^{\pi^*}$ and $\sum_{\widetilde{k}=1}^{k} d^{\pi_{\widetilde{k}}^O}(s_h, a_h)$ (the expectation of $N_{k,h}$).

**Theorem 4.7.** *For an arbitrary sequence of deterministic policies $\pi_1, \pi_2, ..., \pi_k$, there must exist a sequence of deterministic optimal policies $\pi_1^*, \pi_2^*, ..., \pi_k^*$, such that $\forall h \in [H], s_h \in \mathcal{S}_h, a_h \in \mathcal{A}_h$:*

$$\sum_{\widetilde{k}=1}^{k} d^{\pi_{\widetilde{k}}}(s_h, a_h) \geq \sum_{\widetilde{k}=1}^{k} d^{\pi_{\widetilde{k}}^*}(s_h, a_h) - \frac{1}{\Delta_{\min}}\left(\sum_{\widetilde{k}=1}^{k} V_1^*(s_1) - V_1^{\pi_{\widetilde{k}}}(s_1)\right).$$

### 4.2.3  Main Results and Analysis

The main analysis is based on our discussion about the properties of $\text{Alg}^E$ and $\text{Alg}^O$ in previous sub-sections. In the following, we first discuss the proof sketch for the case when $|\Pi^*| = 1$. The main idea is to show that the unique $\pi^*$ will be "well-covered" by dataset, where we say a policy $\pi^*$ is "well-covered" if for each $(s_h, a_h) \in \mathcal{S}_h \times \mathcal{A}_h$ with $d^{\pi^*}(s_h, a_h) > 0$, $N_{k,h}(s_h, a_h)$ can strictly increase so that the RHS of Eq.(1) in Thm. 4.5 will gradually decay to zero (e.g. $N_{k,h}(s_h, a_h) \geq \widetilde{O}(k)$). To show this, the key observation is that, with high probability, $N_{k,h}(s_h, a_h)$ will not deviate too much from its expectation $\sum_{\widetilde{k}} d^{\pi_{\widetilde{k}}}(s_h, a_h)$ (Lem. F.8), and can be lower bounded by $\sum_{\widetilde{k}=1}^{k} d^{\pi_{\widetilde{k}}^*}(s_h, a_h) - O(\log k) = k d^{\pi^*}(s_h, a_h) - O(\log k)$ as a result of Thm. 4.7. As a result, the clipping operator in Eq.(1) will take effects as long as $k$ is large enough, and $\pi_k^{\text{PVI}}$ will converge to the optimal policy with no regret. All that remains is to show the regret under failure events is also at the constant level because we choose a gradually decreasing failure rate $O(\frac{1}{k^\alpha})$, and $\lim_{K \to \infty} \sum_{k=1}^{K} O(\frac{1}{k^\alpha}) < \infty$ as long as $\alpha > 1$.

However, when $|\Pi^*| > 1$, the analysis becomes more challenging. The main difficulty is that, when the optimal policy is not unique, it is not obvious about the existence of "well-covered" $\pi^*$, since it is not guarantee that how much similarity is shared by the sequence of policies $\pi_1^*, ... \pi_k^*$, especially when $|\Pi^*|$ is exponentially large (e.g. $|\Pi^*| = \Omega((SA)^H)$). We overcome this difficulty by proving the existence of "well-covered" policy in the theorem stated below:

**Theorem 4.8.** *[The existance of well-covered optimal policy] Given an arbitrary tabular MDP, and an arbitrary sequence of deterministic optimal policies $\pi_1^*, \pi_2^*, ... \pi_k^*$ ($\pi_i^*$ may not equal to $\pi_j^*$ for arbitrary $1 \leq i < j \leq k$ when there are multiple deterministic optimal policies), there exists a (possibly stochastic) policy $\pi_{cover}^*$ such that $\forall h \in [H], \forall (s_h, a_h) \in \mathcal{S}_h \times \mathcal{A}_h$ with $d^{\pi_{cover}^*}(s_h, a_h) > 0$:*

$$\sum_{\widetilde{k}=1}^{k} d^{\pi_{\widetilde{k}}^*}(s_h, a_h) \geq \frac{k}{2} \cdot \widetilde{d}^{\pi_{cover}^*}(s_h, a_h), \text{ with } \widetilde{d}^{\pi_{cover}^*}(\cdot, \cdot) := \max\left\{\frac{d_{h,\min}^*(\cdot, \cdot)}{(|\mathcal{Z}_{h,div}^*| + 1)H}, d^{\pi_{cover}^*}(\cdot, \cdot)\right\}.$$

*where $\mathcal{Z}_{h,div}^* := \{(s_h, a_h) \in \mathcal{S}_h \times \mathcal{A}_h | \exists \pi^*, \widetilde{\pi}^* \in \Pi^*, \text{ s.t. } d^{\pi^*}(s_h) > 0, d^{\widetilde{\pi}^*}(s_h) = 0\}$, and $d_{h,\min}^*(s_h, a_h) := \min_{\pi^* \in \Pi^*} d^{\pi^*}(s_h, a_h)$ subject to $d^{\pi^*}(s_h, a_h) > 0$.*

Here we provide some explanation to the above result. According to the definition, $\mathcal{Z}_{h,div}^*$ is the set including the state-action pairs which can be covered by some deterministic policies but is not reachable by some other deterministic policies, and therefore $|\mathcal{Z}_{h,div}^*| \leq SA$ (or even $|\mathcal{Z}_{h,div}^*| \ll SA$). Besides, $d_{h,\min}^*(s_h, a_h)$ denotes the minimal occupancy over all possible deterministic optimal policies which can hit $s_h, a_h$, and therefore, is no less than $d_{\min}$ defined in Thm. 4.5. As a result, we know there exists a "well-covered" $\pi_{cover}^*$, since the accumulative density of its arbitrary reachable states can be lower bounded by $O(k)$. Then, following a similar discussion as the case $|\Pi^*| = 1$, we can finish the proof. We summarize our main result below.

**Theorem 4.9.** *By running an Algorithm satisfying Condition 4.6 as $Alg^O$, running Alg 3 as $Alg^E$ with a bonus term function **Bonus** satisfying Condition 4.4 and $\delta_k = 1/k^\alpha$, for some constant $\alpha > 1$, for arbitrary $K \geq 1$, the exploitation regret of $Alg^E$ can be upper bounded by:*

*(i) When $|\Pi^*| = 1$ (unique optimal deterministic policy):*

$$Regret_K(Alg^E) \leq O\Big(\sum_{h=1}^{H} \sum_{\substack{s_h, a_h: \\ d^{\pi^*}(s_h, a_h) > 0}} \Big(\frac{C_1 + C_2}{\Delta_{\min}} \log \frac{SAH(C_1 + C_2)}{d^{\pi^*}(s_h, a_h)\Delta_{\min}} + \frac{B_1 H}{\Delta_{\min}} \log \frac{B_2 H}{d^{\pi^*}(s_h, a_h)\Delta_{\min}}\Big)\Big).$$

*(ii) When $|\Pi^*| > 1$ (non-unique optimal deterministic policies):*

$$Regret_K(Alg^E) \leq O\Big(\sum_{h=1}^{H} \sum_{\substack{s_h, a_h: \\ d^{\pi^*_{cover}}(s_h, a_h) > 0}} \Big(\frac{C_1 + C_2}{\Delta_{\min}} \log \frac{SAH(C_1 + C_2)}{\widetilde{d}^{\pi^*_{cover}}(s_h, a_h)\Delta_{\min}} + \frac{B_1 SAH}{d_{\min}\Delta_{\min}} \log \frac{B_2 SAH}{d_{\min}\Delta_{\min}}\Big)\Big).$$

*where $\pi^*_{cover}$ and $\widetilde{d}^{\pi^*_{cover}}(s_h, a_h)$ are introduced in Theorem 4.8.*

### 4.2.4 Interpretation of Results in Tabular RL

Recall our objective in Sec. 1 is to establish the benefits of leveraging the tiered structure by showing $Regret_K(Alg^E)$ is constant. This contrasts the lower bound of online algorithms that continuously increases with the episode number $K$, which corresponds to the regret suffered by users in $G^E$ without leveraging the tiered structure, while $Regret_K(Alg^O)$ keeps (near-)optimal as before. In Appx. H, we also provide some simulation results as a verification of our theoretical discovery.

One limitation of our results is that our bounds have additional dependence on $d^{\pi^*}$ (or even $1/d^{\pi^*}$) compared to most of the regret bounds in the online setting, although similar dependence on $\log d^{\pi^*}$ also appeared in a few recent works [e.g., $\lambda_h^+$ in Thms. 8 and 9 of Papini et al., 2021]. Besides, according to the lower and upper bound of online RL in gap-dependent settings [Simchowitz and Jamieson, 2019], $C_1 + C_2$ in Cond. 4.6 have dependence on $O(\Delta_{\min}^{-1})$, which implies that in the regret bound in Thm. 4.9, the dependence on $\Delta_{\min}$ would be $O(\Delta_{\min}^{-2})$. For the former, in Appx. C.2, we prove a lower bound, showing that $\log \frac{1}{d^{\pi^*}}$ is unavoidable when $Alg^O$ is allowed to behave adversarially without violating Cond. 4.6; for the latter, we note that in the analysis of MAB setting (Sec.4.1), specifying the detailed behavior of $Alg^O$ can help tighten the bound. Therefore, we conjecture that our results can be improved by putting more constraints on the behavior of $Alg^O$, which we leave to future work.

## 5 Conclusion

In this paper, we identify the tiered structure in many real-world applications and study the potential advantages of leveraging it by interacting with users from different groups with different strategies. Under the gap-dependent setting, we provide theoretical evidence of benefits by deriving constant regret for the exploitation policy while maintaining the optimality of the online learning policy.

As for the future work, we propose several potentially interesting directions. **(i)** As we mentioned in Section 4.2.4, it is worth investigating the possibility of improving the regret bound of $Alg^E$ by considering a more concrete choice of $Alg^O$, or maybe other choices for $Alg^E$. **(ii)** It would be interesting to relax our constraint on the optimality of $Alg^O$ by introducing the notion of budget $C$ as the tolerance on the sub-optimality of $Alg^O$. As a result, our setting and the decoupling exploration and exploitation setting can be regarded as special cases of a more general framework when $C = 0$ and $C = \infty$. **(iii)** We assume that the users from different groups share the same transition and reward function, and it would also be interesting to extend our results to more general settings, where the group ID serves as context and will affect the dynamics [Abbasi-Yadkori and Neu, 2014, Modi et al., 2018]. **(iv)** We only consider the setting with two tiers, and it may be worth studying the possibility and potential benefits under the setting with multiple tiers.

## Acknowledgements

JH's research activities on this work were conducted during his internship at MSRA. NJ's last involvement was in December 2021. NJ also acknowledges funding support from ARL Cooperative Agreement W911NF-17-2-0196, NSF IIS-2112471, NSF CAREER award, and Adobe Data Science Research Award. The authors thank Yuanying Cai for valuable discussion.

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
