# A   Detailed Related Work

**Online RL**   The online RL/MAB is the most basic framework studying the trade-off between exploration and exploitation [Auer et al., 2002, Slivkins, 2019, Lattimore and Szepesvári, 2020], where the agent targets at exploring the MDP to identify good actions as fast as possible to minimize the accumulative regrets. In tabular MDPs [Jaksch et al., 2010, Dann et al., 2017, Jin et al., 2018], the regret lower bounds for non-stationary MDPs [Domingues et al., 2021] have been achieved by Azar et al. [2017], Zanette and Brunskill [2019]. Recently, there has been interests in studying the gap-dependent regret [Simchowitz and Jamieson, 2019, Xu et al., 2021, Dann et al., 2021, Yang et al., 2021, He et al., 2021], where the agent can achieve $\log$ dependence on the number of episodes $K$ under additional dependence on (the inverse of) the minimal gap $\frac{1}{\Delta_{\min}}$. Simchowitz and Jamieson [2019] reports that, similar to stochastic MABs [Lattimore and Szepesvári, 2020], the regret in the gap-dependent setting must scale as $\Omega(\log K)$, which implies that the $\log K$ upper bound is asymptotically tight. However, all of these works treat the customers equivalently and ignore the opportunities of leveraging the tiered structure. Recently, [Papini et al., 2021] achieved similar constant regret in online setting with linear function approximation. Comparing with ours, they investigated the benefits of good features, while we focus on the benefits of considering a different learning protocol. Besides, although their results were established on a more general linear setting, their assumptions on the feature and the uniqueness of optimal policy are quite restrictive in tabular setting.

**Offline RL**   Offline RL considers how to learn a good policy with a fixed dataset [Levine et al., 2020]. Without the requirement of exploration, offline RL prefers algorithms with strong guarantees for exploitation and safety, and Pessimism in the Face of Uncertainty (PFU) becomes a major principle for achieving this both theoretically and empirically [Yin and Wang, 2021, Uehara and Sun, 2022, Liu et al., 2020, Xie et al., 2021, Buckman et al., 2020, Kumar et al., 2020, Fujimoto and Gu, 2021, Yu et al., 2020]. Similar to the offline setting, we choose $\mathrm{Alg}^{\mathrm{E}}$ to be a pessimistic algorithm. However, we still consider to interact the environment with $\mathrm{Alg}^{\mathrm{E}}$, although we ignore the data collected by $\mathrm{Alg}^{\mathrm{E}}$ for now and leave the investigation of its value to future work. As another difference, offline RL assumes the dataset is fixed and only the final performance matters, whereas we evaluate the accumulative regret of $\mathrm{Alg}^{\mathrm{E}}$.

# B   More Discussion about Framework. 1 and Motivating Examples

In this section, we try to justify that our Frw. 1 is an appropriate abstraction for our motivating examples and user-interaction real-world applications.

In the standard online learning protocol, at iteration $k$, the algorithm Alg compute a policy $\pi_k$ based on previous exploration data, the environment samples a user $u_k$ from $\mathrm{G}^{\mathrm{O}}$ and $\mathrm{G}^{\mathrm{E}}$ according to the probability where $P(u_k \in \mathrm{G}^{\mathrm{O}})$ and $P(u_k \in \mathrm{G}^{\mathrm{E}})$, respectively (note that $P(u_k \in \mathrm{G}^{\mathrm{O}}) + P(u_k \in \mathrm{G}^{\mathrm{E}}) = 1$). After that, $\pi_k$ will interact with $u_k$ and obtain a new trajectory for the future learning, while $u_k$ suffers loss $V_1^* - V^{\pi_k}$, and the expected accumulative loss suffered by users from two groups till step $K$ is

$$\mathrm{Regret}_K(\mathrm{Alg}) := \mathbb{E}[\sum_{k=1}^{K} V_1^*(s_1) - V^{\pi_k}(s_1)]$$

Now, we consider a realistic assumption about the probability of $u_k$ from different groups:

**Assumption 1** (Assumption on the Ratio between Users from Different Groups). *We assume that:*

$$\frac{P(u_k \in G^E)}{P(u_k \in G^O)} = C, \quad \forall k \geq 1$$

*for some constant $C$.*

Based on the assumption above, if we do not leverage the tiered structure, and treat the users from different groups equivalently, then the loss suffered by each group will be proportional to the size of

that group. Therefore, even if we assume Alg is near-optimal, the expected loss for each group will scale with $\log K$, i.e.:

$$\text{Loss}_K(\mathrm{G}^{\mathrm{O}}) :=\mathbb{E}[\sum_{k=1}^{K} \mathbb{I}[u_k \in \mathrm{G}^{\mathrm{O}}](V_1^*(s_1) - V^{\pi_k}(s_1))]$$

$$=\frac{1}{1+C}\mathbb{E}[\sum_{k=1}^{K} V_1^*(s_1) - V^{\pi_k}(s_1)] = O(\frac{\log K}{1+C}) \tag{2}$$

$$\text{Loss}_K(\mathrm{G}^{\mathrm{E}}) :=\mathbb{E}[\sum_{k=1}^{K} \mathbb{I}[u_k \in \mathrm{G}^{\mathrm{E}}](V_1^*(s_1) - V^{\pi_k}(s_1))]$$

$$=\frac{C}{1+C}\mathbb{E}[\sum_{k=1}^{K} V_1^*(s_1) - V^{\pi_k}(s_1)] = O(\frac{C\log K}{1+C}) \tag{3}$$

Besides, we can also leverage the tiered structure, and consider an alternative protocol below:

---

**Algorithm 4:** Online Interaction Protocol after Leveraging Tiered Structure

---

1 **Initialize**: $D_1 \leftarrow \{\}$, $\quad k = 1$, $\quad \pi_1^{\mathrm{O}} \leftarrow \text{Alg}^{\mathrm{O}}(D_1)$, $\quad \pi_1^{\mathrm{E}} \leftarrow \text{Alg}^{\mathrm{E}}(D_1)$
2 **for** $k = 1, 2, ...$ **do**
3 $\quad$ User $u_k$ comes.
4 $\quad$ **if** $u_k \in G^O$ **then**
5 $\quad\quad$ Use $\pi_k^{\mathrm{O}}$ to interact with $u_k$, and collect data $\tau_k^{\mathrm{O}}$.
6 $\quad\quad$ $D_{k+1} = D_k \cup \{\tau_k^{\mathrm{O}}\}$, $\quad \pi_{k+1}^{\mathrm{O}} \leftarrow \text{Alg}^{\mathrm{O}}(D_k)$, $\quad \pi_{k+1}^{\mathrm{E}} \leftarrow \text{Alg}^{\mathrm{E}}(D_k)$
7 $\quad$ **end**
8 $\quad$ **else**
9 $\quad\quad$ $\pi_k^{\mathrm{E}}$ interacts with $u_k$, and collect data $\tau_k^{\mathrm{E}}$. // We do not use $\tau_k^{\mathrm{E}}$ for now.
10 $\quad\quad$ $D_{k+1} = D_k \cup \{\tau_k^{\mathrm{E}}\}$, $\quad \pi_{k+1}^{\mathrm{O}} \leftarrow \pi_k^{\mathrm{O}}$, $\quad \pi_{k+1}^{\mathrm{E}} \leftarrow \pi_k^{\mathrm{E}}$
11 $\quad$ **end**
12 **end**

---

In another word, in this new protocol, we use two policies at different exploitation level to interact with users from different groups, and only update policies if user comes from group $\mathrm{G}^{\mathrm{O}}$. Note that in expectation, $\{u_k \in \mathrm{G}^{\mathrm{O}}\}$ will happen for $\frac{K}{1+C}$ times, and therefore we have:

$$\text{Loss}'_K(\mathrm{G}^{\mathrm{O}}) :=\mathbb{E}[\sum_{k=1}^{K} \mathbb{I}[u_k \in \mathrm{G}^{\mathrm{O}}](V_1^*(s_1) - V^{\pi_k^{\mathrm{O}}}(s_1))]$$

$$\approx \text{Regret}_{K/(1+C)}(\text{Alg}^{\mathrm{O}})$$

$$\text{Loss}'_K(\mathrm{G}^{\mathrm{E}}) :=\mathbb{E}[\sum_{k=1}^{K} \mathbb{I}[u_k \in \mathrm{G}^{\mathrm{E}}](V_1^*(s_1) - V^{\pi_k^{\mathrm{E}}}(s_1))]$$

$$\approx C\text{Regret}_{K/(1+C)}(\text{Alg}^{\mathrm{E}})$$

where $\text{Regret}_{(.)}(\text{Alg}^{\mathrm{O}})$ and $\text{Regret}_{(.)}(\text{Alg}^{\mathrm{E}})$ are originally defined in Def. 2.1, and they are exactly the metric we used to measure the performance of $\text{Alg}^{\mathrm{E}}$ and $\text{Alg}^{\mathrm{O}}$ under our Frw. 1.

Based on our results in Sec. 4.1 and 4.2, we know that under our framework, it is possible to achieve that:

$$\text{Loss}'_K(\mathrm{G}^{\mathrm{O}}) =\text{Regret}_{K/(1+C)}(\text{Alg}^{\mathrm{O}}) = O(\log \frac{K}{1+C}) \tag{4}$$

$$\text{Loss}'_K(\mathrm{G}^{\mathrm{E}}) =C\text{Regret}_{K/(1+C)}(\text{Alg}^{\mathrm{E}}) = C \cdot \text{constant} \tag{5}$$

where *constant* means independence of $K$ but may include dependence on other parameters such as $S, A, H, \Delta$. Comparing with Eq.(2), (3), (4), and (5), we can see that users from $\mathrm{G}^{\mathrm{O}}$ will suffer less regret than before because we "transfer" some the regret from $\mathrm{G}^{\mathrm{E}}$ to $\mathrm{G}^{\mathrm{O}}$, and the additional regret suffered by $\mathrm{G}^{\mathrm{O}}$ can be compensated in other forms as we discussed in Sec. 1.

**Remark** Besides, our methods and results can be applied to those scenarios suggested by the decoupling setting [Avner et al., 2012, Rouyer and Seldin, 2020], where $\text{Alg}^\text{E}$ does not necessarily interact with the environment, and we omit the discussion here.

## C  Lower Bounds

### C.1  Regret Lower Bounds for Tabular MDP without Strictly Positive Gap Assumption

We first recall a Theorem from [Dann et al., 2017]:

**Theorem C.1** (Theorem C.1 in [Dann et al., 2017])**.** *There exist positive constant $c$, $\delta_0 > 0$, $\varepsilon_0 > 0$, such that for every $\varepsilon \in (0, \varepsilon_0)$, $S \geq 4$, $A \geq 2$ and for every algorithm Alg and $n \leq \frac{cASH^3}{\varepsilon^2}$ there is a fixed-horizon episodic MDP $M_{hard}$ with time-dependent transition probabilities and $\check{S}$ states and $A$ actions so that returning an $\varepsilon$-optimal policy after n episodes is at most $1 - \delta_0$.*

**Theorem 3.1.** *[Lower Bound for $\text{Alg}^E$ without Gap Assumption] There exist positive constants $c, \varepsilon_0, \delta_0$, such that, for arbitrary $S \geq 4$, $A \geq 2$, $H \geq 2$, $K \geq \frac{c}{\varepsilon_0^2} H^3 SA$, and arbitrary algorithm pair $(Alg^O, Alg^E)$, there must exist a hard tabular MDP $M_{hard}$, $\mathbb{E}_{(Alg^O, Alg^E), M_{hard}} \left[ \sum_{k=1}^K V^* - V^{\pi_k^E} \right] \geq \delta_0 \sqrt{cH^3 SAK}$, where the expectation is taken over the randomness of algorithms and MDP.*

*Proof.* Suppose we have a pair algorithm $(\text{Alg}^\text{O}, \text{Alg}^\text{E})$, we can construct a PAC algorithm with $\text{Alg}^\text{O}$ and $\text{Alg}^\text{E}$, in the following way:

- Input: $K$.

- For $k = 1, 2, ..., K$, run $\text{Alg}^\text{O}$ to collect data and run $\text{Alg}^\text{E}$ to generate a sequence of policies $\pi_1^\text{E}, ..., \pi_K^\text{E}$.

- Uniformly randomly select an index from $\{1, 2, ..., K\}$, and denote it as $K_\text{PAC}$

- Output $\pi_{K_\text{PAC}}^\text{E}$.

In the following, we denote such an algorithm as $\text{Alg}_\text{PAC}$. Then, for an arbitrary MDP $M$, we must have:

$$\mathbb{E}_{\text{Alg}_\text{PAC}, M}[V^* - V^{\pi_{K_\text{PAC}}}] = \frac{1}{K} \mathbb{E}_{(\text{Alg}^\text{O}, \text{Alg}^\text{E}), M}[\sum_{k=1}^K V^* - V^{\pi_k}]$$

As a result of Markov inequality and that $V^* - V^\pi \geq 0$ for arbitrary $\pi$, for arbitrary $\varepsilon > 0$, we have:

$$\Pr(V^* - V^{\pi_{K_\text{PAC}}} \geq \varepsilon) \leq \frac{\mathbb{E}_{\text{Alg}_\text{PAC}, M}[V^* - V^{\pi_{K_\text{PAC}}}]}{\varepsilon} \tag{6}$$

Since the above holds for arbitrary $\varepsilon \in (0, \varepsilon_0)$, by choosing $\varepsilon = \bar\varepsilon := \sqrt{cH^3 SA/K}$, since $K > \frac{c}{\varepsilon_0^2} H^3 SA$, we have $\bar\varepsilon < \varepsilon_0$ and

$$\Pr(V^* - V^{\pi_{K_\text{PAC}}} \geq \bar\varepsilon) \leq \frac{\mathbb{E}_{\text{Alg}_\text{PAC}, M}[V^* - V^{\pi_{K_\text{PAC}}}]}{\sqrt{cH^3 SA/K}}. \tag{7}$$

Because $K \leq cH^3 SA/\bar\varepsilon^2$, Thm. C.1 implies that, for $\text{Alg}_\text{PAC}$, for arbitrary $S \geq 4$, $A \geq 2$, $H \geq 1$, there must exists a hard MDP $M_{hard}$, such that:

$$\frac{\mathbb{E}_{\text{Alg}_\text{PAC}, M}[V^* - V^{\pi_{K_\text{PAC}}}]}{\sqrt{cH^3 SA/K}} \geq \delta_0$$

and it is equivalent to:

$$\mathbb{E}_{(\text{Alg}^\text{O}, \text{Alg}^\text{E}), M}[\sum_{k=1}^K V^* - V^{\pi_k}] = K \cdot \mathbb{E}_{\text{Alg}_\text{PAC}, M}[V^* - V^{\pi_{K_\text{PAC}}}] \geq \delta_0 \sqrt{cH^3 SAK} \tag{8}$$

which finishes the proof.

$\square$

**Remark C.2** (Regret lower bound when $\tau^E$ is used in Framework 1). *Our techniques can be extended to the case when $\tau^E$ are used by $Alg^O$ and $Alg^E$, and establish the same $O(\sqrt{H^3SAK})$ lower bound. Because the only difference would be in this new setting, at iteration $k$, $Alg^O$ and $Alg^E$ will use $2k$ trajectories to compute $\pi_k^O$ and $\pi_k^E$, which only double the sample size comparing with $k$ trajectories in Framework 1. Therefore, with the same techniques (and choosing $\bar\varepsilon = \sqrt{cH^3SA/(2K)}$), one can obtain a lower bound which differs from Eq.(8) by constant.*

## C.2 Lower Bound for the Dependence on $\log d_{\min}$ when $|\Pi^*| = 1$

In this section, because we will conduct discussion on multiple different MDPs, and in order to distinguish them, we will introduce a subscript of $M$ to highlight which MDP we are discussing. Therefore, we revise some key notations and re-introduce them here.

**Notation**  Given arbitrary tabular MDP $M = \{\mathcal{S}, \mathcal{A}, \mathbb{P}, r, H\}$. We use $\Pi_M^*$ to denote the set of deterministic optimal policies of $M$. For each deterministic optimal policy $\pi_M^* \in \Pi_M^*$, we define $d_{\min}^{\pi_M^*}$ to be the minimal non-zero occupancy of the reachable state action by $\pi_M^*$, i.e.

$$d_{\min}^{\pi_M^*} := \min_{h, s_h, a_h} d^{\pi_M^*}(s_h, a_h), \quad s.t. \quad d^{\pi_M^*}(s_h, a_h) > 0 \tag{9}$$

Then, we define:

$$d_{M,\min} := \min_{\pi_M^* \in \Pi_M^*} d_{\min}^{\pi_M^*}, \quad \pi_{M,d_{\min}}^* := \arg\min_{\pi_M^* \in \Pi_M^*} d_{\min}^{\pi_M^*}, \tag{10}$$

Different from regret analysis in online setting [Simchowitz and Jamieson, 2019, Xu et al., 2021, Dann et al., 2021], our regret bound in Thm. 4.9 has additional dependence on $\log d^{\pi^*}$, even if the optimal policy is unique. In Thm. C.3, we show that the dependence of $\log d^{\pi^*}$ is unavoidable if we do not have other assumptions about the behavior of $Alg^O$ besides Cond. 4.6. In another word, even if constrained by satisfies Cond. 4.6, $Alg^O$ can be arbitrarily adversarial so that $\log d^{\pi^*}$ exists in the lower bound. We defer the proof to Appx. C.2

**Theorem C.3.** *For arbitrary $S, A, H \geq 3$, arbitrary $\Delta_{\min} > 0$ and $d_{\min} > 0$, if there exists an MDP $M = \{\mathcal{S}, \mathcal{A}, \mathbb{P}, r, H\}$ such that $|\mathcal{S}| = S$, $|\mathcal{A}| = A$ $d_{M,\min} = d_{\min}$ and the minimal gap is lower bound by $\Delta_{\min}$, then there exists a hard MDP $M^+ = \{\mathcal{S}^+, \mathcal{A}, \mathbb{P}^+, r^+, H\}$ with $|\mathcal{S}^+| = S + 1$, minimal gap lower bounded by $3\Delta_{\min}/4$ and $d_{M^+,\min} = d_{M,\min}/4$, and an adversarial choice of $Alg^O$ satisfying Cond. 4.6, such that when $K$ is large enough, the expected Pseudo-Regret of $Alg^E$ is lower bounded by:*

$$\mathbb{E}_{Alg^O, M, Alg^E}[\sum_{k=1}^K V^* - V^{\pi_k^E}] \geq O\Big((C_1 + C_2) \log \frac{C_1 + C_2}{d_{M^+,\min}\Delta_{\min}}\Big)$$

*Proof.* The proof is divided into three steps.

**Step 1: Construction of the Hard MDP Instance**  Now, we construct a hard MDP instance $M^+ := \{\mathcal{S}^+, \mathcal{A}^+, \mathbb{P}^+, r^+, H\}$ based on $M$ by expanding the state space with an absorbing state $s_{absorb}$ for layer $h \geq 2$ (we use $h$ in $s_{h,absorb}$ to distinguish the absorbing state at different time step), and define the transition and reward function by:

$$\forall a_1 \in \mathcal{A}_1, s_2 \in \mathcal{S}_2, \quad \mathbb{P}^+(s_{2,absorb}|s_1, a_1) = \frac{d_{M,\min}}{4},$$

$$\mathbb{P}^+(s_2|s_1, a_1) = (1 - \frac{d_{M,\min}}{4})\mathbb{P}(s_2|s_1, a_1)$$

$$r^+(s_1, a_1) = (1 - \frac{d_{M,\min}}{4})r(s_1, a_1)$$

$$\forall h \geq 2, s_h \in \mathcal{S}_h, a_h \in \mathcal{A}_h, \quad \mathbb{P}^+(\cdot|s_h, a_h) = \mathbb{P}(\cdot|s_h, a_h), \quad r^+(s_h, a_h) = r(s_h, a_h)$$

$$\forall h \geq 2, a_h \in \mathcal{A}_h, \quad \mathbb{P}^+(s_{h+1,absorb}|s_{h,absorb}, a_h) = 1$$

$$\forall H \geq h \geq 2, a_h \in \mathcal{A}_h, \quad r(s_{h,absorb}, a_h) = \Delta_{\min}\mathbb{I}[a_h = a_h^*]$$

Briefly speaking, at the initial state, by taking arbitrary action, with probability $d_{M,\min}/2$, it will transit to absorbing state at layer 2, and the agent can not escape from the absorbing state till the end of the episodes. Besides, at the absorbing states, for each layer $2 \leq h \leq H$, there always exists an optimal action $a_h^*$ with reward $\Delta_{\min}$ and taking any the other actions will lead to 0 reward. Moreover, $M^+$ agrees with $M$ for all the transition and rewards when $h \geq 2$.

Easy to see that:

$$V_{M^+}^*(s_1) - Q_{M^+}^*(s_1, a_1) = (1 - \frac{d_{M,\min}}{4})(V_M^*(s_1) - Q_M^*(s_1, a_1))$$

Therefore, if $V_M^*(s_1) - Q_M^*(s_1, a_1) > 0$, we still have:

$$V_{M^+}^*(s_1) - Q_{M^+}^*(s_1, a_1) \geq \frac{3}{4}(V_M^*(s_1) - Q_M^*(s_1, a_1)) \geq \frac{3}{4}\Delta_{\min}$$

Combining with the transition and reward functions in absorbing states, we can conclude that the gap of $M^+$ is still $O(\Delta_{\min})$.

**Step 2: Construction of Adversarial Alg$^O$** Let's use $\Pi_{M^+}^*$ to denote the set of deterministic optimal policies at MDP $M^+$. It's easy to see that, for arbitrary $\pi_{M^+}^* \in \Pi_{M^+}^*$, there must exists an optimal policy $\pi_M^* \in \Pi_M^*$ agrees with $\pi_{M^+}^*$ at all non-absorbing states (and vice versa), i.e.

$$\pi_{M^+}^*(s_h) = \pi_M^*(s_h), \quad \forall h \in [H], s_h \in \mathcal{S}_h$$

Then, for arbitrary $\pi_{M^+}^* \in \Pi_{M^+}^*$, we have:

$$d^{\pi_{M^+}^*}(s_{h,absorb}) = \frac{d_{M,\min}}{4} \leq (1 - \frac{d_{M,\min}}{4})d_{M,\min} < d^{\pi_{M^+}^*}(s_{h'}), \quad \forall h' \in [H], s_{h'} \in \mathcal{S}_{h'}$$

which implies that

$$d_{M^+,\min} = \frac{d_{M,\min}}{4}$$

and $s_{h,absorb}$ are the hardest state to reach for all deterministic optimal policies. In the following, we randomly choose an optimal deterministic policy $\pi_{M^+}^*$ from $\Pi_{M^+}^*$, and randomly select one action $\bar{a}_H$ from $\mathcal{A}_H$ with $\bar{a}_H \neq a_H^*$ and fix them in the following discussion.

Based on the definition above, we define a deterministic policy in $M^+$, which agree with $\pi_{M^+}^*$ for all states except $s_H$:

$$\forall h \in [H], \quad \pi_{M^+}(s_h) = \begin{cases} \pi_{M^+}^*(s_h), & \text{if } s_h \neq s_{H,absorb}, \\ \bar{a}_H, & \text{if } s_h = s_{H,absorb}, \end{cases}.$$

Now, we are ready to design the adversarial choice of Alg$^O$ satisfying the condition 4.6. We consider the following algorithm:

$$\text{Alg}^O(k) = \begin{cases} \pi_{M^+}, & \text{if } k \leq k_{\sup}, \\ \pi_{M^+}^*, & \text{if } k > k_{\sup}, \end{cases};$$

where $k_{\sup}$ is defined to be:

$$k_{\sup} := \sup_{k \in \mathbb{N}^+} : \{k \leq \frac{1}{d_{M^+,\min}\Delta_{\min}}(C_1 + C_2 \log k)\} \approx O(\frac{C_1 + C_2}{d_{M^+,\min}\Delta_{\min}} \log \frac{C_1 + C_2}{d_{M^+,\min}\Delta_{\min}})$$

We can easily verify that Cond. 4.6 will not be violated, since

$$\forall k \geq 1, \quad \sum_{k=1}^{K} V^* - V^{\pi_k^O}$$

$$\leq d_{M^+,\min} \cdot (V^*(s_{2,absorb}) - V^{\pi_k^O}(s_{2,absorb})) \cdot \min\{k, k_{\sup}\}$$

$$\leq d_{M^+,\min} \cdot (V^*(s_{H,absorb}) - V^{\pi_k^O}(s_{H,absorb})) \cdot \frac{1}{d_{M^+,\min}\Delta_{\min}}(C_1 + C_2 \log \min\{k, k_{\sup}\})$$

$$= d_{M^+,\min} \cdot \Delta_{\min} \cdot \frac{1}{d_{M^+,\min}\Delta_{\min}}(C_1 + C_2 \log k)$$

$$\leq C_1 + C_2 \log k$$

**Step 3: Lower Bound of Alg$^{\mathrm{E}}$ under the Choice of Adversarial Alg$^{\mathrm{O}}$**   Now, we can derive an lower bound for Alg$^{\mathrm{E}}$. Since in the first $k_{\mathrm{sup}}$ steps, Alg$^{\mathrm{E}}$ can only observe what happens if action $\bar{a}_H$ is taken at $s_{H,absorb}$, and therefore, it has no idea about which action among $\mathcal{A}_H \setminus \bar{a}_H$ is the optimal action $a_H^*$. We use $\mathcal{M}^+$ to denote a set of MDPs by permuting the position of $a_H^*$ in $M^+$. Since $|\mathcal{A}_H| = A$, we have $|\mathcal{M}^+| = A - 1$ and $M^+ \in \mathcal{M}^+$.

Then, we uniformly sample an MDP from $\mathcal{M}^+$ and run the adversarial Alg$^{\mathrm{O}}$ above to generate the data for Alg$^{\mathrm{E}}$ to learn. We use $M_i^+$ with $i = 1, 2..., A - 1$ to refer to the MDPs in $\mathcal{M}^+$ and use index $i$ to refer to the position of the optimal action at $s_{H,absorb}$ in each MDP. For the simplicity of the notation, we use $A$ as the index to refer to the position of $\bar{a}_H$.

Because Alg$^{\mathrm{E}}$ do not have prior knowledge about which MDP in $\mathcal{M}^+$ is sampled, we have:

$$
\mathbb{E}_{\bar{M}^+,\mathrm{Alg}^{\mathrm{O}},\mathrm{Alg}^{\mathrm{E}}}\Big[\sum_{k=1}^{K} V^* - V^{\pi_k^{\mathrm{E}}}\Big]
$$

$$
\geq \mathbb{E}_{\bar{M}^+,\mathrm{Alg}^{\mathrm{O}},\mathrm{Alg}^{\mathrm{E}}}\Big[\sum_{k=1}^{k_{\mathrm{sup}}} V^* - V^{\pi_k^{\mathrm{E}}}\Big]
$$

$$
= \frac{1}{A-1} \sum_{i\in[A-1]} \mathbb{E}_{M_i^+,,\mathrm{Alg}^{\mathrm{O}},\mathrm{Alg}^{\mathrm{E}}}\Big[\sum_{k=1}^{k_{\mathrm{sup}}} V^* - V^{\pi_k^{\mathrm{E}}}\Big]
$$

$$
\geq \frac{d_{M,\min}\Delta_{\min}}{A-1} \sum_{k=1}^{k_{\mathrm{sup}}} \sum_{i\in[A-1]} \sum_{j\in[A],j\neq i} \mathrm{Pr}_{\mathrm{Alg}^{\mathrm{O}},M_i}\big(\pi_k^{\mathrm{E}}(s_{H,absorb}) = j\big)
$$

$$
\text{(Drop the probability that } \pi_k^{\mathrm{E}} \text{ is sub-optimal at non-absorbing states)}
$$

$$
= \frac{d_{M,\min}\Delta_{\min}}{A-1} \sum_{k=1}^{k_{\mathrm{sup}}} \Big( \sum_{j\in[A],j\neq A-1} \mathrm{Pr}_{\mathrm{Alg}^{\mathrm{O}},M_i}\big(\pi_k^{\mathrm{E}}(s_{H,absorb}) = j\big)
$$

$$
\qquad\qquad + \sum_{i\in[A-2]} \sum_{j\in[A],j\neq i} \mathrm{Pr}_{\mathrm{Alg}^{\mathrm{O}},M_i}\big(\pi_k^{\mathrm{E}}(s_{H,absorb}) = j\big) \Big)
$$

$$
= \frac{d_{M,\min}\Delta_{\min}}{A-1} \sum_{k=1}^{k_{\mathrm{sup}}} \Big( \sum_{i\in[A-2]} \mathrm{Pr}_{\mathrm{Alg}^{\mathrm{O}},M_i}\big(\pi_k^{\mathrm{E}}(s_{H,absorb}) = i\big)
$$

$$
\qquad\qquad + \sum_{i\in[A-2]} \sum_{j\in[A],j\neq i} \mathrm{Pr}_{\mathrm{Alg}^{\mathrm{O}},M_i}\big(\pi_k^{\mathrm{E}}(s_{H,absorb}) = j\big) \Big)
$$

$$
\text{(Alg}^{\mathrm{E}} \text{ can not distinguish between } M_i^+)
$$

$$
= \frac{d_{M,\min}\Delta_{\min}}{A-1} \sum_{k=1}^{k_{\mathrm{sup}}} \Big( \sum_{i\in[A-2]} \sum_{j\in[A]} \mathrm{Pr}_{\mathrm{Alg}^{\mathrm{O}},M_i}\big(\pi_k^{\mathrm{E}}(s_{H,absorb}) = j\big)
$$

$$
= \frac{A-2}{A-1} d_{M,\min}\Delta_{\min} k_{\mathrm{sup}}
$$

$$
= O\Big((C_1 + C_2)\log \frac{C_1 + C_2}{d_{M^+,\min}\Delta_{\min}}\Big) = O\Big((C_1 + C_2)\log \frac{C_1 + C_2}{d_{M,\min}\Delta_{\min}}\Big)
$$

$\square$

# D  Analysis for Bandit Setting

## D.1  The Optimality of Alg$^{\text{O}}$ in Alg 2

From Theorem 8.1 of [Lattimore and Szepesvári, 2020], we can show the following guarantee for the UCB algorithm in Alg 2 with revised bonus function:

$$\mathbb{E}[\sum_{k=1}^{K} \mu_1 - \mu_{\pi_k^{\text{O}}}] \leq \sum_{i:\Delta_i>0} \Delta_i + \frac{1}{\Delta_i}(8\alpha \log f(K) + 8\sqrt{\pi\alpha \log f(K)} + 28)$$

$$= O(\sum_{i:\Delta_i>0} \frac{\alpha}{\Delta_i} \log AK)$$

where we assume $K \geq A$. Since $\alpha$ is just at the constant level, the above regret still matches the lower bound.

## D.2  Analysis for LCB

**Outline**   In this section, we establish regret bound for Alg. 2. We first provide the proof of two key Lemma: Lem. 4.2 and Lem. 4.3. After that, in Lem. 4.3, we try to combine the above two results and prove that for those arm $i$ with $\Delta_i > 0$, when $k$ is large enough, we are almost sure (with high probability) that LCB will not take arm $i$. Finally, we conclude this section with the proof of Thm. 4.1.

**Definition of $c_f$**   Under the choice $f(k) = 1 + 16A^2(k+1)^2$, we use $c_f$ to denote the minimal positive constant independent with $\alpha, A$ and arbitrary $\Delta_i$ with $\Delta_i > 0$, such that

$$\forall \Delta_i > 0, \forall \lambda \in [1, 4A], \quad \text{as long as} \quad k \geq c_f \frac{\alpha\lambda}{\Delta_i^2} \log(1 + \frac{\alpha A}{\Delta_i}),$$

$$\text{we have} \quad k \geq \frac{32\alpha\lambda}{\Delta_i^2} \log f(k) \tag{11}$$

**Lemma 4.2.**  *[Blessing of Pessimism] With the choice that $f(k) = 1 + 16A^2(k+1)^2$, for arbitrary $i$ with $\Delta_i > 0$, for the LCB algorithm in Alg 2, and arbitrary $j$ satisfying $\Delta_j < \Delta_i$, we have:*
$\Pr\left(\{i = \pi_k^E\} \cap \{\Delta_j < \Delta_i\} \cap \left\{N_j(k) \geq \frac{8\alpha \log f(k)}{(\Delta_j - \Delta_i)^2}\right\}\right) \leq \frac{2}{k^{2\alpha}}.$

*Proof.*

$$\Pr(\{i = \pi_k^E\} \cap \{\Delta_j < \Delta_i\} \cap \{N_j(k) \geq \frac{8\alpha \log f(k)}{(\Delta_j - \Delta_i)^2}\})$$

$$\leq \Pr(\{\widehat{\mu}_i(k) - \sqrt{\frac{2\alpha \log f(k)}{N_i(k)}} \geq \widehat{\mu}_j(k) - \sqrt{\frac{2\alpha \log f(k)}{N_j(k)}}\} \cap \{\Delta_j < \Delta_i\} \cap \{N_j(k) \geq \frac{8\alpha \log f(k)}{(\Delta_j - \Delta_i)^2}\})$$

$$= \Pr(\{\widehat{\mu}_i(k) - \mu_i - \sqrt{\frac{2\alpha \log f(k)}{N_i(k)}} \geq \widehat{\mu}_j(k) - \mu_j + (\Delta_j - \Delta_i) - \sqrt{\frac{2\alpha \log f(k)}{N_j(k)}}\}$$

$$\cap \{\Delta_j < \Delta_i\} \cap \{N_j(k) \geq \frac{8\alpha \log f(k)}{(\Delta_j - \Delta_i)^2}\})$$

$$\leq \Pr(\{\widehat{\mu}_i(k) - \mu_i - \sqrt{\frac{2\alpha \log f(k)}{N_i(k)}} \geq \widehat{\mu}_j(k) - \mu_j + \sqrt{\frac{2\alpha \log f(k)}{N_j(k)}}\} \cap \{\Delta_j < \Delta_i\} \cap \{N_j(k) \geq \frac{8\alpha \log f(k)}{(\Delta_j - \Delta_i)^2}\})$$

$$\leq \Pr(\{\widehat{\mu}_i(k) - \mu_i - \sqrt{\frac{2\alpha \log f(k)}{N_i(k)}} \geq \widehat{\mu}_j(k) - \mu_j + \sqrt{\frac{2\alpha \log f(k)}{N_j(k)}}\})$$

$$\leq \Pr(\{\widehat{\mu}_i(k) - \mu_i - \sqrt{\frac{2\alpha \log f(k)}{N_i(k)}} \geq 0\}) + \Pr(\{0 \geq \widehat{\mu}_j(k) - \mu_j + \sqrt{\frac{2\alpha \log f(k)}{N_j(k)}}\})$$

$\leq 2/f(k)^\alpha \leq 2/k^{2\alpha}$

where the last but two step is because of the Azuma-Hoeffding's inequality. $\qquad\square$

**Lemma 4.3.** *[Property of UCB] With the choice that $f(k) = 1 + 16A^2(k+1)^2$, there exists a constant c, for arbitrary $i$ with $\Delta_i > 0$ and arbitrary $\lambda \in [1, 4A]$, in UCB algorithm, we have:*
$\Pr(N_i(k) \geq \frac{k}{\lambda}) \leq \frac{2}{k^{2\alpha-1}}, \quad \forall k \geq \lambda + c \cdot \frac{\alpha\lambda}{\Delta_i^2} \log(1 + \frac{\alpha A}{\Delta_{\min}}).$

*Proof.* We choose $c_f$ defined in Eq.(11) to be the constant $c$ in this Lemma.

The key idea of the proof is that, because $N_i(k) \leq k$ for all $k$, if $N_i(k) \geq \lceil k/\lambda \rceil$, there must exists an iteration $\widetilde{k}$ between $\lceil k/\lambda \rceil - 1$ and $k$, such that $\{N_i(\widetilde{k}) = \lceil k/\lambda \rceil - 1\} \cap \{N_i(\widetilde{k}) = \lceil k/\lambda \rceil\}$ (i.e. $\widetilde{k}$ is the time step that UCB takes arm $i$ for the $\lceil k/\lambda \rceil$-th time). Therefore, for arbitrary fixed $\lambda \in [1, A^2]$, when $k \geq \lambda + c_f \cdot \frac{\alpha\lambda}{\Delta_i^2} \log(1 + \frac{\alpha\lambda}{\Delta_{\min}})$, we have:

$$\Pr(N_i(k) \geq k/\lambda) = \Pr(N_i(k) \geq \lceil k/\lambda \rceil)$$

$$= \sum_{\widetilde{k}=\lceil k/\lambda \rceil - 1}^{k-1} \Pr(\{N_i(\widetilde{k}) = \lceil k/\lambda \rceil - 1, N_i(\widetilde{k}+1) = \lceil k/\lambda \rceil\} \cap \{\widehat{\mu}_{i^*}(\widetilde{k}) + \sqrt{\frac{2\alpha \log f(\widetilde{k})}{N_{i^*}(\widetilde{k})}} \leq \widehat{\mu}_i(\widetilde{k}) + \sqrt{\frac{2\alpha \log f(\widetilde{k})}{N_i(\widetilde{k})}}\})$$

$$\text{(Union bound.)}$$

$$= \sum_{\widetilde{k}=\lceil k/\lambda \rceil - 1}^{k-1} \Pr(\{N_i(\widetilde{k}) = \lceil k/\lambda \rceil - 1, N_i(\widetilde{k}+1) = \lceil k/\lambda \rceil\}$$

$$\cap \{\widehat{\mu}_{i^*}(\widetilde{k}) - \mu_{i^*} + \sqrt{\frac{2\alpha \log f(\widetilde{k})}{N_{i^*}(\widetilde{k})}} \leq \widehat{\mu}_i(\widetilde{k}) - \mu_i - \Delta_i + \sqrt{\frac{2\alpha \log f(\widetilde{k})}{N_i(\widetilde{k})}}\})$$

$$\text{(Subtract } \mu_{i^*} \text{ at both sides)}$$

$$\leq \sum_{\widetilde{k}=\lceil k/\lambda \rceil - 1}^{k-1} \Pr(\{N_i(\widetilde{k}) = \lceil k/\lambda - 1 \rceil, N_i(\widetilde{k}+1) = \lceil k/\lambda \rceil\} \cap \{\widehat{\mu}_{i^*}(\widetilde{k}) - \mu_{i^*} + \sqrt{\frac{2\alpha \log f(\widetilde{k})}{N_{i^*}(\widetilde{k})}} \leq 0\})$$

$$+ \Pr(\{N_i(\widetilde{k}) = \lceil k/\lambda \rceil - 1), N_i(\widetilde{k}) = \lceil k/\lambda \rceil + 1)\} \cap \{0 \leq \widehat{\mu}_i(\widetilde{k}) - \mu_i - \Delta_i + \sqrt{\frac{2\alpha \log f(\widetilde{k})}{N_i(\widetilde{k})}}\})$$

$$(12)$$

$$\leq \sum_{\widetilde{k}=\lceil k/\lambda \rceil - 1}^{k-1} \Pr(\{\widehat{\mu}_{i^*}(\widetilde{k}) - \mu_{i^*} + \sqrt{\frac{2\alpha \log f(\widetilde{k})}{N_{i^*}(\widetilde{k})}} \leq 0\})$$

$$+ \Pr(\{N_i(\widetilde{k}) = \lceil k/\lambda \rceil - 1), N_i(\widetilde{k}+1) = \lceil k/\lambda \rceil)\} \cap \{0 \leq \widehat{\mu}_i(\widetilde{k}) - \mu_i - \sqrt{\frac{2\alpha \log f(\widetilde{k})}{N_i(\widetilde{k})}}\})$$

$$\text{(Under our choice of } k, \text{ and } N_i(\widetilde{k}) = \lceil k/\lambda \rceil - 1, \frac{\log f(\widetilde{k})}{N_i(\widetilde{k})} \leq \frac{\log f(k)}{k/\lambda - 1} \leq \frac{\Delta_i^2}{8\alpha})$$

$$\leq \sum_{\widetilde{k}=\lceil k/\lambda \rceil - 1}^{k-1} \Pr(\{\widehat{\mu}_{i^*}(\widetilde{k}) - \mu_{i^*} + \sqrt{\frac{2\alpha \log f(\widetilde{k})}{N_{i^*}(\widetilde{k})}} \leq 0\}) + \Pr(\{0 \leq \widehat{\mu}_i(\widetilde{k}) - \mu_i - \sqrt{\frac{2\alpha \log f(\widetilde{k})}{N_i(\widetilde{k})}}\})$$

$$\leq \sum_{\widetilde{k}=\lceil k/\lambda \rceil - 1}^{k-1} \frac{2}{f(\widetilde{k})^\alpha} \qquad\qquad \text{(Azuma-Hoeffding Inequality)}$$

$$\leq \frac{2k}{f(k/\lambda - 1)^\alpha} = \frac{2k}{(16A^2k^2/\lambda^2 + 1)^\alpha} \leq \frac{2}{k^{2\alpha-1}} \qquad\qquad (\lambda \in [1, 4A])$$

where the step (12) is because:

$$\{\widehat{\mu}_{i^*}(\widetilde{k}) - \mu_{i^*} + \sqrt{\frac{2\alpha \log f(\widetilde{k})}{N_{i^*}(\widetilde{k})}} < \widehat{\mu}_i(\widetilde{k}) - \mu_i - \Delta_i + \sqrt{\frac{2\alpha \log f(\widetilde{k})}{N_i(\widetilde{k})}}\}$$

$$\in \{0 < \widehat{\mu}_i(\widetilde{k}) - \mu_i - \Delta_i + \sqrt{\frac{2\alpha \log f(\widetilde{k})}{N_i(\widetilde{k})}}\} \cup \{\widehat{\mu}_{i^*}(\widetilde{k}) - \mu_{i^*} + \sqrt{\frac{2\alpha \log f(\widetilde{k})}{N_{i^*}(\widetilde{k})}} \le 0\}$$

$\square$

**Lemma D.1.** *Given an arm $i$, we separate all the arms into two parts depending on whether its gap is larger than $\Delta_i$ and define $G_i^{lower} := \{\iota | \Delta_\iota > \Delta_i/2\}$ and $G_i^{upper} := \{\iota | \Delta_\iota \le \Delta_i/2\}$. With the choice that $f(k) = 1 + 16A^2(k+1)^2$, there is a constant c, such that for arbitrary $i$ with $\Delta_i > 0$, for the LCB algorithm in Alg 2, we have:*

$$\Pr(i = \pi_k^E) \le 2/k^{2\alpha} + 2A/k^{2\alpha-1}, \quad \forall k \ge k_i := 8\alpha c \Big( \sum_{\iota \in G_i^{lower}} \frac{1}{\Delta_\iota^2} + \frac{4|G_i^{upper}|}{\Delta_i^2} \Big) \log(1 + \frac{\alpha A}{\Delta_{\min}}) \tag{13}$$

*where c is the constant considered in Lem. 4.3 (i.e. $c_f$ defined in Eq.(11)).*

*Proof.* We want to remark that the constants in the definition of $k_i$ (i.e. 8 in "$8\alpha c$" and 4 in "$4|G_i^{\text{upper}}|$") can be replaced by others, but we choose them carefully in order to make sure some steps in the proof of this Lemma and Thm. 4.1 can go through.

The main idea of the proof is to use Lem. 4.3 to show that, for those arm $i$ with $\Delta_i > 0$, when $k \ge k_i$, $N_\iota(k)$ will be small for those $\iota \in G_i^{\text{lower}}$. As a result, there must exist an arm $j \in G_i^{\text{upper}}$, such that $N_j(k)$ is large than the threshold considered in Lem. 4.2 and therefore, with high probability, arm $i$ will not be preferred.

First, we try to apply Lem. 4.3 to upper bound the quantity $N_j(k)$ for those arm $j \in G_i^{\text{lower}}$. For each $j \in G_i^{\text{lower}}$, we define the following quantity, which measures the magnitude of $1/\Delta_j^2$ with $k$:

$$\gamma_{k,j} := \frac{k}{\frac{8\alpha c}{\Delta_j^2} \log(1 + \frac{\alpha A}{\Delta_{\min}})}.$$

We only consider $k \ge k_i$, where we always have $\gamma_{k,j} \ge 1$ based on the definition of $k_i$.

Next, we separately consider two cases depending on whether $\gamma_{k,j} > 2A$ or not.

**Case 1:** $\gamma_{k,j} > 2A$: In this case, $\Delta_j$ is relatively large (or say more sub-optimal) comparing with iteration $k$. For arbitrary $k \ge k_i$, we have

$$k = \gamma_{k,j} \cdot \frac{8\alpha c}{\Delta_j^2} \log(1 + \frac{\alpha A}{\Delta_{\min}}) \ge 2A \cdot \frac{8\alpha c}{\Delta_j^2} \log(1 + \frac{\alpha A}{\Delta_{\min}}) \ge 2A + \frac{2\alpha c A}{\Delta_i^2} \log(1 + \frac{\alpha A}{\Delta_{\min}}).$$

which implies that $k$ satisfying the condition of applying Lemma 4.3 with $\lambda = 2A$, and we can conclude that:

$$\Pr(N_k(j) \ge \frac{k}{2A}) \le \frac{2}{k^{2\alpha-1}}.$$

**Case 2:** $\gamma_{k,j} \le 2A$: Note that

$$\frac{4\alpha c}{\Delta_j^2} \log(1 + \frac{\alpha A}{\Delta_{\min}}) = \frac{k}{2\gamma_{k,j}}.$$

Since $2\gamma_{k,j}$ locates in the interval $[1, 4A]$ and:

$$k = \gamma_{k,j} \cdot \frac{8\alpha c}{\Delta_j^2} \log(1 + \frac{\alpha A}{\Delta_{\min}}) \ge 2\gamma_{k,j} + 2\gamma_{k,j} \cdot \frac{\alpha c}{\Delta_j^2} \log(1 + \frac{\alpha A}{\Delta_{\min}})$$

which satisfies the condition of applying Lem. 4.3 with $\lambda = 2\gamma_{k,j}$. Therefore, we have:

$$\Pr(N_k(j) \ge \frac{4\alpha c}{\Delta_j^2} \log(1 + \frac{\alpha A}{\Delta_{\min}})) = \Pr(N_k(j) \ge \frac{k}{2\gamma_{k,j}}) \le \frac{2}{k^{2\alpha-1}} \tag{14}$$

Combining the above two cases, we can conclude that, for arbitrary $j \in G_i^{\text{lower}}$,

$$\Pr(N_j(k) \ge \frac{k}{2A} + \frac{4\alpha c}{\Delta_j^2} \log(1 + \frac{\alpha A}{\Delta_{\min}})) \le \min\{\Pr(N_j(k) \ge \frac{k}{2A}), \Pr(N_j(k) \ge \frac{4\alpha c}{\Delta_j^2} \log(1 + \frac{\alpha A}{\Delta_{\min}}))\} \le \frac{2}{k^{2\alpha-1}}$$

which reflects that with high probability, $\sum_{j \in G_i^{\text{lower}}} N_j(k)$ is small:

$$\Pr(\sum_{j \in G_i^{\text{lower}}} N_j(k) \geq \frac{k}{2} + \sum_{j \in G_i^{\text{lower}}} \frac{4\alpha c}{\Delta_j^2} \log(1 + \frac{\alpha A}{\Delta_{\min}}))$$

$$\leq \sum_{j \in G_i^{\text{lower}}} \Pr(N_j(k) \geq \frac{k}{2|G_i^{\text{lower}}|} + \frac{4\alpha c}{\Delta_j^2} \log(1 + \frac{\alpha A}{\Delta_{\min}}))$$

$$(P(a + b \leq c + d) \leq P(a \leq c) + P(b \leq d))$$

$$\leq \sum_{j \in G_i^{\text{lower}}} \Pr(N_j(k) \geq \frac{k}{2A} + \frac{4\alpha c}{\Delta_j^2} \log(1 + \frac{\alpha A}{\Delta_{\min}})) \qquad (|G_i^{\text{lower}}| \leq A)$$

$$\leq \frac{2|G_i^{\text{lower}}|}{k^{2\alpha-1}} \leq \frac{2A}{k^{2\alpha-1}}$$

Since $\sum_{j \in G_i^{\text{upper}}} N_j(k) + \sum_{j \in G_i^{\text{lower}}} N_j(k) = k$, and note that,

$$k - (\frac{k}{2} + \sum_{j \in G_i^{\text{lower}}} \frac{4\alpha c}{\Delta_j^2} \log(1 + \frac{\alpha A}{\Delta_{\min}})) = \frac{k}{2} - \sum_{j \in G_i^{\text{lower}}} \frac{4\alpha c}{\Delta_j^2} \log(1 + \frac{\alpha A}{\Delta_{\min}}) = \frac{16\alpha c |G_i^{\text{upper}}|}{\Delta_i^2} \log(1 + \frac{\alpha A}{\Delta_{\min}})$$

we have:

$$\Pr(\sum_{j \in G_i^{\text{upper}}} N_j(k) \leq \frac{16\alpha c |G_i^{\text{upper}}|}{\Delta_i^2} \log(1 + \frac{\alpha A}{\Delta_{\min}}))$$

$$= \Pr(\sum_{j \in G_i^{\text{lower}}} N_j(k) \geq \frac{k}{2} + \sum_{j \in G_i^{\text{lower}}} \frac{4\alpha c}{\Delta_j^2} \log(1 + \frac{\alpha A}{\Delta_{\min}}))$$

$$\leq \frac{2A}{k^{2\alpha-1}}$$

Therefore, w.p. $1 - \frac{2A}{k^{2\alpha-1}}$, there exists $j \in G_i^{\text{upper}}$, such that

$$N_j(k) \geq \frac{1}{|G_i^{\text{upper}}|} \sum_{j \in G_i^{\text{upper}}} N_j(k) \geq \frac{16\alpha c}{\Delta_i^2} \log(1 + \frac{\alpha A}{\Delta_{\min}}).$$

Recall our choice of $c$ (Eq.(11)), the above implies that:

$$N_j(k) \geq \frac{32\alpha \log f(k)}{\Delta_i^2}. \tag{15}$$

Therefore,

$$\Pr(\{i = \pi_k^E\} \cap \{k \geq k_i\}) \leq \Pr(\{i = \pi_k^E\} \cap \{k \geq k_i\} \cap \{\exists j \in G_i^{\text{upper}} : N_j(k) \geq \frac{32\alpha \log f(k)}{\Delta_i^2}\})$$

$$+ \Pr(\{k \geq k_i\} \cap \neg\{\exists j \in G_i^{\text{upper}} : N_j(k) < \frac{32\alpha \log f(k)}{\Delta_i^2}\})$$

$$\leq \Pr(\{i = \pi_k^E\} \cap \{k \geq k_i\} \cap \{\exists j \in G_i^{\text{upper}} : N_j(k) \geq \frac{32\alpha \log f(k)}{\Delta_i^2}\}) + \frac{2A}{k^{2\alpha-1}}$$

$$\text{(Eq.(15))}$$

$$\leq \Pr(\{i = \pi_k^E\} \cap \{k \geq k_i\} \cap \{\exists j \in G_i^{\text{upper}} : N_j(k) \geq \frac{8\alpha \log f(k)}{(\Delta_j - \Delta_j)^2}\}) + \frac{2A}{k^{2\alpha-1}}$$

$$\leq \frac{2}{k^{2\alpha}} + \frac{2A}{k^{2\alpha-1}} \qquad \text{(Lem. 4.2)}$$

$\square$

**Lemma D.2** (Integral Lemma). *For arbitrary $k_0 \geq 1$ and $\beta > 1$, we have:*

$$\sum_{k=k_0+1}^{\infty} \frac{1}{k^\beta} \leq \int_{k_0}^{\infty} \frac{1}{x^\beta} dx \leq \frac{1}{(\beta-1)k_0^{\beta-1}}$$

**Theorem 4.1.** *[Exploitation Regret] In Algorithm 2, by choosing arbitrary $\alpha > 1$, there exists an absolute constant c, such that, for arbitrary $K \geq 1$, the pseduo-regret of $Alg^E$ is upper bounded by:*
$Regret_K(Alg^E) \leq \widetilde{O}\left(\frac{A}{\alpha-1} + \alpha \sum_{\Delta_i > 0}(A - i)\left(\frac{1}{\Delta_i} - \frac{\Delta_i}{\Delta_{i-1}^2}\right)\right)$ *where $\Delta_0 := \infty$ so $\frac{\Delta_1}{\Delta_0^2} = 0$.*

*Proof.* Recall the definition of $k_i$ in Eq.(13) in Lem. D.1 above. For $i \geq 2$, if $\Delta_i \neq \Delta_{i-1}$, we have:

$$
\begin{aligned}
k_i - k_{i-1} =& \alpha c\Big(\sum_{\iota \in G_i^{\text{lower}}} \frac{8}{\Delta_\iota^2} - \sum_{\iota \in G_{i-1}^{\text{lower}}} \frac{8}{\Delta_\iota^2} + \frac{32|G_i^{\text{upper}}|}{\Delta_i^2} - \frac{32|G_{i-1}^{\text{upper}}|}{\Delta_{i-1}^2}\Big) \log(1 + \frac{\alpha A}{\Delta_{\min}}) \\
\leq& \alpha c\Big((|G_i^{\text{lower}}| - |G_{i-1}^{\text{lower}}|)\frac{32}{\Delta_i^2} + \frac{32|G_i^{\text{upper}}|}{\Delta_i^2} - \frac{32|G_{i-1}^{\text{upper}}|}{\Delta_{i-1}^2}\Big) \log(1 + \frac{\alpha A}{\Delta_{\min}}) \\
& \qquad\qquad\qquad (\forall \iota \in G_i^{\text{lower}} \setminus G_{i-1}^{\text{lower}}, \text{ we have } 1/\Delta_\iota^2 \leq 4/\Delta_i^2) \\
\leq& \alpha c\Big((|G_{i-1}^{\text{upper}}| - |G_i^{\text{upper}}|)\frac{32}{\Delta_i^2} + \frac{32|G_i^{\text{upper}}|}{\Delta_i^2} - \frac{32|G_{i-1}^{\text{upper}}|}{\Delta_{i-1}^2}\Big) \log(1 + \frac{\alpha A}{\Delta_{\min}}) \\
& \qquad\qquad\qquad (|G_{i-1}^{\text{upper}}| + |G_{i-1}^{\text{lower}}| = |G_i^{\text{upper}}| + |G_i^{\text{lower}}| = A) \\
\leq& 32\alpha c|G_{i-1}^{\text{upper}}|\Big(\frac{1}{\Delta_i^2} - \frac{1}{\Delta_{i-1}^2}\Big) \log(1 + \frac{\alpha A}{\Delta_{\min}}).
\end{aligned}
$$

and if $\Delta_i = \Delta_{i-1}$, we also have:

$$
k_i - k_{i-1} = 0 \leq 32\alpha c|G_{i-1}^{\text{upper}}|\Big(\frac{1}{\Delta_i^2} - \frac{1}{\Delta_{i-1}^2}\Big) \log(1 + \frac{\alpha A}{\Delta_{\min}}).
$$

Moreover, for $i = 1$, with the extended definition that $\Delta_0 = \infty$ (so that $1/\Delta_0^2 = 0$) and $|G_0^{\text{upper}}| = A$, we also have:

$$
\begin{aligned}
k_1 :=& 8\alpha c\Big(\sum_{\iota \in G_1^{\text{lower}}} \frac{1}{\Delta_\iota^2} + \frac{4|G_1^{\text{upper}}|}{\Delta_1^2}\Big) \log(1 + \frac{\alpha A}{\Delta_{\min}}) \\
\leq& 8\alpha c\Big(\sum_{\iota \in G_1^{\text{lower}}} \frac{4}{\Delta_1^2} + \frac{4|G_1^{\text{upper}}|}{\Delta_1^2}\Big) \log(1 + \frac{\alpha A}{\Delta_{\min}}) \\
=& \frac{32\alpha c A}{\Delta_1^2} \log(1 + \frac{\alpha A}{\Delta_{\min}}) \\
=& 32\alpha c|G_0^{\text{upper}}|(\frac{1}{\Delta_1^2} - \frac{1}{\Delta_0^2}) \log(1 + \frac{\alpha A}{\Delta_{\min}}).
\end{aligned}
$$

Therefore, we have (we denote $k_A := \infty$ and $k_0 := 0$):

$$
\begin{aligned}
\lim_{K \to \infty} Regret_K(Alg^E) =& \sum_{k=1}^{\infty} \sum_{j:\Delta_j > 0} \Pr(j = \pi_k^E)\Delta_j \\
=& \sum_{i=1}^{A} \sum_{k=k_{i-1}+1}^{k_i} \sum_{\Delta_j > 0} \Pr(j = \pi_k^E)\Delta_j \\
=& \sum_{i=1}^{A} \sum_{k=k_{i-1}+1}^{k_i} \Big(\sum_{\Delta_j \geq \Delta_{i-1}} \Pr(j = \pi_k^E)\Delta_j + \sum_{\Delta_j < \Delta_{i-1}} \Pr(j = \pi_k^E)\Delta_j\Big) \\
\leq& \sum_{i=1}^{A} \sum_{k=k_{i-1}+1}^{k_i} \Big(\sum_{\Delta_j \geq \Delta_{i-1}} (\frac{2}{k^{2\alpha}} + \frac{2A}{k^{2\alpha-1}})\Delta_j + \sum_{\Delta_j < \Delta_{i-1}} \Pr(j = \pi_k^E)\Delta_j\Big) \\
& \hfill \text{(Lemma D.1)} \\
\leq& \sum_{i=1}^{A} \sum_{k=k_{i-1}+1}^{k_i} \Big(\sum_{\Delta_j \geq \Delta_{i-1}} (\frac{2}{k^{2\alpha}} + \frac{2A}{k^{2\alpha-1}})\Delta_j + \sum_{\Delta_j \leq \Delta_i} \Pr(j = \pi_k^E)\Delta_j\Big) \\
& \hfill (\Delta_i \leq \Delta_{i-1})
\end{aligned}
$$

$$\leq \sum_{i=1}^{A}\sum_{k=k_i+1}^{\infty}\Big(\frac{2}{k^{2\alpha}}+\frac{2A}{k^{2\alpha-1}}\Big)+\sum_{i=1}^{A}\sum_{k=k_{i-1}+1}^{k_i}\sum_{\Delta_j\leq\Delta_i}\Pr(j=\pi_k^{\mathrm{E}})\Delta_j$$

$$\leq \sum_{i=1}^{A}\sum_{k=k_i+1}^{\infty}\frac{2(A+1)}{k^{2\alpha-1}}+\sum_{i=1}^{A}\sum_{k=k_{i-1}+1}^{k_i}\sum_{\Delta_j\leq\Delta_i}\Delta_i$$

(Second term is maximized when $\Pr(i=\pi_k^{\mathrm{E}})=1$.)

$$\leq \widetilde{O}(\frac{A}{\alpha-1})+\sum_{i=1}^{A}\Delta_i\cdot(k_i-k_{i-1})$$

(First term: Lemma D.2 and some simplification; Second term: Definition of $k_i$.)

$$=\widetilde{O}(\frac{A}{\alpha-1})+\sum_{i:\Delta_i>0}\Delta_i\cdot(k_i-k_{i-1})$$

$$\leq \widetilde{O}(\frac{A}{\alpha-1})+\sum_{\Delta_i>0}32\alpha c|G_{i-1}^{\mathrm{upper}}|\Big(\frac{1}{\Delta_i}-\frac{\Delta_i}{\Delta_{i-1}^2}\Big)\log(1+\frac{\alpha A}{\Delta_{\min}})$$

$$\approx \widetilde{O}\left(\frac{A}{\alpha-1}+\sum_{\Delta_i>0}\alpha|G_{i-1}^{\mathrm{upper}}|\Big(\frac{1}{\Delta_i}-\frac{\Delta_i}{\Delta_{i-1}^2}\Big)\right)$$

According to the definition, we always have $|G_{i-1}^{\mathrm{upper}}|\leq A-i+1$, therefore,

$$\lim_{K\to\infty}\mathrm{Regret}_K(\mathrm{Alg}^{\mathrm{E}})=\widetilde{O}\left(\frac{A}{\alpha-1}+\alpha\sum_{\Delta_i>0}(A-i)\Big(\frac{1}{\Delta_i}-\frac{\Delta_i}{\Delta_{i-1}^2}\Big)\right)$$

$\square$

# E   Behavior Analysis of Optimistic Algorithm

**Definition E.1** (Definition of Events).

$$\mathcal{E}_{k,h,\pi}:=\{\pi_{k,h}(s_h)\neq\pi_h(s_h)\},\quad \widetilde{\mathcal{E}}_{k,h,\pi}:=\mathcal{E}_{k,h,\pi}\cap\bigcap_{h'=1}^{h}\mathcal{E}_{k,h'-1,\pi}^{\complement},\quad \bar{\mathcal{E}}_{k,\pi}:=\bigcup_{h=1}^{H}\mathcal{E}_{k,h,\pi},$$

$$\mathcal{E}_{k,h}:=\{\pi_{k,h}(s_h)\notin\Pi_h^*(s_h)\},\quad \widetilde{\mathcal{E}}_{k,h,}:=\mathcal{E}_{k,h}\cap\bigcap_{h'=1}^{h}\mathcal{E}_{k,h'-1}^{\complement},\quad \bar{\mathcal{E}}_k:=\bigcup_{h=1}^{H}\mathcal{E}_{k,h}$$

In another word, $\mathcal{E}_{k,h,\pi}$ means $\pi_k$ disagrees with $\pi$ at state $s_h$ which occurs at step $h$, $\widetilde{\mathcal{E}}_{k,h,\pi}$ means the first disagreement between $\pi_k$ and $\pi$ occurs at step $h$, and $\bar{\mathcal{E}}_{k,\pi}$ denotes the event that there exists one state $s_h$ at some time step $h\in[H]$ such that $\pi_k$ agrees with $\pi_k$ at $s_h$.

Besides, $\mathcal{E}_{k,h}$ denotes the events that $\pi_{k,h}(s_h)$ will not be taken by any optimal policy. Note that here we use $\Pi_h^*(s_h)$ to denote the set of all possible optimal actions at state $s_h$. Given a deterministic optimal policy $\pi^*$, in general $\mathcal{E}_{k,h}\neq\mathcal{E}_{k,h,\pi^*}$ when there are multiple optimal actions at one state.

**Lemma E.2.** *For arbitrary reward function $R$, given a fixed deterministic policy $\pi$, we have:*

$$V_1^\pi(s_1)-V_1^{\pi_k}(s_1)=\mathbb{E}_{\pi_k}[\sum_{h=1}^{H}\mathbb{I}[\widetilde{\mathcal{E}}_{k,h,\pi}](V_h^\pi(s_h)-V_h^{\pi_k}(s_h))]$$

*Proof.*

$$V_1^\pi(s_1)-V_1^{\pi_k}(s_1)=\mathbb{I}[\mathcal{E}_{k,1,\pi}^{\complement}]\Big(Q_1^\pi(s_1,\pi_k)-Q_1^{\pi_k}(s_1,\pi_k)\Big)+\mathbb{I}[\mathcal{E}_{k,1,\pi}]\Big(V_1^\pi(s_1)-V_1^{\pi_k}(s_1)\Big)$$

$$=\mathbb{E}_{\pi_k}[\mathbb{I}[\mathcal{E}_{k,1,\pi}^{\complement}]\Big(V_2^\pi(s_2)-V_2^{\pi_k}(s_2)\Big)]+\mathbb{I}[\widetilde{\mathcal{E}}_{k,1,\pi}]\Big(V_1^\pi(s_1)-V_1^{\pi_k}(s_1)\Big)$$

($\widetilde{\mathcal{E}}_{k,1,\pi}=\mathcal{E}_{k,1,\pi}$ by definition)

$$=\mathbb{E}_{\pi_k}[\mathbb{I}[\mathcal{E}_{k,2,\pi}^{\complement} \cap \mathcal{E}_{k,1,\pi}^{\complement}]\Big(Q_2^\pi(s_2, \pi_k) - Q_2^{\pi_k}(s_2, \pi_k)\Big)]$$

$$+ \mathbb{E}_{\pi_k}[\mathbb{I}[\mathcal{E}_{k,2,\pi} \cap \mathcal{E}_{k,1,\pi}^{\complement}]\Big(V_2^\pi(s_2) - V_2^{\pi_k}(s_2)\Big)] + \mathbb{I}[\widetilde{\mathcal{E}}_{k,1,\pi}]\Big(V_1^\pi(s_1) - V_1^{\pi_k}(s_1)\Big)$$

$$=\mathbb{E}_{\pi_k}[\mathbb{I}[\mathcal{E}_{k,2,\pi}^{\complement} \cap \mathcal{E}_{k,1,\pi}^{\complement}]\Big(Q_2^\pi(s_2, \pi_k) - Q_2^{\pi_k}(s_2, \pi_k)\Big)]$$

$$+ \mathbb{E}_{\pi_k}[\mathbb{I}[\widetilde{\mathcal{E}}_{k,2,\pi}]\Big(V_2^\pi(s_2) - V_2^{\pi_k}(s_2)\Big)] + \mathbb{I}[\widetilde{\mathcal{E}}_{k,1,\pi}]\Big(V_1^\pi(s_1) - V_1^{\pi_k}(s_1)\Big)$$

$$=...$$

$$=\mathbb{E}_{\pi_k}[\sum_{h=1}^{H} \mathbb{I}[\widetilde{\mathcal{E}}_{k,h,\pi}](V_h^\pi(s_h) - V_h^{\pi_k}(s_h))]$$

$\square$

**Lemma E.3** (Relationship between Density Difference and Policy Disagreement Probability).

$$d^{\pi_k}(s_h, a_h) \geq d^\pi(s_h, a_h) - \min\{\Pr(\bar{\mathcal{E}}_{k,\pi}|\pi_k), \ d^\pi(s_h, a_h)\}, \quad \forall s_h \in \mathcal{S}_h, a_h \in \mathcal{A}_h, h \in [H]$$

*where we use* $\Pr(\bar{\mathcal{E}}_{k,\pi}|\pi_k)$ *as a short note of* $\mathbb{E}_{s_1,a_1,s_2,a_2...,s_H,a_H \sim \pi_k}[\bar{\mathcal{E}}_{k,\pi}].$

*Proof.* By applying Lemma E.2 with $\delta_{s_h,a_h} := \mathbb{I}[S_h = s_h, A_h = a_h]$ as reward function, we have:

$$d^\pi(s_h, a_h) - d^{\pi_k}(s_h, a_h) = V_1^\pi(s_1; \delta_{s_h,a_h}) - V_1^{\pi_k}(s_1; \delta_{s_h,a_h})$$

$$=\mathbb{E}_{\pi_k}[\sum_{h'=1}^{h} \mathbb{I}[\widetilde{\mathcal{E}}_{k,h',\pi}](V_{h'}^\pi(s_{h'}; \delta_{s_h,a_h}) - V_{h'}^{\pi_k}(s_{h'}; \delta_{s_h,a_h}))]$$

$$(V_{h'}^\pi = V_{h'}^{\pi_k} = 0 \text{ for all } h' \geq h+1)$$

$$\leq \mathbb{E}_{\pi_k}[\sum_{h'=1}^{h} \mathbb{I}[\widetilde{\mathcal{E}}_{k,h',\pi}]V_{h'}^\pi(s_{h'}; \delta_{s_h,a_h})] \qquad (V_{h'}^{\pi_k}(s_h'; \delta_{s_h,a_h}) \geq 0)$$

$$\leq \mathbb{E}_{\pi_k}[\sum_{h'=1}^{h} \mathbb{I}[\widetilde{\mathcal{E}}_{k,h',\pi}]] \qquad (V_{h'}^\pi(s_h'; \delta_{s_h,a_h}) \leq 1)$$

$$\leq \mathbb{E}_{s_1,a_1,s_2,a_2...,s_H,a_H \sim \pi_k}[\bar{\mathcal{E}}_{k,\pi}] = \Pr(\bar{\mathcal{E}}_{k,\pi}|\pi_k)$$

which implies that,

$$d^{\pi_k}(s_h, a_h) \geq d^\pi(s_h, a_h) - \Pr(\bar{\mathcal{E}}_{k,\pi}|\pi_k)$$

Combining with $d^{\pi_k} \geq 0$, we finish the proof. $\square$

**Definition E.4** (Conversion to Optimal Deterministic Policy). *Given arbitrary deterministic policy* $\pi = \{\pi_1, ..., \pi_H\}$, *we use* $\Pi^* \circ \pi = \{\pi_1^*, ..., \pi_H^*\}$ *to denote an optimal deterministic policy, such that:*

$$\pi_h^*(s_h) = \begin{cases} \pi_h(s_h), & \text{if } \pi_h(s_h) \in \Pi_h^*(s_h); \\ Select(\Pi_h^*(s_h)), & \text{otherwise.} \end{cases}$$

*where Select is a function which returns the first optimal action from* $\Pi_h^*(s_h)$.

In another word, $\Pi^* \circ \pi$ agrees with $\pi$ if $\pi_h(s_h)$ is one of the optimal action at state $s_h$. Otherwise, $\Pi^* \circ \pi$ takes one of a fixed optimal action from $\Pi_h^*(s_h)$. In order to make sure $\Pi^* \circ$ is a deterministic mapping, we assume function $Select$ only choose the first optimal action in $\Pi^*(s_h)$ (ordered by index of action).

**Theorem 4.7.** *For an arbitrary sequence of deterministic policies* $\pi_1, \pi_2, ..., \pi_k$, *there must exist a sequence of deterministic optimal policies* $\pi_1^*, \pi_2^*, ..., \pi_k^*$, *such that* $\forall h \in [H], s_h \in \mathcal{S}_h, a_h \in \mathcal{A}_h$:

$$\sum_{\widetilde{k}=1}^{k} d^{\pi_{\widetilde{k}}}(s_h, a_h) \geq \sum_{\widetilde{k}=1}^{k} d^{\pi_{\widetilde{k}}^*}(s_h, a_h) - \frac{1}{\Delta_{\min}}\Big(\sum_{\widetilde{k}=1}^{k} V_1^*(s_1) - V_1^{\pi_{\widetilde{k}}}(s_1)\Big).$$

*Proof.* For each $\pi_k$, we construct an optimal deterministic policy $\pi_k^* := \Pi^* \circ \pi_k$, where $\Pi^* \circ$ is defined in Def. E.4. By applying Lemma E.2 with the reward function in MDP, and $\pi = \pi_k^*$, we have:

$$V_1^{\pi_k^*}(s_1) - V_1^{\pi_k}(s_1) = \mathbb{E}_{\pi_k}\Big[\sum_{h=1}^{H} \mathbb{I}[\widetilde{\mathcal{E}}_{k,h,\pi_k^*}](V_h^{\pi_k^*}(s_h) - V_h^{\pi_k}(s_h))\Big]$$

$$\geq \mathbb{E}_{\pi_k}\Big[\sum_{h=1}^{H} \mathbb{I}[\widetilde{\mathcal{E}}_{k,h,\pi_k^*}](V_h^{\pi_k^*}(s_h) - Q_h^{\pi_k^*}(s_h, \pi_k(s_h)))\Big]$$

$$\geq \mathbb{E}_{\pi_k}\Big[\sum_{h=1}^{H} \mathbb{I}[\widetilde{\mathcal{E}}_{k,h,\pi_k^*}]\Delta_{\min}\Big] = \Delta_{\min} \Pr(\bar{\mathcal{E}}_{k,\pi_k^*}|\pi_k)$$

Therefore, we have:

$$\Pr(\bar{\mathcal{E}}_{k,\pi_k^*}|\pi_k) \leq \frac{1}{\Delta_{\min}}(V_1^{\pi_k^*}(s_1) - V_1^{\pi_k}(s_1))$$

By applying Lemma E.3, we have:

$$d^{\pi_k}(s_h, a_h) \geq d^{\pi_k^*}(s_h, a_h) - \frac{1}{\Delta_{\min}}\Big(V_1^*(s_1) - V_1^{\pi_k}(s_1)\Big), \quad \forall s_h \in \mathcal{S}_h, a_h \in \mathcal{A}_h, h \in [H]$$

After the same discussion for all $k \in [K]$, and the above inequality of each $k$ together, we have:

$$\sum_{k=1}^{K} d^{\pi_k}(s_h, a_h) \geq \sum_{k=1}^{K} d^{\pi_k^*}(s_h, a_h) - \frac{1}{\Delta_{\min}}\Big(\sum_{k=1}^{K} V_1^*(s_1) - V_1^{\pi_k}(s_1)\Big)$$

$\square$

**Corollary E.5** (Unique Optimal Policy). *When $|\Pi^*| = 1$, Thm. 4.7 implies that:*

$$\sum_{k=1}^{K} d^{\pi_k}(s_h, a_h) \geq K d^{\pi^*}(s_h, a_h) - \frac{1}{\Delta_{\min}}\Big(\sum_{k=1}^{K} V_1^*(s_1) - V_1^{\pi_k}(s_1)\Big)$$

**Theorem 4.8.** *[The existance of well-covered optimal policy] Given an arbitrary tabular MDP, and an arbitrary sequence of deterministic optimal policies $\pi_1^*, \pi_2^*, ...\pi_k^*$ ($\pi_i^*$ may not equal to $\pi_j^*$ for arbitrary $1 \leq i < j \leq k$ when there are multiple deterministic optimal policies), there exists a (possibly stochastic) policy $\pi_{cover}^*$ such that $\forall h \in [H], \forall (s_h, a_h) \in \mathcal{S}_h \times \mathcal{A}_h$ with $d^{\pi_{cover}^*}(s_h, a_h) > 0$:*

$$\sum_{\widetilde{k}=1}^{k} d^{\pi_{\widetilde{k}}^*}(s_h, a_h) \geq \frac{k}{2} \cdot \widetilde{d}^{\pi_{cover}^*}(s_h, a_h), \text{ with } \widetilde{d}^{\pi_{cover}^*}(\cdot, \cdot) := \max\left\{\frac{d_{h,\min}^*(\cdot, \cdot)}{(|\mathcal{Z}_{h,div}^*| + 1)H}, d^{\pi_{cover}^*}(\cdot, \cdot)\right\}.$$

*where $\mathcal{Z}_{h,div}^* := \{(s_h, a_h) \in \mathcal{S}_h \times \mathcal{A}_h | \exists \pi^*, \widetilde{\pi}^* \in \Pi^*, \text{ s.t. } d^{\pi^*}(s_h) > 0, d^{\widetilde{\pi}^*}(s_h) = 0\}$, and $d_{h,\min}^*(s_h, a_h) := \min_{\pi^* \in \Pi^*} d^{\pi^*}(s_h, a_h)$ subject to $d^{\pi^*}(s_h, a_h) > 0$.*

*Proof.* For arbitrary $h \in [H]$, we define:

$$N_{I_K^*}(s_h, a_h) := \sum_{k=1}^{K} \mathbb{I}[d^{\pi_{\widetilde{k}}^*}(s_h, a_h) > 0]$$

In another word, $N_{I_K^*}(\cdot, \cdot)$ denotes the number of optimal policies in the sequence, which can hit state $s_h$ and take action $a_h$.

Next, we define $\mathcal{Z}_h^*$, $\mathcal{Z}_h^{\text{insuff}}$ and $\Pi_{h,\text{insuff}}$ as

$$\mathcal{Z}_h^* := \{(s_h, a_h) \in \mathcal{S}_h, \mathcal{A}_h | \exists \pi^* \in \Pi^*, \text{ s.t. } d^{\pi^*}(s_h, a_h) > 0\}$$

$$\mathcal{Z}_h^{\text{insuff}} := \{(s_h, a_h) | (s_h, a_h) \in \mathcal{Z}_h^* : N_{I_K^*}(s_h, a_h) < \frac{K}{2(|\mathcal{Z}_{h,\text{div}}^*| + 1)H}\}$$

$$I_h^{\text{insuff}} := \{k \in [K] : \exists (s_h, a_h) \in \mathcal{Z}_h^{\text{insuff}}, \text{ s.t. } d^{\pi_k}(s_h, a_h) > 0\}$$

In a word, $\mathcal{Z}_h^*$ is the collection of states actions reachable by at least one optimal policy, $\mathcal{Z}_h^{\text{insuff}}$ is a collection of "insufficiently hitted" states actions at step $h$, which are only covered by a small portion of optimal policies in the sequence, and $I_h^{\text{insuff}}$ is a collection of the index of the optimal policies in the sequence, which cover at least one state action pair in $\mathcal{Z}_h^{\text{insuff}}$.

Note that we must have $\mathcal{Z}_h^{\text{insuff}} \subset \mathcal{Z}_{h,\text{div}})$, because if one state action pair $s_h, a_h$ is reachable by arbitrary deterministic policy, then $N_{I_k^*}(s_h, a_h) = K$. Then, we have:

$$|I_h^{\text{insuff}}| < |\mathcal{Z}_h^{\text{insuff}}| \cdot \frac{K}{2(|\mathcal{Z}_{h,\text{div}}| + 1)H} \leq |\mathcal{Z}_{h,\text{div}}| \cdot \frac{K}{2(|\mathcal{Z}_{h,\text{div}}| + 1)H} \leq \frac{K}{2H}$$

We define $I_{1:H}^{\text{suff}} := I_K \setminus \bigcup_{h=1}^{H} I_h^{\text{insuff}}$. Intuitively, $I_{1:H}^{\text{suff}}$ is the set including the indices of optimal policies in the sequence only hitting those states which are covered by most of the other optimal policies. In fact, $I_{1:H}^{\text{suff}}$ is non-empty since:

$$|I_{1:H}^{\text{suff}}| \geq K - \frac{K}{2H} \cdot H = \frac{K}{2}$$

We use $\pi_{I_{1:H}^{\text{suff}}}^*$ to denote the average mixture policy over $\{\pi_i^* : i \in I_{1:H}^{\text{suff}}\}$, a direct result is that:

$$\sum_{k=1}^{K} d^{\pi_k^*}(s_h, a_h) \geq \sum_{k \in I_{1:H}^{\text{suff}}} d^{\pi_k^*}(s_h, a_h) = |I_{1:H}^{\text{suff}}| \cdot d^{\pi_{I_{1:H}^{\text{suff}}}^*} \geq \frac{K}{2} \cdot d^{\pi_{I_{1:H}^{\text{suff}}}^*}$$

On the other hand, for all $s_h, a_h$ such that $d^{\pi_{I_{1:H}^{\text{suff}}}^*}(s_h, a_h) > 0$, we must have $(s_h, a_h) \notin \mathcal{Z}_h^{\text{insuff}}$, and therefore:

$$\sum_{k=1}^{K} d^{\pi_k^*}(s_h, a_h) \geq \frac{K}{2(|\mathcal{Z}_{h,\text{div}}| + 1)H} d_{h,\min}^*(s_h, a_h)$$

Combining the above two inequalities, we finish the proof. $\qquad\square$

# F  Analysis of Pessimistic Value Iteration

In this section, we provide analysis for Alg. 3. Our analyses base on an extension of the Clipping Trick in [Simchowitz and Jamieson, 2019] into our setting.

## F.1  Underestimation and Some Concrete Choices of Bonus Term

**Lemma F.1** (Underestimation). *Given a **Bonus** satisfying Cond. 4.4, for arbitrary dataset $D_k$ consisting of $k$ trajectories by a sequence of policies $\pi_1, ..., \pi_k$, by running Alg 3 with $D_k$ and the bonus term $b(\cdot, \cdot)$ returned by **Bonus**$(D_k, \delta_k)$, on the events $\mathcal{E}_{\textbf{Bonus}}$ defined in Cond. 4.4:*

$$\forall h \in [H], \forall s_h \in \mathcal{S}_h, a_h \in \mathcal{A}_h, \quad \widehat{Q}_h(s_h, a_h) \leq Q^{\pi_{\widehat{Q}, h}}(s_h, a_h) \leq Q^*(s_h, a_h) \qquad (16)$$

*where we use $\pi_{\widehat{Q}} = \{\pi_{\widehat{Q}, 1}, ..., \pi_{\widehat{Q}, H}\}$ to denote the greedy policy w.r.t. $\widehat{Q}$.*

*Proof.* We only prove the first inequality holds, since the second one holds directly because of the definition of optimal policy.

First of all, $V_{H+1} = 0 \leq V_{H+1}^{\pi_{\widehat{Q}}}$ holds directly, which implies that Eq.(16) holds at step $h = H$ as a result of the deterministic reward function.

Now, we conduct the induction. Suppose Eq.(16) already holds for $h + 1$, which implies that:

$$\widehat{V}_{h+1}(s_{h+1}) = \widehat{Q}_{h+1}(s_{h+1}, \pi_{\widehat{Q}, h+1}(s_{h+1})) \leq Q_{h+1}^{\pi_{\widehat{Q}}}(s_{h+1}, \pi_{\widehat{Q}, h+1}(s_{h+1})) = V_{h+1}^{\pi_{\widehat{Q}}}(s_{h+1}) \quad (17)$$

then, at step $h$, we have:

$$\begin{aligned}
Q_h(s_h, a_h) - Q^{\pi_{\widehat{Q}}}(s_h, a_h) &= \widehat{P}_h \widehat{V}_{h+1}(s_h, a_h) - b_h(s_h, a_h) - P_h V_{h+1}^\pi(s_h, a_h) \\
&= \underbrace{(\widehat{P}_h - P_h)\widehat{V}_{h+1}(s_h, a_h) - b_h(s_h, a_h)}_{\text{part 1}} + \underbrace{P_h(\widehat{V}_{h+1} - V_{h+1}^{\pi_{\widehat{Q}}})(s_h, a_h)}_{\text{part 2}}
\end{aligned}$$

As we can see, part 1 is non-positive with probability $1 - \delta$ as a result of Cond. 4.4, while part 2 is also less than or equal to zero because of the induction condition in Eq.(17). $\qquad\square$

**Choice 1: Naive Bound** According to Hoeffding inequality, with probability $1 - \delta/(SAH)$, we have the following holds for each $s_h, a_h, h$

$$|\widehat{P}_h V_{h+1} - P_h V_{h+1}| \leq \|\widehat{P}_h - P_h\|_1 \|V_{h+1}\|_\infty \leq \|\widehat{P}_h - P_h\|_1 H \leq c_1 HS\sqrt{\frac{\log(SAH/\delta)}{N(s_h, a_h)}}$$

which implies that condition 4.4 holds with probability $1 - \delta$ as long as:

$$b_h(N, \delta) := HS\sqrt{\frac{\log(SAH/\delta)}{N}}$$

**Choice 2: Adaptive Bonus Term based on the "Bernstein Trick"** One can also consider an analogue of the bonus term functions in Alg. 3 of [Simchowitz and Jamieson, 2019], which is originally designed for optimistic algorithms. We omit the discussions here.

## F.2 Definition of "Surplus" in Pessimistic Algorithms and the Clipping Trick

We consider the pessimistic algorithm, and denote the estimation of value function as $\widehat{Q}, \widehat{V}$. We assume they are pessimistic estimation, i.e.:

$$V_h^*(s_h) = Q_h^*(s_h, \pi^*) \geq Q_h^*(s_h, \pi_{\widehat{Q}}) \geq V_h^{\pi_k}(s_h) \geq \widehat{Q}_h(s_h, \pi_{\widehat{Q}}) \geq \widehat{Q}_h(s_h, \pi^*).$$

**Definition F.2** (Definition of Surplus in Pessimistic Algorithm setting)**.** *We define the surplus in Pessimistic Algorithm setting:*

$$\mathbf{E}_{k,h}(s_h, a_h) = r(s_h, a_h) + \mathbb{P}_h\widehat{V}_{k,h+1}(s_h, a_h) - \widehat{Q}_{k,h}(s_h, a_h).$$

Because of the underestimation, different from the surplus in overestimation cases [Simchowitz and Jamieson, 2019], here we flip the role between $\widehat{Q}$ and $r + \mathbb{P}\widehat{V}$ to make sure the quantity is non-negative (with high probability).

Based on our definition, we have the following lemma:

**Lemma F.3.** *Under the same condition as Lemma F.1, for arbitrary $h, s_h$, the policy $\pi_k^{PVI}$ returned by Alg.3 satisfying:*

$$V_h^{\pi_k^{PVI}}(s_h) - \widehat{V}_{k,h}(s_h) = \mathbb{E}_{\pi_k^{PVI}}\Big[\sum_{h'=h}^{H} \mathbf{E}_{k,h'}(s_{h'}, a_{h'})|s_h\Big]$$

*Moreover, for arbitrary optimal deterministic or non-deterministic policy $\pi^*$, we have:*

$$V_h^*(s_h) - \widehat{V}_{k,h}(s_h) \leq V_h^*(s_h) - \widehat{Q}_{k,h}(s_h, \pi^*) \leq \mathbb{E}_{\pi^*}\Big[\sum_{h'=h}^{H} \mathbf{E}_{k,h'}(s_{h'}, a_{h'})\Big]$$

*Proof.*

$$V_h^{\pi_k^{PVI}}(s_h) - \widehat{V}_{k,h}(s_h)$$
$$= \mathbb{E}_{a_h \sim \pi_k^{PVI}}[r(s_h, a_h) + \mathbb{P}_h V_{h+1}^{\pi_k^{PVI}}(s_h, a_h) - \widehat{Q}_{k,h}(s_h, a_h) \pm \mathbb{P}_h\widehat{V}_{k,h+1}(s_h, a_h)]$$
$$= \mathbb{E}_{\pi_k^{PVI}}[r(s_h, a_h) + \mathbb{P}_h\widehat{V}_{k,h+1}(s_h, a_h) - \widehat{Q}_{k,h}(s_h, a_h) + \mathbb{P}_h(V_{h+1}^{\pi_k^{PVI}} - \widehat{V}_{k,h+1})(s_h, a_h)]$$
$$= \mathbb{E}_{\pi_k^{PVI}}\Big[\sum_{h'=h}^{H} \mathbf{E}_{k,h'}(s_{h'}, a_{h'})|s_h\Big]$$

Besides, given arbitrary optimal policy $\pi^*$, we have:

$$V_h^*(s_h) - \widehat{V}_{k,h}(s_h)$$
$$= V_h^*(s_h) - \widehat{Q}_{k,h}(s_h, \pi_k^{PVI}) \leq V_h^*(s_h) - \widehat{Q}_{k,h}(s_h, \pi^*) \qquad (\pi_k^{PVI} \text{ is greedy policy w.r.t. } \widehat{Q}_k)$$

$$=\mathbb{E}_{a_h \sim \pi^*}[r(s_h, a_h) + \mathbb{P}_h \widehat{V}_{k,h+1}(s_h, a_h) - \widehat{Q}_{k,h}(s_h, a_h) + \mathbb{P}_h(V^*_{h+1} - \widehat{V}_{k,h+1})(s_h, a_h)]$$

$$\leq \mathbb{E}_{\pi^*}[\sum_{h'=h}^{H} \mathbf{E}_{k,h'}(s_{h'}, a_{h'})]$$

$\square$

**Lemma F.4.** *Under the same condition as Lemma F.1, we have:*

$$\mathbf{E}_{k,h} \leq \min\{H - h + 1, 2B_1\sqrt{\frac{\log(B_2/\delta_k)}{N_{k,h}(s_h, a_h)}}\}.$$

*Proof.*

$$\mathbf{E}_{k,h} := r(s_h, a_h) + \mathbb{P}_h \widehat{V}_{k,h+1}(s_h, a_h) - \widehat{Q}_{k,h}(s_h, a_h)$$

$$= \mathbb{P}_h \widehat{V}_{k,h+1} - \widehat{\mathbb{P}}_{k,h} \widehat{V}_{k,h+1} + b_{k,h}(s_h, a_h) \leq 2b_{k,h}(s_h, a_h) \leq 2B_1\sqrt{\frac{\log(B_2/\delta_k)}{N_{k,h}(s_h, a_h)}}.$$

On the other hand, because the reward function is always locates in $[0, 1]$ and $\widehat{Q}$ is always larger than zero, we have $\mathbf{E}_{k,h}(s_h, a_h) \leq H - h + 1 \leq H$. $\square$

In the following, we define

$$\ddot{\mathbf{E}}_{k,h}(s_h, a_h) := \text{clip}[\mathbf{E}_{k,h}(s_h, a_h)|\varepsilon_{\text{Clip}}].$$

where $\varepsilon_{\text{Clip}} := \frac{\Delta_{\min}}{2H+2}$, and $\text{Clip}[x|\varepsilon] := x \cdot \mathbb{I}[x \geq \varepsilon]$. Then, we recursively define

$$\ddot{Q}^{\pi}_{k,h}(s_h, a_h) = \mathbb{E}_{\pi_h}[r(s_h, a_h) - \ddot{\mathbf{E}}_{k,h}(s_h, a_h) + \mathbb{P}_h \ddot{V}^{\pi}_{k,h+1}(s_h, a_h)|s_h, a_h], \quad \ddot{V}^{\pi}_{k,h}(s_h) := \ddot{Q}^{\pi}_{k,h}(s_h, \pi_h)$$

Note that although different optimal policies $\pi^*$ and $\widetilde{\pi}^*$ have the same optimal value $V^*$, $\ddot{V}^{\pi^*}$ may no longer equal to $\ddot{V}^{\widetilde{\pi}^*}$ because they may have different state occupancy and $\ddot{V}$ depends on $\mathbf{E}$. Therefore, in the following, when we consider the $\ddot{V}$ for optimal policies, we will always specify which optimal policy we are referring to.

**Lemma F.5** (Relationship between $\ddot{V}^{\pi^*}$, $V^{\pi_k^{\text{PVI}}}$ and $\widehat{V}_{k,h}$). *Under the same condition as Lemma F.1, for arbitrary optimal policy $\pi^*$, we have:*

$$\ddot{V}^{\pi^*}_{k,h}(s_h) \leq \widehat{V}_{k,h}(s_h) + (H - h + 1)\varepsilon_{\text{Clip}} \leq V^{\pi_k^{\text{PVI}}}_h(s_h) + (H - h + 1)\varepsilon_{\text{Clip}}$$

*Proof.* Note that:

$$\ddot{\mathbf{E}}_{k,h}(s_h, a_h) \geq \mathbf{E}_{k,h}(s_h, a_h) - \varepsilon_{\text{Clip}}$$

Therefore,

$$V^*_h(s_h) - \ddot{V}^{\pi^*}_h(s_h) = \mathbb{E}_{\pi^*}[\sum_{h=h'}^{H} \ddot{\mathbf{E}}_{k,h'}(s_{h'}, a_{h'})|s_h]$$

$$\geq \mathbb{E}_{\pi^*}[\sum_{h'=h}^{H} \mathbf{E}_{k,h'}(s_{h'}, a_{h'}) - \varepsilon_{\text{Clip}}|s_h]$$

$$\geq V^*_h(s_h) - \min\{\widehat{Q}_{k,h}(s_h, \pi^*), \widehat{V}_{k,h}(s_h)\} - (H - h + 1)\varepsilon_{\text{Clip}} \quad \text{(Lemma F.3)}$$

$$\geq V^*_h(s_h) - \min\{Q^{\pi_k^{\text{PVI}}}_h(s_h, \pi^*), V^{\pi_k^{\text{PVI}}}_h(s_h)\} - (H - h + 1)\varepsilon_{\text{Clip}}$$
$$\text{(Underestimation (Lemma F.1))}$$

Therefore,

$$\ddot{V}^{\pi^*}_{k,h}(s_h) \leq \widehat{V}_{k,h}(s_h) + (H - h + 1)\varepsilon_{\text{Clip}} \leq V^{\pi_k^{\text{PVI}}}_h(s_h) + (H - h + 1)\varepsilon_{\text{Clip}}$$

$\square$

### F.3 Additional Lemma for the Analysis of the Regret of Alg$^E$ when Optimal Deterministic Policies are non-unique

We first introduce a useful Lemma related to the clipping operator from [Simchowitz and Jamieson, 2019]

**Lemma F.6** (Lemma B.3 in [Simchowitz and Jamieson, 2019]). *Let $M \geq 2$, $a_1, \dots a_m \geq 0$ and $\varepsilon \geq 0$. $Clip[\sum_{i=1}^m a_i | \varepsilon] \leq 2 \sum_{i=1}^m Clip[a_i | \frac{\varepsilon}{2m}]$.*

Next, based on definition of $d_{\min}$ in Eq.(10), we have the following Lemma:

**Lemma F.7.** *Given arbitrary deterministic policy $\pi$, if $\pi \notin \Pi^*$, we have:*

$$V_1^*(s_1) - V_1^\pi(s_1) \geq d_{\min}\Delta_{\min}$$

*Proof.* We use $\pi^* := \Pi^* \circ \pi$ to denote the converted deterministic optimal policy, where $\Pi^* \circ$ is defined in Def. E.4. As a direct application of Lemma E.2, we have:

$$V_1^*(s_1) - V_1^\pi(s_1) = V_1^{\pi^*}(s_1) - V_1^\pi(s_1) = \mathbb{E}\left[\sum_{h=1}^H \mathbb{I}[\widetilde{\mathcal{E}}_h](V_h^{\pi^*}(s_h) - V_h^\pi(s_h))\right]$$

$$\geq \Delta_{\min}\mathbb{E}\left[\sum_{h=1}^H \mathbb{I}[\widetilde{\mathcal{E}}_h]\right]$$

$$\geq \Delta_{\min} \Pr(\widetilde{\mathcal{E}}_{h_{init}})$$

$$\geq \Delta_{\min} d_{\min}$$

where we use $\widetilde{\mathcal{E}}_h$ to denote the event that at step $h$, $\pi$ first disagrees with $\pi^*$, or equivalently, $\pi$ first take non-optimal action; in the second inequality, we define $h_{init} := \min_{h \in [H]}, \quad s.t. \quad \Pr(\widetilde{\mathcal{E}}_h) > 0$. Besides, the last inequality is because:

$$\Pr(\widetilde{\mathcal{E}}_{h_{init}}) = \sum_{s_{h_{init}} \in \mathcal{S}_{h_{init}}} \mathbb{I}[\pi_h^*(s_h) \neq \pi_h(s_h)]d^\pi(s_h)$$

$$= \sum_{s_{h_{init}} \in \mathcal{S}_{h_{init}}} \mathbb{I}[\pi_h^*(s_h) \neq \pi_h(s_h)]d^{\pi^*}(s_h)$$

$$\geq \sum_{s_{h_{init}} \in \mathcal{S}_{h_{init}}} \mathbb{I}[\pi_h^*(s_h) \neq \pi_h(s_h)]d_{\min}$$

$$\geq d_{\min}$$

where the last step is because, according to the definition of $h_{init}$, there is at least one $s_h \in \mathcal{S}_h$ such that $\mathbb{I}[\pi_h^*(s_h) \neq \pi_h(s_h)] = 1$. $\qquad\square$

### F.4 Upper Bound for the Regret of Alg$^E$

**Theorem 4.5.** *By running Algorithm 3 with confidence level $\delta_k$, a function **Bonus** satisfying Condition 4.4, and a dataset $D = \{\tau_1, \dots \tau_k\}$ consisting of $k$ complete trajectories generated by executing a sequence of policies $\pi_1, \dots, \pi_k$, on the event $\mathcal{E}_{Bonus}$ defined in Condition 4.4:*

$$V_1^*(s_1) - V_1^{\pi_k^{PVI}}(s_1) \leq 2\mathbb{E}_{\pi^*}\left[\sum_{h=1}^H Clip\left[\min\left\{H, 2B_1\sqrt{\frac{\log(B_2/\delta_k)}{N_{k,h}(s_h, a_h)}}\right\}\Bigg| \varepsilon_{Clip}\right]\right]. \qquad (1)$$

*where $\pi^*$ can be an arbitrary optimal policy, $\varepsilon_{Clip} := \frac{\Delta_{\min}}{2H+2}$ if $|\Pi^*| = 1$ and $\varepsilon_{Clip} := \frac{d_{\min}\Delta_{\min}}{2SAH}$ if $|\Pi^*| > 1$, where $d_{\min} := \min_{\pi \in \Pi^*, h \in [H], s_h \in \mathcal{S}_h, a_h \in \mathcal{A}_h} d^\pi(s_h, a_h)$ subject to $d^\pi(s_h, a_h) > 0$.*

*Proof.* We separately discuss the cases when there are unique or multiple deterministic optimal policies.

**Case 1: Unique Deterministic Optimal Policy**   For arbitrary $h, s_h$, suppose $\pi_k^{\mathrm{PVI}}(s_h) \notin \Pi^*(s_h)$, we have:

$$
\begin{aligned}
V_h^*(s_h) - \ddot{V}_{k,h}^{\pi^*}(s_h) \geq & V_h^*(s_h) - \widehat{V}_{k,h}(s_h) - (H - h + 1)\varepsilon_{\mathrm{Clip}} \\
\geq & \frac{1}{2}\Big(V_h^*(s_h) - \widehat{V}_{k,h}(s_h)\Big) + \frac{1}{2}\Big(V_h^*(s_h) - V_h^{\pi_k^{\mathrm{PVI}}}(s_h)\Big) - \frac{\Delta_{\min}}{2} \\
\geq & \frac{1}{2}\Big(V_h^*(s_h) - \widehat{V}_{k,h}(s_h)\Big) + \frac{1}{2}\Big(V_h^*(s_h) - Q_h^*(s_h, \pi_k^{\mathrm{PVI}})\Big) - \frac{\Delta_{\min}}{2} \\
= & \frac{1}{2}\Big(V_h^*(s_h) - \widehat{V}_{k,h}(s_h)\Big) + \frac{\Delta_h(s_h, \pi_k^{\mathrm{PVI}}(s_h))}{2} - \frac{\Delta_{\min}}{2} \\
\geq & \frac{1}{2}\Big(V_h^*(s_h) - \widehat{V}_{k,h}(s_h)\Big)
\end{aligned}
$$

Recall the definition of Events in Def.E.1, and note that when the optimal policy is unique, the events $\widetilde{\mathcal{E}}_{k,h,\pi^*}, \bar{\mathcal{E}}_{k,h,\pi^*}$ collapse to $\widetilde{\mathcal{E}}_{k,h}, \bar{\mathcal{E}}_{k,h}$, respectively. For arbitrary optimal policy $\pi^*$, we have:

$$
\begin{aligned}
V_1^*(s_1) - \ddot{V}_1^{\pi^*}(s_1) = & V_1^{\pi^*}(s_1) - \ddot{V}_1^{\pi^*}(s_1) \\
= & \mathbb{I}[\mathcal{E}_{k,1}]\Big(V_1^*(s_1) - \ddot{V}_1^{\pi^*}(s_1)\Big) + \mathbb{I}[\mathcal{E}_{k,1}^{\complement}]\Big(V_1^*(s_1) - \ddot{V}_1^{\pi^*}(s_1)\Big) \\
\geq & \mathbb{I}[\mathcal{E}_{k,1}]\frac{1}{2}\Big(V_1^*(s_1) - \widehat{V}_{k,1}(s_1)\Big) + \mathbb{I}[\mathcal{E}_{k,1}^{\complement}]\mathbb{P}(V_2^* - \ddot{V}_2^{\pi^*})(s_1, \pi^*) \\
\geq & \dots \\
\geq & \frac{1}{2}\mathbb{E}_{\pi^*}\Big[\sum_{h=1}^H \mathbb{I}[\widetilde{\mathcal{E}}_{k,h}](V_h^*(s_h) - \widehat{V}_{k,h}(s_h))\Big]
\end{aligned}
$$

Besides, on the other hand,

$$
\begin{aligned}
V_1^*(s_1) - V_1^{\pi_k^{\mathrm{PVI}}}(s_1) = & \mathbb{I}[\mathcal{E}_{k,1}](V_1^{\pi^*} - V_1^{\pi_k^{\mathrm{PVI}}}(s_1)) + \mathbb{I}[\mathcal{E}_{k,1}^{\complement}](V_1^{\pi^*} - Q_1^{\pi_k^{\mathrm{PVI}}}(s_1, \pi^*)) \\
= & \mathbb{I}[\mathcal{E}_{k,1}](V_1^{\pi^*} - V_1^{\pi_k^{\mathrm{PVI}}}(s_1)) + \mathbb{I}[\mathcal{E}_{k,1}^{\complement}]\mathbb{P}_1(V_2^* - V_2^{\pi_k^{\mathrm{PVI}}})(s_1, \pi^*)) \\
= & \dots \\
= & \mathbb{E}_{\pi_k^{\mathrm{PVI}}}\Big[\sum_{h=1}^H \mathbb{I}[\widetilde{\mathcal{E}}_{k,h}](V_h^{\pi^*}(s_h) - V_h^{\pi_k^{\mathrm{PVI}}}(s_h))\Big] \\
\leq & \mathbb{E}_{\pi_k^{\mathrm{PVI}}}\Big[\sum_{h=1}^H \mathbb{I}[\widetilde{\mathcal{E}}_{k,h}](V_h^{\pi^*}(s_h) - \widehat{V}_{k,h}(s_h))\Big]
\end{aligned}
$$

Combining the above two results and Lemma F.4, we finish the discussion for Case 1.

**Case 2: Non-unique Optimal Deterministic Policies**   From Lemma F.3, we know that,

$$
V_1^*(s_1) - V_1^{\pi_k^{\mathrm{PVI}}}(s_1) \leq V_1^*(s_1) - \widehat{V}_{k,1}(s_1) \leq \mathbb{E}_{\pi^*}\Big[\sum_{h=1}^H \mathbf{E}_{k,h}(s_h, a_h)\Big]
$$

where $\pi^*$ can be arbitrary optimal policy. Combining with Lemma F.7, we know that:

$$
\begin{aligned}
V_1^*(s_1) - V_1^{\pi_k^{\mathrm{PVI}}}(s_1) \leq & \mathrm{Clip}\Big[\mathbb{E}_{\pi^*}\Big[\sum_{h=1}^H \mathbf{E}_{k,h}(s_h, a_h)\Big]\big|d_{\min}\Delta_{\min}\Big] \\
\leq & 2\sum_{h=1}^H \sum_{s_h \in \mathcal{S}_h, a_h \in \mathcal{A}_h} \mathrm{Clip}\Big[d^{\pi^*}(s_h, a_h)\mathbf{E}_{k,h}(s_h, a_h)\big|\frac{d_{\min}\Delta_{\min}}{2SAH}\Big] \\
& \hspace{8cm} \text{(Lemma F.6)} \\
\leq & 2\sum_{h=1}^H \mathbb{E}_{\pi^*}\Big[\mathrm{Clip}\Big[\mathbf{E}_{k,h}(s_h, a_h)\big|\frac{d_{\min}\Delta_{\min}}{2SAH}\Big]\Big]
\end{aligned}
$$

where the last inequality is because $\mathrm{Clip}[\alpha x|\varepsilon] < \alpha\mathrm{Clip}[x|\varepsilon]$ as long as $\alpha < 1$. Combining with Lemma F.4, we finish the proof. $\qquad\square$

Next, we introduce a useful Lemma from [Dann et al., 2017]:

**Lemma F.8** (Lemma 7.4 in [Dann et al., 2017]). *Let $\mathcal{F}_i$ for $i, 1...$ be a filtration and $X_1, ...X_n$ be a sequence of Bernoulli random variables with $\Pr(X_i = 1|\mathcal{F}_{i-1}) = P_i$ with $P_i$ being $\mathcal{F}_{i-1}$-measurable and $X_i$ being $\mathcal{F}_i$ measurable. It holds that*

$$\Pr(\exists n : \sum_{i=1}^{n} X_i < \sum_{i=1}^{n} P_i/2 - W) \le e^{-W}$$

**Definition of Good Events** We first introduce some notations about good events which holds with high probability. We override the definition in Cond. 4.4 by assigning $\delta = \delta_k = 1/k^\alpha$ at iteration $k$, i.e.:

$$\mathcal{E}_{\mathbf{Bonus},k} := \bigcap_{h\in[H],s_h\in\mathcal{S}_h,a_h\in\mathcal{A}_h} \left\{ |\widehat{P}_{k,h}V_{k,h+1}(s_h,a_h) - P_hV_{k,h+1}(s_h,a_h)| < b_{k,h}(s_h,a_h) \right\}$$

$$\cap \left\{ b_{k,h}(s_h,a_h) \le B_1\sqrt{\frac{\log(B_2\cdot k^\alpha)}{N_{k,h}(s_h,a_h)}} \right\},$$

with $\quad b_k = \{b_{k,1}, ..., b_{k,H}\} \leftarrow \mathbf{Bonus}(D_k, 1/k^\alpha)$.

Besides, we use $\mathcal{E}_{\mathbf{Con},k}$ to denote the concentration event that

$$\mathcal{E}_{\mathbf{Con},k} := \bigcap_{h\in[H],s_h\in\mathcal{S}_h,a_h\in\mathcal{A}_h} \left\{ N_{k,h}(s_h,a_h) \ge \frac{1}{2}\sum_{k'=1}^{k} d^{\pi_{k'}^{\mathrm{O}}}(s_h,a_h) - \alpha\log(SAHk) \right\}$$

Finally, we use $\mathcal{E}_{\mathrm{Alg}^{\mathrm{O}},k}$ to denote the good events that the regret of $\mathrm{Alg}^{\mathrm{O}}$ is only at the level $\log k$:

$$\mathcal{E}_{\mathrm{Alg}^{\mathrm{O}},k} := \{\sum_{\widetilde{k}=1}^{k} V_1^*(s_1) - V_1^{\pi_{\widetilde{k}}^{\mathrm{O}}}(s_1) < C_1 + \alpha C_2 \log k\}$$

Based on Cond. 4.4, Cond. 4.6 and Lemma F.8, we have:

$$\Pr(\mathcal{E}_{\mathrm{Alg}^{\mathrm{O}},k}) \ge 1 - \frac{1}{k^\alpha}, \quad \Pr(\mathcal{E}_{\mathbf{Bonus},k}) \ge 1 - \frac{1}{k^\alpha}$$

$$\Pr(\mathcal{E}_{\mathbf{Con},k}) \ge 1 - SAH \cdot \exp(-\alpha\log(SAHk)) = 1 - \frac{SAH}{(SAHk)^\alpha} \ge 1 - \frac{1}{k^\alpha}$$

**Lemma F.9.** *[One Step Sub-optimality Gap Conditioning on Good Events] At iteration $k$, on the good events $\mathcal{E}_{\mathbf{Bonus},k}, \mathcal{E}_{\mathbf{Con},k}$ and $\mathcal{E}_{\mathrm{Alg}^{\mathrm{O}},k}$, the sub-optimality gap of $\pi_k^E$ can be upper bounded by:*

*(i) when $|\Pi^*| = 1$ (i.e. the optimal deterministic policy is unique):*

$$V_1^*(s_1) - V^{\pi_k^E}(s_1) \le 2\mathbb{E}_{\pi^*}\left[ \sum_{h=1}^{H} Clip\left[ \mathbb{I}[k < \bar\tau_{s_h,a_h}^{\pi^*}]\} \cdot H + \mathbb{I}[k \ge \bar\tau_{s_h,a_h}^{\pi^*}] \cdot B_1\sqrt{\frac{8\alpha\log(B_2k)}{kd^{\pi^*}(s_h,a_h)}} \Big| \varepsilon_{Clip} \right] \right]$$

*(ii) when $|\Pi^*| > 1$ (i.e. there are multiple optimal deterministic policy):*

$$V_1^*(s_1) - V^{\pi_k^E}(s_1) \le 2\mathbb{E}_{\pi_{cover}^*}\left[ \sum_{h=1}^{H} Clip\left[ \mathbb{I}[k < \bar\tau_{s_h,a_h}^{\pi_{cover}^*}]\} \cdot H + \mathbb{I}[k \ge \bar\tau_{s_h,a_h}^{\pi_{cover}^*}] \cdot B_1\sqrt{\frac{4\alpha\log(B_2k)}{k\widetilde{d}^{\pi_{cover}^*}(s_h,a_h)}} \Big| \varepsilon_{Clip}' \right] \right]$$

*where $\varepsilon_{Clip} := \frac{\Delta_{\min}}{2H+2}$ and $\varepsilon_{Clip}' := \frac{d_{\min}\Delta_{\min}}{2SAH}$; $\pi_{cover}^*$ and $\widetilde{d}^{\pi_{cover}^*}(s_h,a_h)$ are defined in Thm. 4.8; besides,*

$$\bar\tau_{s_h,a_h}^{\pi} := c_\tau \frac{\alpha(C_1+C_2)}{d^\pi(s_h,a_h)\Delta_{\min}} \log \frac{\alpha SAH(C_1+C_2)}{d^\pi(s_h,a_h)\Delta_{\min}}, \quad \bar\tau_{s_h,a_h}^{\pi_{cover}^*} := c_\tau' \frac{\alpha(C_1+C_2)}{\widetilde{d}^{\pi_{cover}^*}(s_h,a_h)\Delta_{\min}} \log \frac{\alpha SAH(C_1+C_2)}{\widetilde{d}^{\pi_{cover}^*}(s_h,a_h)\Delta_{\min}}.$$

*for some constant $c_\tau, c_\tau'$.*

*Proof.* We first discuss the case when $|\Pi^*| = 1$.

**Case 1: unique optimal deterministic policy** As a result of Thm. 4.5, on the event $\mathcal{E}_{\mathbf{Bonus},k}$, we show that the sub-optimality gap of $\pi_k^{\mathrm{E}}$ can be upper bounded by:

$$V_1^*(s_1) - V_1^{\pi_k^{\mathrm{E}}}(s_1) \leq 2\mathbb{E}_{\pi^*}\Big[\sum_{h=1}^{H} \ddot{\mathbf{E}}_{k,h}(s_h, a_h)\Big] = \sum_{h=1}^{H}\sum_{s_h, a_h} d^{\pi^*}(s_h, a_h)\ddot{\mathbf{E}}_{k,h}(s_h, a_h)$$

Because of Lemma F.4, the above further implies that:

$$V_1^*(s_1) - V_1^{\pi_k^{\mathrm{E}}}(s_1) \leq 2\mathbb{E}_{\pi^*}\Big[d\min\{H - h + 1, 2B_1\sqrt{\frac{\alpha\log(B_2 k)}{N_{k,h}(s_h, a_h)}}\}\Big].$$

Because of Thm. 4.7, on the event $\mathcal{E}_{\mathbf{Con},k}$ and $\mathcal{E}_{\mathrm{Alg}^{\mathrm{O}},k}$, we further have:

$$
\begin{aligned}
N_{k,h}(s_h, a_h) &\geq \frac{1}{2}\sum_{k'=1}^{k} d^{\pi_{k'}^{\mathrm{O}}}(s_h, a_h) - \alpha\log(SAHk) \\
&\geq \frac{1}{2}\sum_{k'=1}^{k} d^{\pi_{k'}^*}(s_h, a_h) - \alpha\log(SAHk) - \frac{1}{\Delta_{\min}}(C_1 + \alpha C_2 \log k) \\
&\geq \frac{k}{2}d^{\pi^*}(s_h, a_h) - \alpha\log(SAHk) - \frac{1}{\Delta_{\min}}(C_1 + \alpha C_2 \log k)
\end{aligned}
$$

Now, we define that,

$$\tau_{s_h, a_h}^{\pi^*} := \inf_{t:\forall t' \geq t}\{\frac{1}{4}td^{\pi^*}(s_h, a_h) \geq \alpha\log(SAHt) + \frac{1}{\Delta_{\min}}(C_1 + \alpha C_2 \log t)\}$$

there must exists a constant $c_\tau$ independent with $C_1, C_2, \alpha, d^{\pi^*}(s_h, a_h)$ and $\Delta_{\min}$, such that:

$$\forall h \in [H], s_h \in \mathcal{S}_h, a_h \in \mathcal{A}_h, \quad \tau_{s_h, a_h}^{\pi^*} \leq \bar\tau_{s_h, a_h}^{\pi^*} := c_\tau \frac{\alpha(C_1 + C_2)}{d^{\pi^*}(s_h, a_h)\Delta_{\min}}\log\frac{\alpha SAH(C_1 + C_2)}{d^{\pi^*}(s_h, a_h)\Delta_{\min}}.$$

Easy to check that, for arbitrary $k \geq \bar\tau_{s_h, a_h}^{\pi^*}$, on the good events, we can verify that $N_{k,h} \geq \frac{k}{4}d^{\pi^*}(s_h, a_h) \geq \frac{\bar\tau_{s_h, a_h}^{\pi^*}}{4}d^{\pi^*}(s_h, a_h) \geq \frac{c_\tau}{4} > 0$, and as a result, we have:

$$
\begin{aligned}
&V_1^*(s_1) - V_1^{\pi_k^{\mathrm{E}}}(s_1) \\
&\leq 2\mathbb{E}_{\pi^*}\Big[\sum_{h=1}^{H}\mathrm{Clip}\Big[\min\{H - h + 1, 2B_1\sqrt{\frac{\alpha\log(B_2 k)}{N_{k,h}(s_h, a_h)}}\}\Big|\varepsilon_{\mathrm{Clip}}\Big]\Big] \\
&\leq 2\mathbb{E}_{\pi^*}\Big[\sum_{h=1}^{H}\mathrm{Clip}\Big[\mathbb{I}[k < \bar\tau_{s_h, a_h}^{\pi^*}]\} \cdot H + \mathbb{I}[k \geq \bar\tau_{s_h, a_h}^{\pi^*}] \cdot B_1\sqrt{\frac{\alpha\log(B_2 k)}{kd^{\pi^*}(s_h, a_h)/4}}\Big|\varepsilon_{\mathrm{Clip}}\Big]\Big]
\end{aligned}
$$

**Case 2: multiple optimal deterministic policies** The discussion are similar. As a result of Thm. 4.8, on the event $\mathcal{E}_{\mathbf{Con},k}$ and $\mathcal{E}_{\mathrm{Alg}^{\mathrm{O}},k}$, we further have:

$$
\begin{aligned}
N_{k,h}(s_h, a_h) &\geq \frac{k}{4} \cdot \max\{\frac{d_{h,\min}^*(s_h, a_h)}{(|\mathcal{Z}_{h,\mathrm{div}}| + 1)H}, d^{\pi_{\mathrm{cover}}^*}(s_h, a_h)\} - \alpha\log(SAHk) - \frac{1}{\Delta_{\min}}\Big(\sum_{k=1}^{K} V_1^*(s_1) - V_1^{\pi_k}(s_1)\Big) \\
&\geq \frac{k}{4}\widetilde{d}^{\pi_{\mathrm{cover}}^*}(s_h, a_h) - \alpha\log(SAHk) - \frac{1}{\Delta_{\min}}(C_1 + \alpha C_2 \log k)
\end{aligned}
$$

Similarly, we define that,

$$\tau_{s_h, a_h}^{\pi_{\mathrm{cover}}^*} := \inf_{t:\forall t' \geq t}\{\frac{t}{8}\widetilde{d}^{\pi_{\mathrm{cover}}^*}(s_h, a_h) \geq \alpha\log(SAHt) + \frac{1}{\Delta_{\min}}(C_1 + \alpha C_2 \log t)\}$$

there must exists a constant $c_\tau'$ independent with $C_1, C_2, \alpha, \widetilde{d}^{\pi_{\mathrm{cover}}^*}(s_h, a_h)$ and $\Delta_{\min}$, such that:

$$\forall h \in [H], s_h \in \mathcal{S}_h, a_h \in \mathcal{A}_h, \quad \tau_{s_h, a_h}^{\pi_{\mathrm{cover}}^*} \leq \bar\tau_{s_h, a_h}^{\pi_{\mathrm{cover}}^*} := c_\tau' \frac{\alpha(C_1 + C_2)}{\widetilde{d}^{\pi_{\mathrm{cover}}^*}(s_h, a_h)\Delta_{\min}}\log\frac{\alpha SAH(C_1 + C_2)}{\widetilde{d}^{\pi_{\mathrm{cover}}^*}(s_h, a_h)\Delta_{\min}}.$$

For arbitrary $k \geq \bar{\tau}_{s_h,a_h}^{\pi_{\text{cover}}^*}$, on the good events, we can verify that $N_{k,h} \geq \frac{k}{8} d^{\pi_{\text{cover}}^*}(s_h, a_h) > 0$, and as a result, we have:

$$V_1^*(s_1) - V_1^{\pi_k^{\text{E}}}(s_1) \leq 2\mathbb{E}_{\pi_{\text{cover}}^*}\Big[\sum_{h=1}^{H}\text{Clip}\Big[\mathbb{I}[k < \bar{\tau}_{s_h,a_h}^{\pi_{\text{cover}}^*}]\} \cdot H + \mathbb{I}[k \geq \bar{\tau}_{s_h,a_h}^{\pi_{\text{cover}}^*}] \cdot B_1\sqrt{\frac{8\alpha \log(B_2 k)}{k\widetilde{d}^{\pi_{\text{cover}}^*}(s_h, a_h)}}\Big|\varepsilon_{\text{Clip}}'\Big]\Big]$$

$\square$

Now, we are ready to prove the main theorem.

**Theorem 4.9.** *By running an Algorithm satisfying Condition 4.6 as $\text{Alg}^O$, running Alg 3 as $\text{Alg}^E$ with a bonus term function **Bonus** satisfying Condition 4.4 and $\delta_k = 1/k^\alpha$, for some constant $\alpha > 1$, for arbitrary $K \geq 1$, the exploitation regret of $\text{Alg}^E$ can be upper bounded by:*

*(i) When $|\Pi^*| = 1$ (unique optimal deterministic policy):*

$$\text{Regret}_K(\text{Alg}^E) \leq O\Big(\sum_{h=1}^{H}\sum_{\substack{s_h,a_h: \\ d^{\pi^*}(s_h,a_h)>0}}\Big(\frac{C_1 + C_2}{\Delta_{\min}}\log\frac{SAH(C_1+C_2)}{d^{\pi^*}(s_h,a_h)\Delta_{\min}} + \frac{B_1 H}{\Delta_{\min}}\log\frac{B_2 H}{d^{\pi^*}(s_h,a_h)\Delta_{\min}}\Big)\Big).$$

*(ii) When $|\Pi^*| > 1$ (non-unique optimal deterministic policies):*

$$\text{Regret}_K(\text{Alg}^E) \leq O\Big(\sum_{h=1}^{H}\sum_{\substack{s_h,a_h: \\ d^{\pi_{\text{cover}}^*}(s_h,a_h)>0}}\Big(\frac{C_1 + C_2}{\Delta_{\min}}\log\frac{SAH(C_1+C_2)}{\widetilde{d}^{\pi_{\text{cover}}^*}(s_h,a_h)\Delta_{\min}} + \frac{B_1 SAH}{d_{\min}\Delta_{\min}}\log\frac{B_2 SAH}{d_{\min}\Delta_{\min}}\Big)\Big).$$

*where $\pi_{\text{cover}}^*$ and $\widetilde{d}^{\pi_{\text{cover}}^*}(s_h, a_h)$ are introduced in Theorem 4.8.*

*Proof.* Because the expectation and summation are linear, we have:

$$\mathbb{E}[\sum_{k=1}^{K} V^* - V^{\pi_k^{\text{E}}}] = \sum_{k=1}^{K}\mathbb{E}[V^* - V^{\pi_k^{\text{E}}}]$$

Therefore, in the following, we first provide an upper bound for each $\mathbb{E}[V^* - V^{\pi_k^{\text{E}}}]$. Note that the expected regert at step $k$ can be upper bounded by:

$$\mathbb{E}_{\text{Alg}^O, M, \text{Alg}^E}[V_1^*(s_1) - V_1^{\pi_k^{\text{E}}}(s_1)]$$
$$= \Pr(\mathcal{E}_{\textbf{Bonus},k} \cap \mathcal{E}_{\textbf{Con},k} \cap \mathcal{E}_{\text{Alg}^O,k})\mathbb{E}_{\text{Alg}^O,M,\text{Alg}^E}[V_1^*(s_1) - V_1^{\pi_k^{\text{E}}}(s_1)|\mathcal{E}_{\textbf{Bonus},k} \cap \mathcal{E}_{\textbf{Con},k} \cap \mathcal{E}_{\text{Alg}^O,k}]$$
$$+ \Pr(\mathcal{E}_{\textbf{Bonus},k}^{\complement} \cup \mathcal{E}_{\textbf{Con},k}^{\complement} \cup \mathcal{E}_{\text{Alg}^O,k}^{\complement})\mathbb{E}_{\text{Alg}^O,M,\text{Alg}^E}[V_1^*(s_1) - V_1^{\pi_k^{\text{E}}}(s_1)|\mathcal{E}_{\textbf{Bonus},k}^{\complement} \cup \mathcal{E}_{\textbf{Con},k}^{\complement} \cup \mathcal{E}_{\text{Alg}^O,k}^{\complement}]$$
$$\leq \Pr(\mathcal{E}_{\textbf{Bonus},k} \cap \mathcal{E}_{\textbf{Con},k} \cap \mathcal{E}_{\text{Alg}^O,k})\mathbb{E}_{\text{Alg}^O,M,\text{Alg}^E}[V_1^*(s_1) - V_1^{\pi_k^{\text{E}}}(s_1)|\mathcal{E}_{\textbf{Bonus},k} \cap \mathcal{E}_{\textbf{Con},k} \cap \mathcal{E}_{\text{Alg}^O,k}] + \frac{3H}{k^\alpha}$$

Easy to see that $\lim_{K \to \infty}\sum_{k=1}^{K}\frac{1}{k^\alpha} < \frac{\alpha}{\alpha-1} < \infty$ as long as $\alpha > 1$, therefore, in the following, we mainly focus on the first part, and separately discuss its upper bound for the case when $|\Pi^*| = 1$ or $|\Pi^*| > 1$.

**Case 1: $|\Pi^*| = 1$ (Unique Optimal Policy)** We use $\pi^*$ to denote the unique optimal policy and define:

$$\tau_{s_h,a_h,\varepsilon_{\text{Clip}}}^{\pi^*} := \inf_{t, \forall t' \geq t}\{B_1\sqrt{\frac{\alpha \log(B_2 t)}{td^{\pi^*}(s_h,a_h)/4}} < \varepsilon_{\text{Clip}}\}$$

Recall that $\varepsilon_{\text{Clip}} := \Delta_{\min}/(2H + 2)$, it's easy to verify that, there exists a constant $c_{\text{Clip}}$ such that,

$$\tau_{s_h,a_h,\varepsilon_{\text{Clip}}}^{\pi^*} \leq \widetilde{\tau}_{s_h,a_h,\varepsilon_{\text{Clip}}}^{\pi^*} := c_{\text{Clip}}\frac{\alpha H^2}{d^{\pi^*}(s_h,a_h)\Delta_{\min}^2}\log\frac{\alpha B_2 H}{d^{\pi^*}(s_h,a_h)\Delta_{\min}}$$

Then, we have:

$$\lim_{K\to\infty}\sum_{k=1}^{K}2\mathbb{E}_{\pi^*}\Big[\sum_{h=1}^{H}\mathrm{Clip}\Big[\mathbb{I}[k<\bar\tau^{\pi^*}_{s_h,a_h}]\}\cdot H+\mathbb{I}[k\geq\bar\tau^{\pi^*}_{s_h,a_h}]\cdot B_1\sqrt{\frac{\alpha\log(B_2 k)}{kd^{\pi^*}(s_h,a_h)/4}}\Big|\varepsilon_{\mathrm{Clip}}\Big]\Big]$$

$$=2\mathbb{E}_{\pi^*}\Big[\sum_{h=1}^{H}\Big(\sum_{k=1}^{\bar\tau^{\pi^*}_{s_h,a_h}}H+\sum_{k=\bar\tau^{\pi^*}_{s_h,a_h}+1}^{K}\mathrm{Clip}\Big[B_1\sqrt{\frac{\alpha\log(B_2 k)}{kd^{\pi^*}(s_h,a_h)/4}}\Big|\varepsilon_{\mathrm{Clip}}\Big]\Big)\Big]$$

$$\leq2\mathbb{E}_{\pi^*}[\sum_{h=1}^{H}\sum_{k=1}^{\bar\tau^{\pi^*}_{s_h,a_h}}H]+2\mathbb{E}_{\pi^*}\Big[\sum_{h=1}^{H}\int_{x=\bar\tau^{\pi^*}_{s_h,a_h}}^{\widetilde\tau^{\pi^*}_{s_h,a_h,\varepsilon_{\mathrm{Clip}}}}B_1\sqrt{\frac{\alpha\log(B_2 x)}{xd^{\pi^*}(s_h,a_h)/4}}dx\Big]$$

$$\leq2\mathbb{E}_{\pi^*}[\sum_{h=1}^{H}\sum_{k=1}^{\bar\tau^{\pi^*}_{s_h,a_h}}H]+2\sum_{h=1}^{H}\sum_{\substack{s_h,a_h:\\d^{\pi^*}(s_h,a_h)>0}}B_1\sqrt{4\alpha d^{\pi^*}(s_h,a_h)}\int_{x=\bar\tau^{\pi^*}_{s_h,a_h}}^{\widetilde\tau^{\pi^*}_{s_h,a_h,\varepsilon_{\mathrm{Clip}}}}\sqrt{\frac{\log(B_2 x)}{x}}dx$$

For the first part, we have:

$$\mathbb{E}_{\pi^*}[\sum_{h=1}^{H}\sum_{k=1}^{\bar\tau^{\pi^*}_{s_h,a_h}}H]=\sum_{h=1}^{H}\sum_{\substack{s_h,a_h:\\d^{\pi^*}(s_h,a_h)>0}}d^{\pi^*}(s_h,a_h)\cdot H\cdot\bar\tau^{\pi^*}_{s_h,a_h}$$

$$\leq c_\tau\frac{\alpha H(C_1+C_2)}{\Delta_{\min}}\cdot\sum_{h=1}^{H}\sum_{\substack{s_h,a_h:\\d^{\pi^*}(s_h,a_h)>0}}\log\frac{\alpha SAH(C_1+C_2)}{d^{\pi^*}(s_h,a_h)\Delta_{\min}}$$

For the second part, we have:

$$\mathbb{E}_{\pi^*}\Big[\sum_{h=1}^{H}\int_{x=\bar\tau^{\pi^*}_{s_h,a_h}}^{\widetilde\tau^{\pi^*}_{s_h,a_h,\varepsilon_{\mathrm{Clip}}}}B_1\sqrt{\frac{\alpha\log(B_2 x)}{xd^{\pi^*}(s_h,a_h)/4}}dx\Big]$$

$$\leq\sum_{h=1}^{H}\sum_{\substack{s_h,a_h:\\d^{\pi^*}(s_h,a_h)>0}}B_1\sqrt{4\alpha d^{\pi^*}(s_h,a_h)}\int_{x=\bar\tau^{\pi^*}_{s_h,a_h}}^{\widetilde\tau^{\pi^*}_{s_h,a_h,\varepsilon_{\mathrm{Clip}}}}\sqrt{\frac{\log(B_2 x)}{x}}dx$$

$$\leq2\sum_{h=1}^{H}\sum_{\substack{s_h,a_h:\\d^{\pi^*}(s_h,a_h)>0}}B_1\sqrt{4\alpha d^{\pi^*}(s_h,a_h)}\cdot2\Big(\sqrt{\widetilde\tau^{\pi^*}_{s_h,a_h,\varepsilon_{\mathrm{Clip}}}\log B_2\widetilde\tau^{\pi^*}_{s_h,a_h,\varepsilon_{\mathrm{Clip}}}}-\sqrt{\bar\tau^{\pi^*}_{s_h,a_h}\log B_2\bar\tau^{\pi^*}_{s_h,a_h}}\Big)$$

(Lemma F.10)

$$\leq c_2\frac{\alpha B_1 H}{\Delta_{\min}}\sum_{h=1}^{H}\sum_{\substack{s_h,a_h:\\d^{\pi^*}(s_h,a_h)>0}}\log\frac{\alpha B_2 H}{d^{\pi^*}(s_h,a_h)\Delta_{\min}}$$

where in the last step, we drop the term $-\sqrt{\bar\tau^{\pi^*}_{s_h,a_h}\log B_2\bar\tau^{\pi^*}_{s_h,a_h}}$, and $c_2$ is a constant.

Combining the above results, we have:

$$\mathbb{E}_{\mathrm{Alg}^{\mathrm{O}},M,\mathrm{Alg}^{\mathrm{E}}}[\sum_{k=1}^{K}V_1^*(s_1)-V_1^{\pi_k^{\mathrm{E}}}(s_1)]$$

$$\leq\sum_{k=1}^{K}\frac{3H}{k^\alpha}+c_\tau\frac{\alpha(C_1+C_2)}{\Delta_{\min}}\cdot\sum_{h=1}^{H}\sum_{\substack{s_h,a_h:\\d^{\pi^*}(s_h,a_h)>0}}\log\frac{\alpha SAH(C_1+C_2)}{d^{\pi^*}(s_h,a_h)\Delta_{\min}}$$

$$+c_2\frac{\alpha B_1 H}{\Delta_{\min}}\sum_{h=1}^{H}\sum_{\substack{s_h,a_h:\\d^{\pi^*}(s_h,a_h)>0}}\log\frac{\alpha B_2 H}{d^{\pi^*}(s_h,a_h)\Delta_{\min}}$$

$$\leq \frac{3\alpha H}{\alpha - 1} + c_{\mathrm{Alg^E}} \cdot \Big( \sum_{h=1}^{H} \sum_{\substack{s_h,a_h: \\ d^{\pi^*}(s_h,a_h)>0}} \frac{\alpha(C_1+C_2)}{\Delta_{\min}} \log \frac{\alpha SAH(C_1+C_2)}{d^{\pi^*}(s_h,a_h)\Delta_{\min}} + \frac{\alpha B_1 H}{\Delta_{\min}} \log \frac{\alpha B_2 H}{d^{\pi^*}(s_h,a_h)\Delta_{\min}} \Big)$$

where $c_{\mathrm{Alg^E}}$ is some constant.

**Case 2:** $|\Pi^*| > 1$ **(Non-Unique Optimal Policy)**  Similar to the discussion above, we define:

$$\tau^{\pi^*_{\mathrm{cover}}}_{s_h,a_h,\varepsilon'_{\mathrm{Clip}}} := \inf_{t,\forall t' \geq t} \Big\{ B_1 \sqrt{\frac{8\alpha \log(B_2 t)}{t\widetilde{d}^{\pi^*_{\mathrm{cover}}}(s_h,a_h)}} < \varepsilon'_{\mathrm{Clip}} \Big\}$$

Recall that $\varepsilon'_{\mathrm{Clip}} := d_{\min}\Delta_{\min}/(2SAH)$, it's easy to verify that, there exists a constant $c_{\mathrm{Clip}}$ such that,

$$\tau^{\pi^*_{\mathrm{cover}}}_{s_h,a_h,\varepsilon'_{\mathrm{Clip}}} \leq \widetilde{\tau}^{\pi^*_{\mathrm{cover}}}_{s_h,a_h,\varepsilon'_{\mathrm{Clip}}} := c_{\mathrm{Clip}} \frac{\alpha(SAH)^2}{\widetilde{d}^{\pi^*_{\mathrm{cover}}}(s_h,a_h)(d_{\min}\Delta_{\min})^2} \log \frac{\alpha B_2 SAH}{\widetilde{d}^{\pi^*_{\mathrm{cover}}}(s_h,a_h)d_{\min}\Delta_{\min}}$$

Following a similar discussion, we have:

$$\lim_{K\to\infty} \sum_{k=1}^{K} 2\mathbb{E}_{\pi^*_{\mathrm{cover}}} \Big[ \sum_{h=1}^{H} \mathrm{Clip}\Big[ \mathbb{I}[k < \bar{\tau}^{\pi^*_{\mathrm{cover}}}_{s_h,a_h}]\} \cdot H + \mathbb{I}[k \geq \bar{\tau}^{\pi^*_{\mathrm{cover}}}_{s_h,a_h}] \cdot B_1 \sqrt{\frac{8\alpha \log(B_2 k)}{k\widetilde{d}^{\pi^*_{\mathrm{cover}}}(s_h,a_h)}} \Big| \varepsilon'_{\mathrm{Clip}} \Big] \Big]$$

$$\leq 2\mathbb{E}_{\pi^*_{\mathrm{cover}}}[\sum_{h=1}^{H} \sum_{k=1}^{\bar{\tau}^{\pi^*_{\mathrm{cover}}}_{s_h,a_h}} H] + 2\sum_{h=1}^{H} \sum_{\substack{s_h,a_h: \\ d^{\pi^*_{\mathrm{cover}}}(s_h,a_h)>0}} B_1 \sqrt{8\alpha d^{\pi^*_{\mathrm{cover}}}(s_h,a_h)} \int_{x=\bar{\tau}^{\pi^*_{\mathrm{cover}}}_{s_h,a_h}}^{\widetilde{\tau}^{\pi^*_{\mathrm{cover}}}_{s_h,a_h,\varepsilon'_{\mathrm{Clip}}}} \sqrt{\frac{\log(B_2 x)}{x}} dx$$

$$\leq c'_\tau \frac{\alpha H(C_1+C_2)}{\Delta_{\min}} \cdot \sum_{h=1}^{H} \sum_{\substack{s_h,a_h: \\ d^{\pi^*_{\mathrm{cover}}}(s_h,a_h)>0}} \log \frac{\alpha SAH(C_1+C_2)}{\widetilde{d}^{\pi^*_{\mathrm{cover}}}(s_h,a_h)\Delta_{\min}}$$

$$+ c'_2 \frac{\alpha B_1 SAH}{d_{\min}\Delta_{\min}} \sum_{h=1}^{H} \sum_{\substack{s_h,a_h: \\ d^{\pi^*_{\mathrm{cover}}}(s_h,a_h)>0}} \log \frac{\alpha B_2 SAH}{d_{\min}\Delta_{\min}} \qquad\qquad (\text{Note that } \widetilde{d}^{\pi^*_{\mathrm{cover}}} \geq d^{\pi^*_{\mathrm{cover}}})$$

Therefore, we have:

$$\mathbb{E}_{\mathrm{Alg^O},M,\mathrm{Alg^E}}[\sum_{k=1}^{K} V_1^*(s_1) - V_1^{\pi_k^{\mathrm{E}}}(s_1)]$$

$$\leq \frac{3\alpha H}{\alpha - 1} + c'_{\mathrm{Alg^E}} \cdot \Big( \sum_{h=1}^{H} \sum_{\substack{s_h,a_h: \\ d^{\pi^*_{\mathrm{cover}}}(s_h,a_h)>0}} \frac{\alpha(C_1+C_2)}{\Delta_{\min}} \log \frac{\alpha SAH(C_1+C_2)}{\widetilde{d}^{\pi^*_{\mathrm{cover}}}(s_h,a_h)\Delta_{\min}} + \frac{\alpha B_1 SAH}{d_{\min}\Delta_{\min}} \log \frac{\alpha B_2 SAH}{d_{\min}\Delta_{\min}} \Big)$$

$$\square$$

**Lemma F.10** (Computation of Integral)**.** *Suppose $p \geq 1$, $b \geq a \geq e/p$, then we have:*

$$\int_a^b \sqrt{\frac{\log px}{x}} dx \leq 2(\sqrt{b \log pb} - \sqrt{a \log pa})$$

*Proof.*

$$\int_a^b \sqrt{\frac{\log px}{x}} dx \leq \int_a^b \frac{1}{\sqrt{x \log px}} + \sqrt{\frac{\log px}{x}} dx = 2\int_a^b (\sqrt{x \log px})' = 2(\sqrt{b \log pb} - \sqrt{a \log pa})$$

$$\square$$

# G   Doubling Trick for Alg$^O$ Satisfying Cond. G.1

As we briefly mentioned in Sec.4.2.2, Cond.4.6 may not holds for some algorithms with near-optimal regret guarantees. For example, in [Simchowitz and Jamieson, 2019, Xu et al., 2021, Dann et al., 2021], although these algorithms are anytime, they require a confidence interval $\delta$ as input at the beginning of the algorithm and fix it during the running, which we abstract into the Cond.G.1 below:

**Condition G.1** (Alternative Condition of Alg$^O$). *Alg$^O$ is an algorithm which returns a deterministic policies $\pi_{\widetilde{k}}^O$ at each iteration $\widetilde{k}$, and for arbitrary fixed $k \geq 2$, with probability $1 - \delta$, we have the following holds:*

$$\sum_{\widetilde{k}=1}^{k} V_1^*(s_1) - V_1^{\pi_{\widetilde{k}}^O}(s_1) \leq C_1 + C_2 \log \frac{k}{\delta}$$

*where $C_1, C_2$ are some parameters depending on $S, A, H$ and $\Delta_h(s_h, a_h)$ and independent with $k$.*

As a result, no matter how small $\delta$ is chosen at the beginning, when $k \geq \lceil 1/\delta \rceil$, the Cond. 4.6 can not be directly guaranteed. To overcome this issue, we present a new framework in Alg 5 inspired by doubling trick.

---

**Algorithm 5:** Tiered RL Algorithm with Doubling Trick

---

1 **Input**: $\alpha > 1$.
2 $K_0 = 1, \quad k = 1, \quad \pi_{1,1}^E \leftarrow \text{Alg}^E(\{\})$.
3 **for** $n = 1, 2, ...$ **do**
4 $\quad$ $K_n \leftarrow 2K_n, \quad \delta_{n-1} = 1/K_n^\alpha, \quad D_{n,1} \leftarrow \{\}$
5 $\quad$ **for** $k = 1, ..., K_n$ **do**
6 $\quad\quad$ // Here we do not update $\pi^E$
7 $\quad\quad$ $\pi_{n,k+1}^O \leftarrow \text{Alg}^O(D_{n,k}, \delta_n)$.
8 $\quad\quad$ $\pi_{n,k+1}^E = \begin{cases} \pi_{n-1,K_{n-1}/2+\lceil k/2 \rceil}^E, & \text{If } k \leq K_n/2, \\ \text{Alg}^E(D_{n,k}, 1/k^\alpha), & \text{Otherwise.} \end{cases}$
9 $\quad\quad$ $\tau_{k+1} \sim \pi_{n,k+1}^O$
10 $\quad\quad$ $D_{n,k+1} = D_{n,k} \cup \tau_{n,k+1}$
11 $\quad$ **end**
12 **end**

---

The basic idea is to iteratively run Alg$^O$ satisfying Cond.G.1 from scratch while gradually doubling the number of iterations (i.e. $K_n$) and shrinking the confidence level $\delta_n$ rather than runnning with a fixed $\delta$ forever. Besides, another crucial part is the computation of $\pi_k^E$. Instead of continuously updating $\pi^E$ with the data collected before, we only update the exploitation policy when $k \geq K_n/2$ for each outer loop $n$. As we will discuss in Lemma G.2, Alg$_{n,k}^E$ will behave as if the dataset is generated by another online algorithm satisfying Cond. 4.6, and therefore, the analysis based on Cond. 4.6 can be adapted here, which we summarize to Thm. G.3 below.

**Lemma G.2.** *By running an algorithm satisfying Cond. G.1 in Alg. 5 as Alg$^O$, for arbitrary $n \geq 1$ and $K_n/2 + 1 \leq k < K_n/2$, we have:*

$$\Pr(\sum_{k=1}^{K} V^* - V^{\pi_{n,k}^O} > C_1' + \alpha C_2' \log k) \leq 1/k^\alpha$$

*with $C_1' = C_1 + (\alpha + 1)C_2 \log 2$ and $C_2' = \frac{\alpha+1}{\alpha} C_2$.*

*Proof.* Based on Cond. G.1, we know that:

$$\Pr(\sum_{k'=1}^{k} V_1^*(s_1) - V_1^{\pi_{k'}^O}(s_1) > C_1 + C_2 \log \frac{k}{\delta_n}) \leq \delta_n$$

Since $\delta_n = 1/K_n^\alpha$ and $k \geq K_n/2$, we have:

$$\Pr(\sum_{k'=1}^k V_1^*(s_1) - V_1^{\pi_{k'}^O}(s_1) > C_1 + (1+\alpha)C_2 \log 2k)$$

$$= \Pr(\sum_{k'=1}^k V_1^*(s_1) - V_1^{\pi_{k'}^O}(s_1) > C_1 + C_2 \log(2k)^{1+\alpha})$$

$$\leq \Pr(\sum_{k'=1}^k V_1^*(s_1) - V_1^{\pi_{k'}^O}(s_1) > C_1 + C_2 \log \frac{k}{\delta_n}) \qquad ((2k)^{1+\alpha} \geq 2kK_n^\alpha > k/\delta_n)$$

$$\leq \delta_n \leq 1/k^\alpha$$

$\square$

Now, we are ready to upper bound the regret of $\mathrm{Alg}^E$:

**Theorem G.3.** *By choosing an arbitrary algorithm satisfying Cond. G.1 as $\mathrm{Alg}^O$, choosing Alg. 3 as $\mathrm{Alg}^E$ and choosing a bonus function satisfying Cond. 4.4 as **Bonus**, the Pseudo regret of $\pi_{n,k}^E$ in Alg. 5 can be upper bounded by:*

*(i) $|\Pi^*| = 1$ (unique optimal deterministic policy):*

$$\mathbb{E}[\sum_{n=1}^N \sum_{k=1}^{K_n} V_1^*(s_1) - V^{\pi_{n,k}^E}] \leq 2H + \frac{9\alpha H}{\alpha - 1}$$

$$+ 3c_{\mathrm{Alg}^E} \cdot \Big( \sum_{h=1}^H \sum_{\substack{s_h, a_h: \\ d^{\pi^*}(s_h, a_h) > 0}} \frac{\alpha(C_1' + C_2')}{\Delta_{\min}} \log \frac{\alpha SAH(C_1' + C_2')}{d^{\pi^*}(s_h, a_h)\Delta_{\min}} + \frac{\alpha B_1 H}{\Delta_{\min}} \log \frac{\alpha B_2 H}{d^{\pi^*}(s_h, a_h)\Delta_{\min}} \Big)$$

*(ii) $|\Pi^*| > 1$ (non-unique optimal deterministic policies):*

$$\mathbb{E}[\sum_{n=1}^N \sum_{k=1}^{K_n} V_1^*(s_1) - V^{\pi_{n,k}^E}] \leq 2H + \frac{9\alpha H}{\alpha - 1}$$

$$+ 3c_{\mathrm{Alg}^E}' \cdot \Big( \sum_{h=1}^H \sum_{\substack{s_h, a_h: \\ d^{\pi_{cover}^*}(s_h, a_h) > 0}} \frac{\alpha(C_1' + C_2')}{\Delta_{\min}} \log \frac{\alpha SAH(C_1' + C_2')}{\widetilde{d}^{\pi_{cover}^*}(s_h, a_h)\Delta_{\min}} + \frac{\alpha B_1 SAH}{d_{\min}\Delta_{\min}} \log \frac{\alpha B_2 SAH}{d_{\min}\Delta_{\min}} \Big)$$

*where $C_1' = C_1 + (\alpha + 1)C_2 \log 2$ and $C_2' = \frac{\alpha+1}{\alpha}C_2$.*

**Remark G.4** ($O(\log^2 K)$-Regret of $\mathrm{Alg}^O$). *Although the regret of $\mathrm{Alg}^E$ stays constant under this framework, it is easy to verify that the pseudo-regret of $\mathrm{Alg}^O$ will be $O(\log^2 K)$ as a result of the doubling trick, which is worse than $O(\log K)$ up to a factor of $\log K$. Therefore, more rigorously speaking, the regret of $\mathrm{Alg}^O$ will be almost near-optimal.*

*Proof.* The key observation is that one can decompose the total expected regret into two parts:

$$\mathbb{E}[\sum_{n=1}^N \sum_{k=1}^{K_n} V^* - V^{\pi_{n,k}^E}] = \mathbb{E}[\sum_{n=1}^N \sum_{k=1}^{K_n/2} V^* - V^{\pi_{n,k}^E}] + \mathbb{E}[\sum_{n=1}^N \sum_{k=K_n/2+1}^{K_n} V^* - V^{\pi_{n,k}^E}]$$

$$= 2\mathbb{E}[\sum_{n=0}^{N-1} \sum_{k=K_n/2+1}^{K_n} V^* - V^{\pi_{n,k}^E}] + \mathbb{E}[\sum_{n=1}^N \sum_{k=K_n/2+1}^{K_n} V^* - V^{\pi_{n,k}^E}]$$

$$\leq 2K_0 H + 3\mathbb{E}[\sum_{n=1}^N \sum_{k=K_n/2+1}^{K_n} V^* - V^{\pi_{n,k}^E}] \qquad (18)$$

Therefore, all we need to do is to upper bound the second part of Eq.(18). As a result of Lemma G.2, we can apply Lemma F.9 to upper bound the regret of the policy sequence $\{\{\pi_{n,k}^E\}_{n=1}^N\}_{k=K_n/2+1}^{K_n}$,

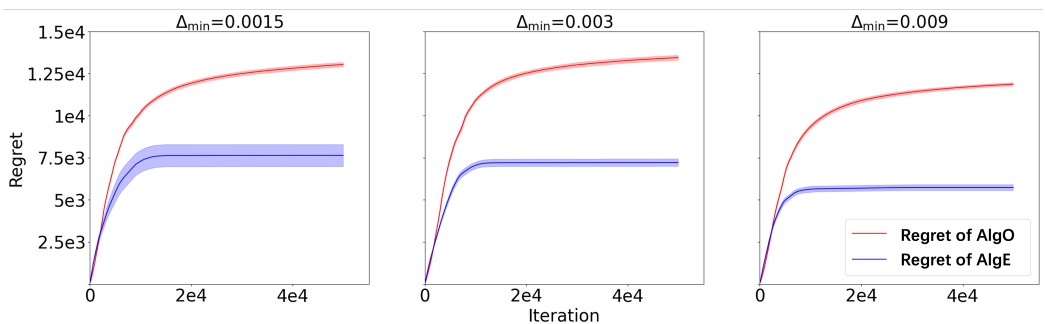

Figure 2: Simulation results with $S = A = H = 5$ and different $\Delta_{\min}$, averaged over 10 different random seeds. Error bars show double the standard errors, which correspond to 95% confidence intervals. Our choice of $\text{Alg}^{\text{E}}$ can achieve constant regret as predicted by theory. We can also see the tendency that larger $\Delta_{\min}$ will result in smaller accumulative regret.

since the Cond. 4.6 is satisfied when generating those policies. Therefore, the Pseudo regret $\mathbb{E}[\sum_{n=1}^{N} \sum_{k=K_n/2+1}^{K_n} V^* - V^{\pi_{n,k}^{\text{E}}}]$ can be upper bounded by extending the results in Thm. 4.9 here, and we finish the proof.

$\square$

## H  Experiments

### H.1  Experiment Setup

**Environment**  We test our algorithms in tabular MDPs with randomly generated transition and rewards functions. To generate the MDP, for each layer $h$ and each state action pair $(s_h, a_h)$, we first sample a random vector $\mathbb{P}(\cdot|s_h, a_h)$, where each element is uniformly sampled from $\{1, 2, 3..., 10\}$, and then normalize it to a valid probability vector. Besides, the reward function is set to $\xi/10$ where $\xi$ is randomly generated from $\{1, 2, ..., 10\}$ to make sure it locates in $[0, 1]$.

**Algorithm**  We implement the StrongEuler algorithm in [Simchowitz and Jamieson, 2019] as $\text{Alg}^{\text{O}}$ and construct the same adaptive bonus term (Alg. 3 in [Simchowitz and Jamieson, 2019]) for $\text{Alg}^{\text{E}}$ to match Cond. 4.4. Although for the convenience of analysis, in our Framework 1, we do not consider to use the data generated by $\text{Alg}^{\text{E}}$, in experiments, we use both $\tau^{\text{O}}$ and $\tau^{\text{E}}$, which slightly improves the performance. Besides, in practice, we observe that the bonus term is quite loose, and it will take a long time before the estimated $Q/V$ value fallen in the interval $[0, H]$, which is the value range of true value functions. Therefore, we introduce a multiplicator $\alpha$ and adjust the bonus term from $b_{k,h}$ to $\alpha \cdot b_{k,h}$, and set $\alpha = 0.25$ for both $\text{Alg}^{\text{O}}$ and $\text{Alg}^{\text{E}}$.

### H.2  Results

We test the algorithms in tabular MDPs with $S = A = H = 5$[5]. Although the minimal gap $\Delta_{\min}$ is hard to control since we generate the MDP in a random way, we filter out three random seeds in MDP construction, which correspond to minimal gaps (approximately) equal to 0.0015, 0.003 and 0.009, respectively. We report the accumulative regret in Fig. 2.

As predicted by our theory, $\text{Alg}^{\text{E}}$ can indeed achieve constant regret in contrast with the continuously increasing regret of $\text{Alg}^{\text{O}}$, which demonstrates the advantage of leveraging tiered structure.

---

[5]The code can be find in `https://github.com/jiaweihhuang/Tiered-RL-Experiments`.