# OpenReview forum: "Tiered Reinforcement Learning: Pessimism in the Face of Uncertainty and Constant Regret"
_NeurIPS.cc/2022/Conference — NeurIPS 2022 Accept_

### Official Review · Reviewer_YYUn · 2022-07-07

**Rating:** 6
**Confidence:** 3
**Soundness:** 4 excellent
**Presentation:** 3 good
**Contribution:** 3 good

**Summary:**

- They investigate a 2-tiered play structure in the stochastic bandit and MDPs, one of which is the "exploiters".
- Those two groups play the same instance simultaneously, and the exploiter group was shown to achieve a (gap-dependent) constant regret by using a conservative strategy (LCB) and exploit information from the other group (explorer). The regret of the explorer remains (near-)optimal.
- Although there exists studies of decoupled exploration-exploitation in bandit settings, the differential is that it can be naturally extended to MDPs by using an algorithm (UCB/LCB) which does not use sample weights, resulting in high compatibility to the MDPs.

**Questions:**

- Are there concrete examples of exploiter groups who share a homogeneous model with explorers?

**Limitations:**

- Limited to the setting where two parties continue to play at the same time.
- Dividing into tiers and favoring one side (the exploiter group) would imply that there are differences that should be favored (e.g., they are wealthy, paying users). Such differences may have an impact on transitions and reward distributions, but they are using a homogeneous model. Therefore, it remains to be seen if there are specific examples that would have an impact in realistic problems.
- In that case, since the method only learns the data from the explorer group, the exploiter group might suffer a linear regret.

**Strengths And Weaknesses:**

- Originality: somewhat weak
	- The idea of decoupled exploration-exploitation was inherited from previous work. Since the previous work considers adversarial setting also, its method uses sample-weights, which has a risk of the "curse of horizon" when extended to MDPs. It is not a surprising idea to use UCB/LCB to avoid the problem.
- Quality: excellent
	- Their analysis is convincing since it proves the problem-dependent constant regret as well as the impossibility of the problem-independent constant regret.
- Clarity: good
	- This paper is well organized and reads naturally from a simple MAB setup to a table RL setup.
- Significance: good
	- It is a significant contribution that a constant regret can be achieved also in reinforcement learning by decoupling the explorers and the exploiters.

---

> ### Author Response · Authors · 2022-08-01
> **Response to Reviewer YYUn**
>
> We thank the reviewer for the valuable comments.
>
> ### Concrete Examples
> We believe that in the scenarios mentioned in Sec. 1 (line 30-43), there are some real-world cases where the factors determining risk tolerance are **independent** w.r.t. the transition and reward of the model, so that we can assume that both two groups share the model.
> For example, in some medical treatment scenarios, people's tolerance on risk is independent w.r.t. their body condition and their reaction to the treatment plan (i.e. the model).
>
>
> Moreover, given the connection between ``decoupling exploration and exploitation setting'' [1,2] and our framework, our algorithms and guarantees can be directly applied to their settings, e.g. ultra-wideband (UWB) communications in [1].
>
> We also have a general remark about our assumption on model sharing, please check our general response above.
>
> [1] Orly Avner, Shie Mannor, and Ohad Shamir. Decoupling Exploration and Exploitation in Multi-Armed Bandits
>
> [2] Chloé Rouyer, Yevgeny Seldin. Tsallis-INF for Decoupled Exploration and Exploitation in Multi-armed Bandits

---

> > ### Comment · Reviewer_YYUn · 2022-08-09
> > **Re:**
> >
> > > Concrete Examples
> > > ･･･people's tolerance on risk is independent w.r.t. their body condition and their reaction to the treatment plan
> >
> > In the medical case, it seems unnatural for me to assume risk-seekers, who want to suffer not only high variance but also low expectation of the outcome (high regret). Risk appetite is usually a tradeoff between high returns and low variances.
> >
> > > Moreover, given the connection between ``decoupling exploration and exploitation setting'' [1,2] and our framework, our algorithms and guarantees can be directly applied to their settings, e.g. ultra-wideband (UWB) communications in [1].
> >
> > Should state transition be considered in UWB problem? The extension to the MDP seems a bit application-oriented compared to the earlier work on MAB, but it does not seem to have broad applicability. Therefore, while interesting, it cannot be rated as absolutely worthy of publication in top venues. With this, I keep the overall rate of 6.

---

> > > ### Author Response · Authors · 2022-08-09
> > > **Response to Further Questions**
> > >
> > > Thanks for letting us know your further concerns.
> > >
> > > ### The medical example
> > >
> > > We are feeling that, maybe similar to Reviewer e8u5, the ''risk'' considered by reviewer is separated from and orthogonal to that we consider in the paper. Here we make a brief explanation and please also refer to our response to Reviewer e8u5 for more clarification.
> > >
> > > By risk, we refer to the risk resulting from **uncerntainty during the learning process**. Take the medical treatment setting as a concrete example, we do not know the exact value of each treatment plan, and we can only estimate it by data with some uncerntainty. Consider the case when we have two treatment plans, and suppose given the history data, plan 1 has a good estimated expected value and also low uncerntainty (small confidence interval), while plan 2 has high uncerntainty (large confidence interval) but its true expected value has chance to be higher than plan 1. By taking plan 2, the patient may suffer the risk resulting from uncertainty (because of limited data). In that case, some patients will still prefer plan 2 because they value more about the chance that plan 2 has higher true value and we refer them as $G^O$, while another group of patients may prefer plan 1 to avoid the risk resulting from uncertainty and we refer them as $G^E$.
> > >
> > > Moreover, in fact, under our meaning of risk, our risk-tolerated group $G^O$ do not need to ''suffer not only high variance but also low expectation of the outcome''. We only expect $G^O$ to balance the exploration and exploitation as normal optimal online learning algorithms (see our summarized objective in lines 63-64). In our algorithms, while establishing the provable benefits for $G^E$, we also constrained the regret by $G^O$ is near-optimal.
> > >
> > >
> > > ### The UWB setting
> > > Thanks for making it clear. The original UWB example is a bandit setting, since the original ''decoupling exploration and exploitation'' paper only focused on bandit setting. We believe the MDP setting extended by considering state transition is reasonable and practical. We used UWB as an example in our rebuttal just in order to highlight that, given any practical MDP examples in decoupling setting (including those extended from bandit setting), our results can be directly applied into them.

---

> > > > ### Comment · Reviewer_YYUn · 2022-08-10
> > > > **Re:**
> > > >
> > > > > plan 2 has high uncerntainty (large confidence interval) but its true expected value has chance to be higher than plan 1.
> > > >
> > > > Such a type of uncertainty may be related to what is called "ambiguity" in decision-making theory. Anyway, people usually avoid an arm with large confidence interval (called ambiguity aversion) especially in safety-related areas such as medical treatments. Assuming ambiguity-seekers is not very convincing for me.
> > > >
> > > > > we also constrained the regret by $G^O$ is near-optimal
> > > >
> > > > If half are explorer, this seems obvious, considering only that group. The expected value of reward for the $T$-th person (in groups E and O combined) seems to worsen as the proportion of group E increases.
> > > >
> > > > I just want to emphasize that I fully recognize the contribution in the theoretical aspect. However, if you are aiming for more points in terms of practicality, such as citing some use-cases that actually have a preference for ambiguity.

---

### Official Review · Reviewer_fFex · 2022-07-10

**Rating:** 7
**Confidence:** 4
**Soundness:** 4 excellent
**Presentation:** 3 good
**Contribution:** 3 good

**Summary:**

This paper formulated and studied the tiered RL problems where the uses are divided into two groups and treated separately. They showed that the constant and log K regret are achievable while keeping the online algorithm near-optimal for bandit and RL settings, respectively.

**Questions:**

1. I have a general question about the Tiered RL Framework. Under the assumption that $Env^O = Env^E,$ if the definition of regret is the same for both groups, why do different groups' risk tolerances matter? Is exploration still necessary if the $Alg^E$ can already achieve constant regret after certain timesteps/episodes? We are eventually looking for an algorithm that can give us lower regret. Then maybe it is also worth looking at how many samples are needed by collecting with the online algorithm such that a constant regret is achievable by using an exploitation algorithm.

2. Since PVI and LCB are widely used in offline settings, can the authors clarify what the novel technical contributions of this paper are?


**Limitations:**

I don't think this paper has any potential negative societal impact.

**Strengths And Weaknesses:**

**Strengths**

1. This paper is well written and organized, and the insights behind the main results are well presented.
2. Both MAB and RL settings are considered in this paper. The regret bound is comparable with the best existing results.

**Weakness**
1. The model assumes the users from two groups share the same transition and reward function, there is still some gap between the model and reality.
2. It would be great if the framework could model the total "toleration budgets" of different groups.

---

> ### Author Response · Authors · 2022-08-01
> **Response to Reviewer fFex**
>
> We thank the reviewer for the valuable comments.
>
> ### Same transition and reward function
> Please check our additional general response above.
>
> ### Budgeted setting
> We agree that this is an interesting future direction which our work builds the foundation for, and we also briefly mentioned it on Line 331.
>
> ### Question about the framework
>
> > why do different groups' risk tolerances matter?
>
> That the definitions of regret are the same for two groups only means that two groups use the same way to measure the performance of their algorithms. The groups' risk *tolerances* are about how much **total** regret they expect to experience over the entire learning process, i.e., one group may prefer a lower total regret than usual.
>
> > Is exploration still necessary...?
>
> Yes, continuous exploration is necessary. The main reason is that we target at achieving constant regret for arbitrarily large $K$ (recall $K$ is the total number of episodes). If the exploration stops after some constant $k_0$ (in comparison to $K$ which is ever-growing), then the failure probability $\delta_{k_0}$ (i.e., $Alg^E$ fails to identify the optimal action(s)) will also be a constant relative to $K$. However small $\delta_{k_0}$ is, it is still a constant bounded away from $0$, and the failure event's contribution to the regret becomes $O(\delta_{k_0} \cdot (K-k_0))$, which is linear in $K$.
>
> More technically speaking, in order to show constant regret, we require not only the RHS of Eq.(1) to decay to zero with a high probability $1-\delta_k$ after some $k \geq K_0$ but also the accumulative failure rate $\sum_{k=K_0+1}^K \delta_k$ is bounded by constant, especially for large $K$, which is guaranteed only if $Alg^O$ continues to explore.
>
>
> ### Technique contributions
> Although PVI and LCB are widely used, coordinating pessimism and optimism algorithms together to achieve our goal is a novel setup that has not been studied before.
> We also briefly highlight our novel technical contributions here and please refer to Sec. 1 for more details:
>
> *   To our knowledge, the high-probability bounds Lemmas 4.2 and 4.3 are new in the stochastic bandit setting.  $O(k^{-O(\alpha)})$ (with $\alpha > 1$) is carefully chosen to guarantee the accumulative failure rate is bounded by constant, i.e. $\sum k^{-O(\alpha)} < +\infty$.
>     Besides, we contribute Lem. D.1 to establish constant regret for $Alg^E$, which can outperform [1] in some cases (see the discussion between lines 184-191).
>  *   There is also some novelty in Thm. 4.5. Previous literature only focused on gap-dependent bound for optimistic algorithms, while we are the first to study pessimism-based algorithms.
>
> * We contribute Thm. 4.7, which bridges low regret and high occupancy on optimal state actions, and our result is general and holds for arbitrary policy sequences $\pi_1,...,\pi_k,...$.
>     Besides, Thm. 4.8 is a novel observation to overcome the difficulty occurring when there are multiple optimal policies (also see the discussion between lines 231-233).
>
>
> [1]: Chloé Rouyer, Yevgeny Seldin. Tsallis-INF for Decoupled Exploration and Exploitation in Multi-armed Bandits.

---

### Official Review · Reviewer_e8u5 · 2022-07-12

**Rating:** 5
**Confidence:** 3
**Soundness:** 3 good
**Presentation:** 3 good
**Contribution:** 3 good

**Summary:**

This work focused on reinforcement learning and proposed a novel framework to utilize the tiered structure in applications. More specifically, the users are divided into two groups: a risk-tolerant group with the UCB-types algorithm and a risk-averse group with the LCB-types algorithm. With this structure, the author shows that there is nearly no advantage in the gap-independent instance but can improve the regret guarantee from log-T to constant in the gap-dependent instance.


**Questions:**

1. For both Algorithm 2 and Algorithm 3, only the data from online policy $\pi^O$ are used to design the policy, and other data from policy $\pi^E$ are ignored, which seems will deteriorate the algorithm performance. Is there some intuition behind it or just to protect the personal privacy of group E?

2. For the Gap-dependent setting, the author mentions recent advances in regret guarantees for reinforcement learning. More recent advanced works have focused on using the function approximation techniques in MDPs. For instance, He et al. [2022] first obtained a logarithmic regret with the gap-dependent setting. Therefore, it is better if the author can mention them or have some discussion about them.

He J, Zhou D, Gu Q. Logarithmic regret for reinforcement learning with linear function approximation. ICML2021



**Limitations:**

This paper provides theoretical guarantees for learning linear bandit and linear mixture MDP. There is no negative societal impact.

**Strengths And Weaknesses:**

Strength:

1. This work proposed a novel structure to utilize the tiered structure in applications and can improve the performance to constant regret guarantee with the gap-dependent instance.

2. This work provides a theoretical lower bound for the gap-independent instance and shows that tiered can provide nearly no help in this case.

3. This work first considers the pessimism algorithm in online reinforcement learning and combines it with the optimistic rule, which may be of independent interest.

Weakness:

1. The algorithms require the initial state to be fixed ($s_1$), which is much more restrictive than previous work in reinforcement learning.

2. The intuition of introducing a tiered structure is not clear. For the tiered structure, the author mentions that users can be divided into multiple groups depending on their different preferences and tolerance of the risk that results from the necessary exploration to improve the policy. However, from the setting, there is no difference between these two groups in reward, transition, or regret. It is unclear what the critical difference between risk-tolerant and risk-averse people is. It is also unclear why we need to introduce the tiered structure and consider the two groups of people individually. A much more reasonable structure to represent different preferences and tolerance is introducing different rewards for different groups.

---

> ### Author Response · Authors · 2022-08-01
> **Response to Reviewer e8u5**
>
> We thank the reviewer for the valuable comments.
>
> ### Fixed initial state is a simplification without loss of generality
>
> Given an arbitrary episodic MDP $M$ with random initial distribution $\mu_0$ and horizon $H$, one can convert it to another MDP $M'$ with a fixed initial state and horizon $H+1$ by introducing a new fixed initial state which can transit to the original initial states of $M$ according to the $\mu_0$ regardless of action. Even without the above conversion, our results easily generalize to random initial states by a slight modification of the proofs.
>
> ### The setting: the reviewer talks about something different from and orthogonal to our setup
>
> > there is no difference between... It is also unclear ... A much more reasonable structure...
>
> The type of difference the reviewer mentioned (''introducing different rewards for different groups'') is reasonable and interesting by itself, but it is separated from and **orthogonal to** the type of difference we consider in the paper. Let us explain: if the agent (the system/algorithm) has full knowledge of the users (i.e., the MDP transitions and rewards are fully known), the two groups might still prefer different policies due to their risk preferences associated with the **randomness of MDP transitions and rewards**---this is what reviewer is talking about, and related studies can be found in the areas of e.g., risk-sensitive RL.
>
> While this is a very reasonable consideration, we are considering something orthogonal: we consider all users have the same MDP transitions and rewards, so when the agent has full information, it will treat all users equally. That said, differences between groups can still exist, because our agent does **not** have full information from the beginning. Rather, it is a learning agent and can exhibit various different behaviors when they interact with the users to learn their transitions and rewards. What our paper is concerned about is the user's risk preference over **the algorithm's learning behavior**. Note that studying user experiences about algorithm's learning behavior (instead of when all the information is known) is very prevalent, e.g., in the fairness ML literature.
>
> So to summarize, we consider the user's risk preference about the algorithm's learning behavior. The reviewer's suggestion is also interesting and can be potentially combined with our notion of risk preference to form a more realistic and complicated setting, and our study of risk preference over learning behaviors provides the foundation for studying this more complicated setting.
>
> As a final remark, the reviewer mentioned that
> > there is no difference between these two groups in reward, transition...
>
> Please check our additional general response above.
> Besides, even with user-group variability, our primary concern in this line of research is still the risk preference of the algorithm's learning behaviors.
>
> ### Not use data collected from $\pi^E$
>
> We have provided some explanation in Sec. 1 (lines 52-55). Briefly speaking, our main objective is to show the benefits of leveraging the tiered structure by designing algorithms with advantages in terms of regret, and we have already achieved this goal even after ignoring the data collected from $\pi^E$.  Of course, utilizing data from both groups should intuitively help (and is probably a good idea practically), but it will complicate the analysis without necessarily improving our results, as the data from $\pi^E$ lacks exploration and provides little additional information. Such a situation is quite common in RL theory, where we discard some data in theory (which could be practically useful) for clean concentration analysis.
>
> ### Additional reference
>
> Thanks for pointing it out. We will cite and add a discussion about that paper after acceptance.

---

> > ### Comment · Reviewer_e8u5 · 2022-08-08
> > **Thank you for your answer**
> >
> > Thank you for your answer, I do not have further questions and I have increased my score.

---

### Official Review · Reviewer_Tg34 · 2022-07-14

**Rating:** 7
**Confidence:** 3
**Soundness:** 3 good
**Presentation:** 4 excellent
**Contribution:** 3 good

**Summary:**

The paper proposes a new framework for the reinforcement learning problem (specifically, the MAB and finite-horizon tabular MDP) when the instances can be divided into two groups; one group which tolerates risky action selections for the sake of exploration, and another group for which the agent just focuses on maximizing the rewards. A goal is to achieve the near-optimal O(logT) regret for the first group while achieving a time-horizon-independent constant regret for the second group.

Under the assumption that the underlying distributions of the sate, action, and reward are identical for the two groups, and that the proportion of the number of instances in the first vs second group is constant and fixed, the authors provide a framework algorithm which achieves the goal with high probability.

Specifically, the core of these algorithms is to make action decisions according to the "pessimism in the face of uncertainty" principle for the second group. For the bandit problem, the authors propose a specific algorithm which employs the usual UCB algorithm for the first group and select the maximizer of the LCB (Lower confidence bound) for the second group.

**Questions:**

Questions

It would be good to have more details on the use of clipping operator. The authors use it in the algorithm but just refer the explanation to other papers

**Limitations:**

I did not find any discussion on societal impact .

**Strengths And Weaknesses:**

Strengths:

1. The authors provide detailed lower bound analysis for this problem. Specifically for the gap-independent case, the authors provide an example where it is impossible to do better than employing the near-optimal regret reinforcement learning algorithm to the whole instance sequence without making any distinction between the two groups.



2. The authors derive the regret guarantee for the "framework" algorithm rather than just a specific algorithm. The framework requries to use a near-optimal regret reinforcement learning algorithm (that satisfies Condition 4.6) for the first group and then use the pessimism in the face of uncertaint principle (satisfying Condition 4.4) for the second group. As long as Conditions 4.6 and 4.4 are satisfied, the algorithm can be diversified according to the specific inner algorithms used. Hence the framework can apply to a wide range of problems with different kinds of reward functions.



3. The regret guarantee hold without the assuming the uniqueness of the optimal policy.

---

> ### Author Response · Authors · 2022-08-01
> **Response to Reviewer Tg34**
>
> We thank the reviewer for the valuable comments.
>
> Regarding your question on the clipping operator: it is defined in Line 246 by $\text{Clip}[x|\epsilon]:=x\cdot \mathbb{I}[x\geq \epsilon]$, and is first used in the upper bound of the (one-step) sub-optimality gap in Eq.(1) in Thm. 4.5. As we briefly introduced in Sec. 4.2.3, $N_{k,h}(s_h,a_h)$ will gradually increase for those $s_h,a_h$ with non-zero $d^{\pi^*}(s_h,a_h)$. Hence, the clipping operator will take effect after some $k\geq k_0$, i.e. $\mathbb{I}[\cdot \geq \epsilon_{Clip}] = 0$, which results in zero sub-optimality gap, and therefore constant regret.

---

### Author Response · Authors · 2022-08-01
**General Response to the Concern about our Assumption on the Similarity of Models in Different Groups**

We note that some reviewers have common concerns about our assumption that different groups share the model, and therefore, we provide a general response here.

We agree that all users being exactly the same is a relatively restrictive assumption.
However, we remark similar (or exactly the same) transitions and rewards enable knowledge transfer/sharing between the different groups. If they are completely different in unrelated ways, there is no point considering both groups together.
Besides, we believe our results can be extended to a more general setting where different groups only share a similar **hidden** model with the help of feature extractors (i.e. function mappings from original state action space to hidden feature space), and in Line 335 we also mentioned future directions about a contextual setting that allows variability across user groups.

---

### Meta-Review · Area_Chair_1VzU · 2022-08-27

**Recommendation:** Accept
**Confidence:** Certain

**Metareview:**

The new two-group RL framework is interesting, even though it is somewhat restricted to assume the exact same model for both groups. Both the gap-independent and the gap-dependent settings are discussed properly, with lower and upper bounds. Overall we believe that the paper is worth publishing at NeurIPS.

**Award:**

No

---

### Decision · Program_Chairs · 2022-09-14

Accept